# MIA: A Framework for Certifiably Robust Time-Series Classification and Forecasting Against Temporally-Localized Perturbations

## Abstract

Recent literature demonstrates that times-series forecasting/classification are sensitive to input perturbations. However, the defenses for time-series models are relatively under-explored. In this paper, we propose **M**asking **I**mputing **A**ggregation (MIA), a plug-and-play framework to provide an arbitrary deterministic time-series model with certified robustness against temporally-localized perturbations (also known as $\ell_0$-norm localized perturbations), which is to our knowledge the first $\ell_0$-norm defense for time-series models. Our main insight is to let an occluding mask move across the input series, guaranteeing that, for an arbitrary localized perturbation there must exist at least one mask that completely occlude the perturbed area, so that the prediction on this masked series is certifiably unaffected. MIA is flexible as it still works even if we only have the query access to the pretrained model. To further validate the superior effectiveness of MIA, we specifically compare MIA to two baselines extended from prior randomized smoothing approaches. Extensive experiments show that MIA yields stronger robustness.

## 1 Introduction

Time series forecasting/classification (TSF/TSC) have been widely applied to help businesses make informed decisions and plans (Miyato et al., 2017; Zhou et al., 2019; Schlegl et al., 2019; Park et al., 2018). However, a wide range of literature demonstrate that time-series models are vulnerable to adversarial input perturbations (Connor et al., 1994; Gelper et al., 2010; Ding et al., 2022; Yang et al., 2020; Dang-Nhu et al., 2020; Oregi et al., 2018; Han et al., 2020), e.g., an elaborately designed imperceptible perturbation could control the prediction (Karim et al., 2020; Fawaz et al., 2019). So far related literature is mainly focusing on detecting the outliers (Ruff et al., 2018; Yairi et al., 2017), the adversarial robustness of time-series models is relatively under-explored, especially $\ell_0$-norm robustness, e.g., (Yoon et al., 2022) only explore the $\ell_2$-norm adversarial robustness for probabilistic forecasting models. In the present work, we focus on the robustness against temporally-localized perturbations, as we notice there already exists corresponding powerful attacks (Yang et al., 2022).

Generally, defenses can be divided into two types, heuristic defenses and certified defenses. Heuristic defense can yield better empirical robustness but lack robustness guarantees. From the experience on image classification (Athalye et al., 2018; Carlini & Wagner, 2017; Athalye & Carlini, 2018), the heuristic defenses would be useless when confronted with the newly designed adaptive attacks, e.g., Athalye et al. (2018) leverage Backward Pass Differentiable Approximation technique to successfully circumvent almost all the heuristic defenses at that time. To end such a "cat and mouse" game between the adaptive attacks and the heuristic defenses, the concept of certified defense is proposed, with unbreakable robustness certificates.

Current certified defenses can produce robustness certificates but often require the user to retrain the base model from scratch, e.g., Yoon et al. (2022); Li et al. (2020); Cohen et al. (2019) retrain the base model as these defenses do perform poorly on naturally-trained models. The requirement for retraining could bring additional challenges when it comes to the real-world deployments. In addition, the certified defenses on sequence-based data are quite under-explored, since almost all the certified defenses are focusing on matrix-based data (e.g. image).

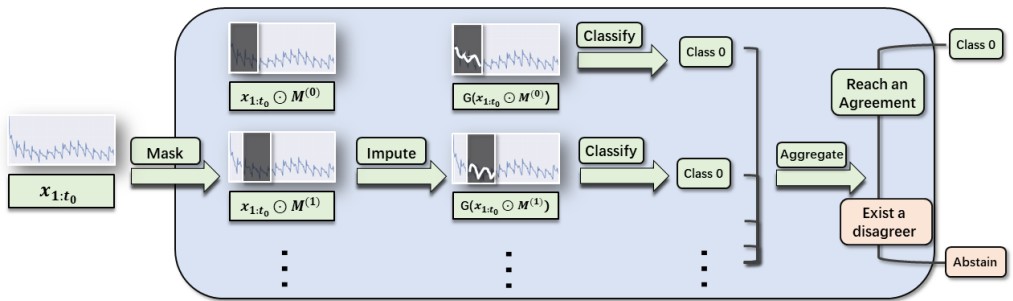

Figure 1: Overview of MIA pipeline. Inputted a series $\mathbf{x}_{1:t_0}$, MIA first masks different periods of $\mathbf{x}_{1:t_0}$ to construct the masked series $\mathbf{x}_{1:t_0} \odot \mathrm{M}^{(k)}, k = 0, \ldots, M$. Then MIA imputes the masked series with the imputation model $G(\cdot)$. We classify the imputed series with the pretrained model. If the predictions of all the imputed series are *Class 0*, MIA will return *Class 0* with the robustness guarantee that the output is clean, otherwise MIA will return **Abstain**.

To address these issues, in this paper, we propose **M**asking **I**mputing **A**ggregation (MIA), a flexible framework to arm an arbitrary TSF/TSC deterministic model with robustness certificates against temporally-localized perturbations. Different from the requirement for retraining in prior defenses, MIA only an imputation model for recovering the masked areas, which can be easily learned in an unsupervised setting. Specifically, MIA works as follows: 1) **masking:** MIA first masked series via sliding a mask through the input series; 2) **imputing:** MIA imputes the masked series with the imputation model; 3) **aggregation (checking agreement):** MIA only returns the the class if the pretrained model outputs the same for all the imputed series, otherwise returns Abstain. With the above three steps, we can guarantee that all the predictions from MIA is clean. Furthermore, we compare MIA to two baselines extended from randomized smoothing, as we notice that randomized smoothing has achieved a widespread success in defending different adversarial attacks. **The contributions are:**

**1)** We propose MIA, a plug-and-play framework to arm an arbitrary TSF/TSC model with certified robustness against temporally-localized perturbations, which is to our knowledge the first $\ell_0$-norm certified defense in time series domain.

**2)** We propose randomized masked training, a specialized training algorithm for training the imputation model of MIA, to further boost the performance of MIA.

**3)** We compare MIA to two baseline methods comprehensively on three aspects. 1) robustness: extensive experiments on different datasets validate that superior robustness of MIA. 2) Practicality: MIA is stronger as it is plug-and-play and do not require retraining. 3) Inference cost: the inference time of MIA is comparable to the time cost of two baselines.

## 2 RELATED WORK

**Heuristic defenses for time-series models.** Prior works on robust TSF/TSC can be divided into two general categories: outlier detection and deep learning. The former is to filter the outliers in a statistical way, including k-Means clustering (Yang et al., 2017), one-class SVM clustering (Schölkopf et al., 2001), Kalman filters (de Bézenac et al., 2020) and support vector data description (Tax & Duin, 2004). The latter leverages the strong representation ability of neural networks to recover the perturbed series, including robust feature-based approaches (Guo et al., 2016; Yang & Fan, 2022), reconstruction-based methods (Li et al., 2021; 2019; Xu et al., 2018; Schlegl et al., 2019), GNN-based methods (Zhao et al., 2020; Deng & Hooi, 2021), association discrepancy (Xu et al., 2022), LSTM-based methods (Hundman et al., 2018; Tariq et al., 2019). However, these empirical methods lack robustness guarantees, hinting that they would be meaningless once a new adaptive attack is found. For that reason, certified defenses are crucial because their mathematical robustness certificates are permanently unbreakable.

**Certified adversarial defenses.** In the field of image classification, there has been much work on the certified defenses, including randomized smoothing (Cohen et al., 2019; Salman et al., 2020), convex polytope (Wong & Kolter, 2018), CROWN-IBP (Zhang et al., 2019) and Lipschitz bounding (Cisse et al., 2017). Among them, the $\ell_0$-norm defenses include derandomized smoothing (Levine & Feizi, 2020a), randomized ablation (Levine & Feizi, 2020b; Zhang et al., 2020) and a series of mask-based

defenses (Xiang & Mittal, 2021; McCoyd et al., 2020; Han et al., 2021; Xiang et al., 2021; 2022). *In stark contrast, the certified defenses for time-series data are rarely explored.* To our knowledge, (Yoon et al., 2022) and (Li et al., 2020) are the only two defenses that produce $\ell_2$-norm robustness certificates, but a common downside is that they both additionally require retraining the base model over Gaussian augmented samples, which imposes a large amount of additional training costs.

## 3 PRELIMINARIES

**Time series classification (TSC).** The time series classification is modeled as: inputted a $t_0$-length series (denoted by $\mathbf{x}_{1:t_0} = [x_1, x_2, \ldots, x_{t_0}]$), TSC model returns a class $f(\cdot) : \mathbf{x}_{1:t_0} \to y$.

**Time series forecasting (TSF).** Given the "past observations" $\mathbf{x}_{1:t_0}$, the forecasting model returns the "future values" $f(\cdot) : \mathbf{x}_{1:t_0} \to \mathbf{x}_{t_0+1,t_0+\tau}$. In this paper we mainly focus on the classic and commonly studied short-term forecasting setting (Ke et al., 2017), which is to forecast a single time point $f(\cdot) : \mathbf{x}_{1:t_0} \to \mathbb{R}$ (not necessarily the next point $x_{t_0+1}$). The short-term forecasting problem is sufficiently representative as the problem of long-term forecasting $f(\mathbf{x}_{1:t_0}) \to \mathbf{x}_{t_0+1:t_0+\tau}$ can be decomposed into $\tau$ short-term forecasting subproblems, in which the $i$-th ($i = 1, \ldots, \tau$) forecaster predicts the $(t_0 + i)$-th time point. We discuss the multivariate tasks later in this paper.

**Definition 1** (Temporally-localized perturbation $\boldsymbol{\delta}_{[t_{\mathrm{adv}}+1:t_{\mathrm{adv}}+L_{\mathrm{adv}}]}$). In a temporally-localized perturbation attack, the adversary is allowed to *perturb an arbitrary subseries w.r.t. the given $\ell_0$-norm constraint*. Let $L_{\mathrm{adv}}$ be the $\ell_0$-norm constraint on the localized perturbation. We can formulate all the perturbed series w.r.t. the $\ell_0$-norm constraint as follows:

$$
\begin{aligned}
&\mathbf{x}_{1:t_0} + \boldsymbol{\delta}_{[t_{\mathrm{adv}}+1:t_{\mathrm{adv}}+L_{\mathrm{adv}}]} \\
=&\mathbf{x}_{1:t_0} + [0, \ldots, 0, \delta_{t_{\mathrm{adv}}+1}, \ldots, \delta_{t_{\mathrm{adv}}+L_{\mathrm{adv}}}, 0, \ldots, 0] \\
=&[x_1, \ldots, \underbrace{x_{t_{\mathrm{adv}}+1} + \delta_{t_{\mathrm{adv}}+1}, \ldots, x_{t_{\mathrm{adv}}+L_{\mathrm{adv}}} + \delta_{t_{\mathrm{adv}}+L_{\mathrm{adv}}}}_{\text{Perturbed subseries}}, \ldots, x_{t_0}]
\end{aligned}
\tag{1}
$$

where unbold $\delta_t$ refers to the single perturbation value added to the $t$-th time point. $t_{\mathrm{adv}} + 1$ and $t_{\mathrm{adv}} + L_{\mathrm{adv}}$ refer to the starting point and the ending point of the perturbation respectively, which explicitly restricts its $\ell_0$ norm as $\|\boldsymbol{\delta}_{[t_{\mathrm{adv}}+1:t_{\mathrm{adv}}+L_{\mathrm{adv}}]}\|_{\ell_0} = (t_{\mathrm{adv}} + L_{\mathrm{adv}}) - (t_{\mathrm{adv}} + 1) + 1 = L_{\mathrm{adv}}$.

**Significance of temporally-localized perturbation.** Temporally-localized perturbation is especially representative in real-world scenarios. Temporally-localized perturbation can represent *short-term volatility* and *local anomaly*, both of which can be regarded as the normal data added with temporally-localized perturbation. The resistance to short-term volatility is important in long-term forecasting/prediction, in which the long-term value is considered unaffected by the short-term volatility. A typical example is the well-known investment philosophy, "Value Investing" (Piotroski, 2000), where the "intrinsic value" of a business is considered to be robust against short-term volatility. Moreover, the detection of local anomaly is practically useful in real-world scenarios. For instance, detecting a subsequent time interval of abnormal heart rate in electronic health records is a problem of local anomaly detection. We can also adopt the method of detecting temporally-localized perturbation for detecting the abnormal network traffic for IoT Time-Series Data. Furthermore, to highlight the risk of temporally-localized perturbations, we empirically show how much a $\ell_0$-norm perturbation can change the output of an undefended forecaster in Appendix. We also compare the attacking performance of $\ell_0$-norm perturbation to $\ell_0$-norm perturbation, and the empirical results suggest that forecasting models might be more sensitive to $\ell_0$-norm perturbations.

## 4 PROPOSED FRAMEWORK: MASKING IMPUTING AGGREGATION

### 4.1 PIPELINE OVERVIEW

MIA includes three steps: 1) masking; 2) imputing; 3) aggregation (checking agreement).

1. **Masking.** We denote a mask by $\mathrm{M}_{[u:v]}$, where $\mathbf{x}_{1:t_0} \odot \mathrm{M}_{[u:v]}$ is replacing the values of $\mathbf{x}_{u:v}$ among $\mathbf{x}_{1:t_0}$ with zeros. Let $L_{\mathrm{mask}}$ be the size of the mask. Inputted a series $\mathbf{x}_{1:t_0}$ and the $\ell_0$ norm of the temporally-localized perturbation $L_{\mathrm{adv}}$, we slide the mask thorough the input series with the step size $\alpha = L_{\mathrm{mask}} - L_{\mathrm{adv}} + 1$, and then obtain the following masked series[1]:

$$
\begin{aligned}
&\mathbf{x}_{1:t_0} \odot \mathrm{M}_{[1+k\alpha \,:\, \min(L_{\mathrm{mask}}+k\alpha, t_0)]}, \ k = 0, \ldots, \lceil (t_0 - L_{\mathrm{mask}})/\alpha \rceil \\
&\text{where} \quad \alpha = L_{\mathrm{mask}} - L_{\mathrm{adv}} + 1
\end{aligned}
\tag{2}
$$

---

[1] $\lceil c \rceil$ returns the smallest integer larger than or equal to $c$

---

**Algorithm 1:** Algorithm of Masking Imputing Aggregation.

---

**Input:** The pretrained TSF/TSC model $f(\cdot)$, the imputation model $G(\cdot)$, the input series $\mathbf{x}_{1:t_0}$, the mask size $L_{\text{mask}}$, the length of temporally-localized perturbation $L_{\text{adv}}$, the discretization parameter $\Delta$ for TSF task.

1 Compute the step size of masking $\alpha \leftarrow L_{\text{mask}} - L_{\text{adv}} + 1$;

2 Generate the masked series via sliding the $L_{\text{mask}}$-size mask
  $\mathbf{x}_{1:t_0} \odot \mathsf{M}_{[1+k\alpha:\min(t_0, L_{\text{mask}}+k\alpha)]},\ k=0,\ldots,\lceil(t_0-L_{\text{mask}})/\alpha\rceil$ ;

3 Utilize the imputation model to impute the masked series
  $\mathbf{x}_{1:t_0}^{(k)} = G(\mathbf{x}_{1:t_0} \odot \mathsf{M}_{[1+k\alpha:\min(t_0, L_{\text{mask}}+k\alpha)]})$;

4 Compute the output (denoted by $y^{(k)}$) for each imputed series, as follows:

$$y^{(k)} = \begin{cases} f(\mathbf{x}_{1:t_0}^{(k)}) & \text{for TSC task} \\ f_{\text{dis}}(\mathbf{x}_{1:t_0}^{(k)}) & f_{\text{dis}}(\cdot)\text{for TSF task} \end{cases}$$

  # $f_{\text{dis}}(\mathbf{x}_{1:t_0}^{(k)})$ is computed as Eq. (5);

5 **if** $y^{(0)} = y^{(1)} = \ldots = y^{(k)}$ **then**
  $\lfloor$ **Output:** $y^{(0)}$.

6 **else**
  $\lfloor$ **Output: Abstain**.

---

We set the step size to $L_{\text{mask}} - L_{\text{adv}} + 1$ for guaranteeing all the temporally-localized perturbations of $L_{\text{adv}}$ can be covered. $\min(L_{\text{mask}} + k\alpha, t_0)$ is to prevent the mask from exceeding $t_0$.

2. **Imputing.** Our second step is to recover the masked values with the imputation model $G(\cdot)$:

$$\mathbf{x}_{1:t_0}^{(k)} = G(\mathbf{x}_{1:t_0} \odot \mathsf{M}_{[1+k\alpha:\min(t_0, L_{\text{mask}}+k\alpha)]})\quad k=0,1,\ldots,\lceil(t_0-L_{\text{mask}})/\alpha\rceil \quad (3)$$

This step is to make $\mathbf{x}_{1:t_0}^{(k)}$ approximate the normal time series, so that the pretrained model could perform similarly on these imputed series. We discuss $G(\cdot)$ later this section.

3. **Aggregation (Checking Agreement).** We input the imputed series $\mathbf{x}_{1:t_0}^{(k)}$ into the pretrained model $f(\cdot)$. If the pretrained model's ouputs on all $\mathbf{x}_{1:t_0}^{(k)}$ reach agreement unanimously, MIA classifier $f_{\text{MIA}}(\mathbf{x}_{1:t_0})$ will output this unanimously approved label/prediction, otherwise output **Abstain** to alert that the input series might have been attacked by the temporally-localized perturbations.

$$f_{\text{MIA}}(\mathbf{x}_{1:t_0}) = \begin{cases} f(\mathbf{x}_{1:t_0}^{(0)}) & f(\mathbf{x}_{1:t_0}^{(0)}) = f(\mathbf{x}_{1:t_0}^{(1)}) = \ldots = f(\mathbf{x}_{1:t_0}^{(\lceil(t_0-L_{\text{mask}})/\alpha\rceil)}) \\ \textbf{Abstain} & \text{Otherwise} \end{cases} \quad (4)$$

**Discretization technique for MIA on TSF.** We note that TSF models are impossible to forecast exactly the identical value on different series, so that MIA would output **Abstain** all the time on TSF. To address this, we substitute the original pretrained forecaster $f(\cdot)$ with its discretized version $f_{\text{dis}}(\cdot)$ in Eq. (4), where $f_{\text{dis}}(\mathbf{x}_{1:t_0}^{(k)}), k=0,\ldots,\lceil(t_0-L_{\text{mask}})/\alpha\rceil$ compute as follow:

$$f_{\text{dis}}(\mathbf{x}_{1:t_0}^{(k)}) = \Delta \cdot \lfloor f(\mathbf{x}_{1:t_0}^{(k)})/\Delta\rfloor \quad (5)$$

where $\Delta$ is a discretization parameter that controls the trade-off between the discretization error and the success rate of achieving agreement. As $\Delta$ decreases, the discretized forecasts retain more information from the original forecasts while the agreement rate decreases. If we take $\Delta = 0.5$, $f_{\text{dis}}(\mathbf{x}_{1:t_0}^{(k)})$ is to round up the value of $f_{\text{dis}}(\mathbf{x}_{1:t_0}^{(k)})$ to the nearest integer.

### 4.2 Discussion on the mask size $L_{\text{mask}}$.

The only requirement of **Masking (Step 1)** is to ensure *for an arbitrary temporally-localized perturbation of $L_{\text{adv}}$, there always exists a mask to occlude that perturbation*. Thus a prerequisite is $L_{\text{mask}} \geq L_{\text{adv}}$. We can control the trade-off between the the imputation quality and the inference cost with $L_{\text{mask}}$. As we increase $L_{\text{mask}}$, the imputation quality will decrease since the number of missing values increases. Meanwhile, the number of masked series decreases subsequently, so the inference cost is reduced. In the extreme case where $L_{\text{mask}} = t_0$ where the masked series are all equal $\mathbf{0}_{1:t_0}$, MIA always outputs $f(G(\mathbf{0}_{1:t_0}))$ regardless of the input series. The imputation quality is extremely poor and the inference cost is the smallest. **The practical implementation of MIA is showed in Algorithm 1.**

**Remark 1** (MIA on Probabilistic Models). We notice a line of time-series forecasting models are probabilistic (e.g., DeepAR (Salinas et al., 2020)), which models the forecasted value $f(\mathbf{x}_{1:t_0})$ as a random distribution $q[y \mid \mathbf{x}_{1:t_0}]$ rather than a single value, as follows:

$$f(\mathbf{x}_{1:t_0}) = \mathbb{E}_{q[x_{t_0+1} \mid \mathbf{x}_{1:t_0}]} [x_{t_0+1}] \tag{6}$$

The exact forecasting value of probabilistic models is inaccessible (prior works perform Monte-Carlo inference for approximation). which makes applying MIA to probabilistic models challenging. Although we can utilize Clopper-Pearson method (Clopper & Pearson, 1934) to estimate the discretized forecasts $f_{\mathrm{dis}}(\mathbf{x}_{1:t_0})$ with a confidence level, the inference cost would be expensive for confidence interval estimation. [2]

### 4.3 Robustness Certificate of MIA

**Proposition 1** (Robustness Certificate of MIA). The forecast/label (not **Abstain**) returned by Algorithm 1 cannot be changed by any temporally-localized perturbation whose $\ell_0$ norm is no larger than $L_{\mathrm{adv}}$ (see proof in Appendix).

**Remark 2** (Robustness Certificate). The robustness certificate is for $f_{\mathrm{MIA}}(\mathbf{x}_{1:t_0})$ rather than $f(\mathbf{x}_{1:t_0})$ because it is almost impossible to derive the certificate for a pretrained model without any requirement. Our aggregation does not allow any tolerance because the certificate would not hold once a disagreement is allowed. Note that, with **Masking (Step 1)**, we can guarantee there exists a masked series that is unaffected, and all other masked series retain the perturbed area. If we allow a disagreer, the ensemble prediction would be totally under the adversary's control, because all except one masked series are perturbed (the only one not affected would become the disagreer). We point out that the certificate also holds for multivariable TSC/TSF. We can easily apply MIA to multivariable tasks through repeating **Masking (Step 1)** and **Imputing (Step 2)** on each variable.

### 4.4 Training Imputation Model $G(\cdot)$

The performance of MIA highly depends on the imputation model $G(\cdot)$. We notice that there already exists much work on time series imputation (Cao et al., 2018; Du et al., 2022; Moritz & Bartz-Beielstein, 2017; Fortuin et al., 2020; Cao et al., 2018; Luo et al., 2019; Yozgatligil et al., 2013). However, all these imputation models aim to recover the discrete missing values, which is not we want. To train an imputation model to recover consecutive missing values, we propose *randomized masked training algorithm*, which minimizes the MSE loss over the masked noisy series, as follows:

$$\mathbb{E}_{\boldsymbol{\delta}_{[1:t_0]} \sim \mathcal{N}(0,\sigma^2)} \ \frac{1}{C+1} \sum_{k=0}^{C} \|G\left((\mathbf{x}_{1:t_0} + \boldsymbol{\delta}_{[1:t_0]}) \odot M_{[1+k\alpha : \min(L_{mask}+k\alpha, t_0)]}\right) - \mathbf{x}_{1:t_0}\|_2^2 \tag{7}$$

where $C = \lceil (t_0 - L_{mask})/\alpha \rceil$ and $\boldsymbol{\delta}_{[1:t_0]} \sim \mathcal{N}(0, \sigma^2)$ is a Gaussian noise series, of which each entry is *i.i.d.* sampled from Gaussian distribution. We specifically add Gaussian noise is to make the imputation model robust to the random noise and avoid overfitting, since prior works (Foster et al., 1992; Passalis et al., 2021; Hwang et al., 1998) show the time series data is generally noisy. We emphasize that we do not add any noise in inference stage.

### 4.5 Comparison to Randomized Smoothing Defenses

Randomized smoothing (Cohen et al., 2019) is a well-know model-agnostic method in the field of certified defenses, which has been applied to defend various types of attacks and achieves superior certified robustness in their respective fields. Comparing MIA to randomized smoothing can better demonstrate the advance of our method. We extend two image-specific randomized smoothing defenses, Derandomized Smoothing (Levine & Feizi, 2020a) and Randomized Ablation (Levine & Feizi, 2020b) to the time series domain, as the baselines.

**Derandomized smoothing for time-series models.** In the time-series version of DS, given a time series $\mathbf{x}_{1:t_0}$ and the base classifier $f(\cdot)$, DS (denoted by $f_{\mathrm{DS}}$) classifies as follows[3]:

$$f_{\mathrm{DS}}(\mathbf{x}_{1:t_0}) = \arg\max_{y \in \mathcal{Y}} \left[ \sum_{\mathbf{x}_{\mathrm{sub}} \in \mathrm{Sub}(\mathbf{x}_{1:t_0}, \eta)} \mathbb{I}\{f(\mathbf{x}_{\mathrm{sub}}) = y\} \right] \tag{8}$$

---

[2]We notice that a recent work (Yoon et al., 2022) derives robustness certificate for probabilistic forecasters, but our definitions of robustness are different. Yoon et al. (2022) bounds the local Lipschitz constant, while our objective is much stricter, aiming to guarantee the forecast is invariant under the perturbation.

[3]$\mathbb{I}\{\}$ is the indicator function.

Table 1: (DistalPhalanxTW) Comparison among three defenses on a TSC dataset.

| Defense | 1 % | 2 % | 3 % | 4 % | 5 % | 6 % | 7 % | 8 % | 9 % | 10 % |
|---|---|---|---|---|---|---|---|---|---|---|
| MIA ($L_{\mathrm{mask}} = 10\%$) | **67.3%** | **66.3%** | **67.3%** | **67.3%** | **66.3%** | **64.4%** | **66.3%** | **64.4%** | **64.4%** | **63.4%** |
| DS ($\eta = 10\%$) | 28.1% | 28.1% | 28.1% | 28.1% | 28.1% | 28.1% | 28.1% | 28.1% | 28.1% | 28.1% |
| RA ($\eta = 10\%$) | 5.8% | 5.8% | 5.8% | 5.8% | 5.8% | 5.8% | 5.8% | 5.8% | 5.8% | 5.8% |
| MIA ($L_{\mathrm{mask}} = 15\%$) | **62.4%** | **65.3%** | **65.3%** | **64.4%** | **64.4%** | **62.4%** | **63.4%** | **65.3%** | **63.4%** | **64.4%** |
| DS ($\eta = 15\%$) | 30.2% | 30.2% | 30.2% | 30.2% | 30.2% | 30.2% | 30.2% | 30.2% | 30.2% | 30.2% |
| RA ($\eta = 15\%$) | 9.4% | 9.4% | 9.4% | 9.4% | 9.4% | 9.4% | 9.4% | 9.4% | 9.4% | 9.4% |

where $\mathrm{Sub}(\mathbf{x}_{1:t_0}, \eta)$ consists of the subsequences $\mathbf{x}_{1:\eta}, \mathbf{x}_{\eta+1:2\eta}, \mathbf{x}_{2\eta+1:3\eta}, \ldots, \mathbf{x}_{t_0-\eta+1:t_0}$. We first let the base classifier make predictions on these subsequences, and then $f_{\mathrm{DS}}(\mathbf{x}_{1:t_0})$ outputs the majority label. The prediction is robust if

$$\sum_{\mathbf{x}_{\mathrm{sub}} \in \mathrm{Sub}(\mathbf{x}_{1:t_0}, \eta)} \mathbb{I}\{f(\mathbf{x}_{\mathrm{sub}}) = \hat{y}\} - \max_{y \neq \hat{y}} \sum_{\mathbf{x}_{\mathrm{sub}} \in \mathrm{Sub}(\mathbf{x}_{1:t_0}, \eta)} \mathbb{I}\{f(\mathbf{x}_{\mathrm{sub}}) = y\} > 2(\eta + L_{\mathrm{adv}} - 1) \quad (9)$$

**Randomized ablation for time-series models.** RA (denoted by $f_{\mathrm{RA}}(\cdot)$) classifies as follows:

$$f_{\mathrm{RA}}(\mathbf{x}_{1:t_0}) = \arg\max_{y \in \mathcal{Y}} \left[ \Pr_{\mathbf{x}_{\mathrm{sub}} \sim \mathrm{Sample}(\mathbf{x}_{1:t_0}, \eta)} [f(\mathbf{x}_{\mathrm{sub}}) = y] \right] \quad (10)$$

where $\mathbf{x}_{\mathrm{sub}} \sim \mathrm{RA}(\mathbf{x}_{1:t_0}, \eta)$ is to randomly sample $\eta$ time points without replacement to construct the subseries $\mathbf{x}_{\mathrm{sub}}$, and ablate all other points. $f_{\mathrm{RA}}(\mathbf{x}_{1:t_0})$ returns the label that $f(\cdot)$ is most likely to classify $\mathbf{x}_{\mathrm{sub}}$ as. $\hat{y} = f_{\mathrm{RA}}(\mathbf{x}_{1:t_0})$ is robust if

$$\Pr_{\mathbf{x}_{\mathrm{sub}} \sim \mathrm{Sample}(\mathbf{x}_{1:t_0}, \eta)} [f(\mathbf{x}_{\mathrm{sub}}) = \hat{y}] > \frac{3}{2} - \frac{\binom{t_0 - L_{\mathrm{adv}}}{\eta}}{\binom{t_0}{\eta}} \quad (11)$$

**Comparison to DS and RA.** We note that the pretrained models of DS and RA make predictions on subseries $f(\mathbf{x}_{\mathrm{sub}})$ instead of normal series. Since the data distribution of the subseries are fundamentally different from the normal data, we can expect that these two defenses would perform poorly on the naturally-trained models. Therefore, we need to train the base classifiers from scratch on the subseries. In stark contrast, MIA is a plug-and-play framework that can be directly applied to TSF/TSC pretrained models. In MIA, the main cost of training stage is preparing the imputation model. We point out that the imputation model of MIA can be trained in an unsupervised manner, saving us from labeling the data. Furthermore, we empirically show that MIA attains a significantly better robustness than DS and RA in Section 5.

## 5 EXPERIMENTS

**Experimental setup.** We evaluate MIA on both TSC and TSF datasets. TSF includes Exchange Rate, Traffic and UCI Electricity (Alexandrov et al., 2019), and TSC datasets include DistalPhalanxTW, MiddlePhalanxTW and ProximalPhalanx (Ismail Fawaz et al., 2019a). We use MLP-Mixer (Tolstikhin et al., 2021), MLP and LSTM (Hochreiter & Schmidhuber, 1997) as the pretrained model. Our experiments are conducted on the clean trainsets, following the common setting of certified adversarial defenses (Yoon et al., 2022; Li et al., 2020; Cohen et al., 2019; Chiang et al., 2020; Zhang et al., 2019). Unless otherwise specified, We use MLP-Mixer as the base model for MIA, DS and RA, and $\Delta = 1.5$. The experiments are conducted on CPU (16 Intel(R) Xeon(R) Gold 5222 CPU @ 3.80GHz) and GPU (one NVIDIA RTX 2080 Ti). More details are omitted to Appendix.

**Evaluation metrics.** For TSC, we evaluate the defense by *certified accuracy* (CA) under the temporally-localized perturbation, which is defined by the fraction of the test samples that are correctly classified and certifiably robust to the perturbation. For TSC, we evaluate the defense by: *forecasting rate* (FR), *mean square error* (MSE) and *mean absolute error* (MAE)[4]. FR is the fraction of the test samples on which MIA outputs the forecast instead of **Abstain**. MSE/MAE measures the mean square error/mean absolute error between MIA forecasts (**Abstain** are excluded) and groundtruth. We omit the evaluation on multivariate tasks to Appendix due to space limitations.[5]

### 5.1 COMPARISON TO PEER METHODS

**Comparison on TSC.** Table 1 reports the certified accuracy of three methods in defending temporally-localized perturbations. The pretrained/base model architectures of three defenses are all

---

[4]We omit the evaluation of MAE to Appendix.

[5]In our experiments, $L_{\mathrm{mask}} = c\%$ or $L_{\mathrm{adv}} = c\%$ or $\eta = c\%$ refer to $c\% \cdot t_0$.

Table 2: (Exchange) Comparison among three certified defenses on TSF dataset.

| Metric | Defense | 1 % | 2 % | 3 % | 4 % | 5 % | 6 % | 7 % | 8 % | 9 % | 10 % |
|---|---|---|---|---|---|---|---|---|---|---|---|
| FR (%) | MIA ($L_{mask} = 10\%$) | **82.2** | **82.2** | **83.2** | **82.2** | **82.2** | **81.2** | **81.2** | **81.2** | **80.2** | **79.2** |
| | DS ($\eta = 10\%$) | 24.8 | 24.8 | 24.8 | 24.8 | 24.8 | 24.8 | 24.8 | 24.8 | 24.8 | 24.8 |
| | RA ($\eta = 10\%$) | 16.8 | 16.8 | 16.8 | 16.8 | 16.8 | 16.8 | 16.8 | 16.8 | 16.8 | 16.8 |
| | MIA ($L_{mask} = 15\%$) | **64.4** | **71.3** | **69.3** | **69.3** | **71.3** | **68.3** | **69.3** | **71.3** | **62.4** | **65.3** |
| | DS ($\eta = 15\%$) | 20.8 | 20.8 | 20.8 | 14.9 | 14.9 | 11.9 | 11.9 | 11.9 | 11.9 | 10.9 |
| | RA ($\eta = 15\%$) | 23.8 | 23.8 | 23.8 | 23.8 | 23.8 | 23.8 | 23.8 | 23.8 | 23.8 | 23.8 |
| MSE | MIA ($L_{mask} = 10\%$) | **0.143** | **0.141** | **0.145** | **0.141** | **0.141** | **0.139** | **0.137** | **0.137** | **0.135** | **0.134** |
| | DS ($\eta = 10\%$) | 0.192 | 0.192 | 0.192 | 0.192 | 0.192 | 0.192 | 0.192 | 0.192 | 0.192 | 0.192 |
| | RA ($\eta = 10\%$) | 0.202 | 0.202 | 0.202 | 0.202 | 0.202 | 0.202 | 0.202 | 0.202 | 0.202 | 0.202 |
| | MIA ($L_{mask} = 15\%$) | **0.126** | **0.134** | **0.130** | 0.132 | 0.137 | 0.132 | 0.129 | 0.130 | 0.123 | 0.125 |
| | DS ($\eta = 15\%$) | 0.144 | 0.144 | 0.144 | **0.065** | **0.065** | **0.069** | **0.069** | **0.069** | **0.069** | **0.060** |
| | RA ($\eta = 15\%$) | 0.248 | 0.248 | 0.248 | 0.248 | 0.248 | 0.248 | 0.248 | 0.248 | 0.248 | 0.248 |

Table 3: Comparison of the inference time (millisecond) of three defenses on TSC datasets.

| Defense \ Model | FCN | | | MLP-Mixer | | | MLP | | | ResNet-18 | | |
|---|---|---|---|---|---|---|---|---|---|---|---|---|
| | 2% | 5% | 10% | 2% | 5% | 10% | 2% | 5% | 10% | 2% | 5% | 10% |
| MIA ($L_{mask} = 10\%$) | 2.0 | 2.0 | 3.0 | 2.7 | 2.7 | 4.6 | 1.7 | 1.7 | 2.7 | 3.8 | 3.8 | 5.3 |
| DS ($\eta = 10\%$) | 0.6 | 0.6 | 0.6 | 2.0 | 2.0 | 2.0 | 0.3 | 0.3 | 0.3 | 2.6 | 2.6 | 2.6 |
| RA ($\eta = 10\%$) | 25.2 | 25.2 | 25.2 | 259.7 | 259.7 | 259.7 | 0.8 | 0.8 | 0.8 | 130.3 | 130.3 | 130.3 |
| MIA ($L_{mask} = 15\%$) | 2.0 | 2.0 | 2.0 | 2.7 | 2.7 | 2.7 | 1.7 | 1.7 | 1.7 | 3.8 | 3.8 | 3.8 |
| DS ($\eta = 15\%$) | 0.6 | 0.6 | 0.6 | 1.9 | 1.9 | 1.9 | 0.3 | 0.3 | 0.3 | 2.6 | 2.6 | 2.6 |
| RA ($\eta = 15\%$) | 25.3 | 25.3 | 25.3 | 260.6 | 260.6 | 260.6 | 0.8 | 0.8 | 0.8 | 130.4 | 130.4 | 130.4 |

Table 4: (Traffic) The performance of MIA on different pretrained models. ($c_1$ $c_2\%$) reports (MSE, FR%) of MIA. **Baseline** is MSE of the pretrained model without MIA. The lowest MSE and the highest FR for each pretrained model is shown in bold-face.

| Model | Baseline | $L_{mask}$ | $\Delta = 1.0$ | | | $\Delta = 1.2$ | | | $\Delta = 1.5$ | | |
|---|---|---|---|---|---|---|---|---|---|---|---|
| | | | 2% | 5% | 10% | 2% | 5% | 10% | 2% | 5% | 10% |
| MLP-Mixer | 0.224 | 2% | **0.065** 72.3% | | | **0.072** 77.3% | | | 0.144 89.1% | | |
| | | 5% | 0.067 75.2% | **0.068** 73.3% | | 0.079 **80.2%** | 0.075 78.2% | | **0.141** 90.1% | 0.143 89.1% | |
| | | 10% | 0.068 **77.2%** | 0.069 76.2% | **0.066** 69.3% | 0.079 80.2% | 0.079 **80.2%** | 0.075 76.2% | 0.143 **91.1%** | **0.141** 90.1% | 0.139 88.1% |
| GRU | 0.243 | 2% | **0.067** 66.3% | | | **0.070** 73.3% | | | **0.143** 89.1% | | |
| | | 5% | 0.072 72.3% | **0.070** 68.3% | | 0.075 76.2% | **0.073** 74.3% | | 0.145 **91.1%** | 0.141 88.1% | |
| | | 10% | 0.070 **74.3%** | 0.071 72.3% | **0.069** 63.4% | 0.074 77.2% | 0.074 77.2% | **0.066** 70.3% | 0.145 91.1% | 0.143 90.1% | 0.137 86.1% |
| LSTM | 0.229 | 2% | **0.068** 66.3% | | | **0.071** 76.2% | | | 0.152 **91.1%** | | |
| | | 5% | 0.069 66.3% | **0.070** 65.3% | | 0.073 **77.2%** | 0.071 76.2% | | **0.149** 89.1% | 0.147 88.1% | |
| | | 10% | 0.070 65.3% | 0.071 **66.3%** | **0.066** 61.4% | 0.073 77.2% | 0.073 77.2% | **0.064** 72.3% | 0.150 89.1% | 0.153 **90.1%** | 0.149 87.1% |
| MLP | 0.222 | 2% | **0.064** 67.3% | | | **0.064** 72.3% | | | 0.148 90.1% | | |
| | | 5% | 0.067 **71.3%** | **0.064** 68.3% | | **0.064** 72.3% | **0.064** 72.3% | | 0.148 **92.1%** | 0.148 90.1% | |
| | | 10% | 0.067 70.3% | 0.067 70.3% | **0.063** 66.3% | **0.064** 72.3% | **0.064** 72.3% | 0.063 70.3% | **0.146** 91.1% | **0.146** 91.1% | 0.144 89.1% |
| ResNet18 | 0.248 | 2% | **0.074** 64.4% | | | 0.087 75.2% | | | **0.149** 88.1% | | |
| | | 5% | 0.077 **68.3%** | **0.077** 65.3% | | 0.088 **79.2%** | 0.089 76.2% | | 0.150 **89.1%** | 0.146 87.1% | |
| | | 10% | 0.077 67.3% | 0.078 **67.3%** | **0.076** 60.4% | 0.086 76.2% | **0.086** 76.2% | 0.083 73.3% | 0.150 88.1% | 0.149 86.1% | **0.145** 83.2% |

**MLP-Mixer.** An interesting observation is that the certified accuracy of DS and RA keeps constant to different $L_{adv}$. The reason is that, the probability score of DS/RA models often concentrates on a single class, causing most classifications (including both (correct and wrong classifications) of DS and RA are of high robustness. The results show that the certified accuracy of MIA is more than twice of DR and RA across different $L_{adv}$. The reason is that the pretrained model of MIA classifies the masked series, while the base model in RS/DS classifies the subseries. MIA can attain a higher certified accuracy because the masked series contains much more information ($t_0 - L_{mask}$ unmasked time points) than the subseries ($\eta$ sampled time points).

**Comparison on TSF.** Table 2 reports FR and MSE of three defenses on Exchange, where the model predicts the next 30 values. Here we utilize discretization technique to make the TSF task feasible to DS and RA. The table shows that MIA offers a significantly higher FR than DS and RA, implying that MIA return the forecasting results much more frequently than other two defenses. The reason for the superior FR is same as TSC. We also observe that DS ($\eta = 15\%$) achieves a lower MSE than MIA at $L_{adv} \geq 4\%$, which partially owing to its low FR. Since a low FR implies that the aggregation step of MIA filters a large portion of distrustful forecasts, reducing the difficulty of achieving lower MSE for the remained forecasts. Based on the results, MIA is better than other two defenses when we jointly consider FR and MSE.

Table 5: (Electricity) ($c_1$ $c_2\%$) report (MSE FR%) of MIA on different pretrained models.

| Model | Baseline | $L_{mask}$ | $\Delta = 1.0$ 2% | 5% | 10% | $\Delta = 1.2$ 2% | 5% | 10% | $\Delta = 1.5$ 2% | 5% | 10% |
|---|---|---|---|---|---|---|---|---|---|---|---|
| MLP-Mixer | 0.388 | 2% | **0.093** 66.3% | | | 0.094 64.4% | | | 0.136 77.2% | | |
| | | 5% | 0.095 **69.3%** | **0.093** **68.3%** | | 0.096 **67.3%** | 0.094 **66.3%** | | 0.134 **78.2%** | 0.134 **78.2%** | |
| | | 10% | 0.095 67.3% | 0.095 67.3% | **0.089** 55.4% | 0.091 64.4% | **0.091** 64.4% | **0.089** 59.4% | **0.134** 77.2% | **0.134** 77.2% | **0.116** 66.3% |
| GRU | 0.420 | 2% | **0.099** 62.4% | | | 0.086 60.4% | | | **0.135** 68.3% | | |
| | | 5% | 0.100 64.4% | 0.101 **63.4%** | | 0.087 **62.4%** | **0.087** 62.4% | | 0.139 **69.3%** | 0.139 **69.3%** | |
| | | 10% | 0.102 **65.3%** | **0.099** 62.4% | 0.103 49.5% | 0.085 61.4% | 0.087 61.4% | **0.086** 53.5% | 0.139 **69.3%** | 0.139 **69.3%** | **0.120** 64.4% |
| LSTM | 0.438 | 2% | 0.100 64.4% | | | **0.082** 52.5% | | | 0.142 70.3% | | |
| | | 5% | **0.100** 66.3% | 0.101 65.3% | | 0.092 55.4% | **0.092** 55.4% | | **0.140** 71.3% | 0.142 70.3% | |
| | | 10% | **0.100** 66.3% | **0.100** 64.4% | 0.110 51.5% | 0.091 56.3% | 0.092 55.4% | 0.094 48.5% | **0.140** 71.3% | 0.142 70.3% | **0.119** 59.4% |
| MLP | 0.402 | 2% | 0.106 73.3% | | | **0.082** 61.4% | | | 0.136 74.3% | | |
| | | 5% | 0.108 **77.2%** | 0.105 **75.2%** | | 0.087 65.3% | 0.085 63.4% | | **0.131** 77.2% | 0.132 76.2% | |
| | | 10% | **0.105** 75.2% | 0.105 75.2% | **0.098** 60.4% | 0.087 61.4% | 0.087 61.4% | **0.084** 55.4% | **0.131** 77.2% | **0.126** 76.2% | **0.108** 66.3% |
| ResNet18 | 0.554 | 2% | 0.093 51.5% | | | 0.080 59.4% | | | **0.136** 66.3% | | |
| | | 5% | **0.089** 51.5% | **0.089** 50.5% | | 0.080 **60.4%** | **0.080** 60.4% | | 0.140 **68.3%** | 0.140 **68.3%** | |
| | | 10% | 0.094 **52.5%** | 0.093 51.5% | **0.092** 43.6% | 0.079 56.4% | 0.081 58.4% | **0.076** 54.5% | 0.141 67.3% | **0.136** 66.3% | **0.121** 61.4% |

Table 6: Comparison of different training algorithms on 3 TSC datasets.

| Model | Training | $L_{mask}$ | DistalPhalanxTW 5% | 10% | 15% | MiddlePhalanxTW 5% | 10% | 15% | ProximalPhalanxTW 5% | 10% | 15% |
|---|---|---|---|---|---|---|---|---|---|---|---|
| MLP-Mixer | Random | 5% | 62.4% | | | 52.5% | | | 65.3% | | |
| | | 10% | 61.4% | 58.4% | | 53.5% | 50.5% | | 73.3% | 71.3% | |
| | | 15% | 59.4% | 59.4% | 58.4% | 59.4% | 54.5% | 49.5% | 68.3% | 67.3% | 61.4% |
| | Masked | 5% | **65.3%** | | | **64.4%** | | | **72.3%** | | |
| | | 10% | **66.3%** | **63.4%** | | **67.3%** | **60.4%** | | **76.2%** | **74.3%** | |
| | | 15% | **64.4%** | **64.4%** | **60.4%** | **66.3%** | **62.4%** | **59.4%** | **76.2%** | **75.2%** | **74.3%** |
| FCN | Random | 5% | 63.4% | | | 53.5% | | | 68.3% | | |
| | | 10% | 66.3% | 63.4% | | 52.5% | 52.5% | | 70.3% | 67.3% | |
| | | 15% | 66.3% | 65.3% | 65.3% | 57.4% | 57.4% | 56.4% | 66.3% | 66.3% | 65.3% |
| | Masked | 5% | **70.3%** | | | **64.4%** | | | **75.2%** | | |
| | | 10% | **70.3%** | **69.3%** | | **66.3%** | **65.3%** | | **73.3%** | **71.3%** | |
| | | 15% | **70.3%** | **67.3%** | **66.3%** | **65.3%** | **63.4%** | **63.4%** | **75.2%** | **73.3%** | **73.3%** |
| MLP | Random | 5% | 62.4% | | | 65.3% | | | 72.3% | | |
| | | 10% | 62.4% | 60.4% | | 59.4% | 54.5% | | 76.2% | 73.3% | |
| | | 15% | 63.4% | 61.4% | 61.4% | 61.4% | 55.4% | 55.4% | 71.3% | 71.3% | 68.3% |
| | Masked | 5% | **64.4%** | | | **69.3%** | | | **79.2%** | | |
| | | 10% | **65.3%** | **64.4%** | | **70.3%** | **67.3%** | | **79.2%** | **78.2%** | |
| | | 15% | **66.3%** | **63.4%** | **63.4%** | **69.3%** | **66.3%** | **66.3%** | **78.2%** | **78.2%** | **77.2%** |
| ResNet-18 | Random | 5% | 61.4% | | | 57.4% | | | 65.3% | | |
| | | 10% | 59.4% | 57.4% | | 57.4% | 56.4% | | 74.3% | 67.3% | |
| | | 15% | 58.4% | 56.4% | 55.4% | 58.4% | 58.4% | 58.4% | 74.3% | 73.3% | 66.3% |
| | Masked | 5% | **65.3%** | | | **67.3%** | | | **78.2%** | | |
| | | 10% | **66.3%** | **64.4%** | | 63.4% | **63.4%** | | **78.2%** | **76.2%** | |
| | | 15% | **64.4%** | **61.4%** | **59.4%** | **65.3%** | **62.4%** | **62.4%** | **78.2%** | **78.2%** | **78.2%** |

**Comparison on inference time.** Table 3 compares the inference time of three defenses, which is averaged among three TSC datasets. We observe that the inference time of MIA is larger than DS, but significantly smaller than RA. Specifically, MIA's larger inference time than DS is owing to the cost in running the imputation model. RA's large inference time is for the confidence interval estimation[6]. We also observe that the inference time of MIA increases with $L_{adv}$, and decreases with $L_{mask}$, because the number of masked series is $\lceil (t_0 - L_{mask})/(L_{mask} - L_{adv} + 1) \rceil + 1$. We omit the inference time analysis on TSF datasets to Appendix.

### 5.2 Analysis of MIA on Different Pretrained Models

Table 4 and Table 5 report the performance of MIA on different pretrained models, where the model forecasts the next 24 points. We observe that MIA ($\Delta = 1.0, 1.5$) consistently lowers MSE as compared to that of the original pretrained models, suggesting MIA could also be an effective plugin for performance improvement. Specifically, MIA improves the forecasting performance in the way of filtering these distrustful forecasts, sacrificing the availability (the decrease of FR) for lower MSE as well as certified robustness, which is a common trade-off in the field of certified defenses (Cohen et al., 2019; Levine & Feizi, 2020a;b; Liu et al., 2021; Han et al., 2021). We can control the trade-off between MSE and FR by $\Delta$, as the decrease of $\Delta$ can reduce MSE and FR.

### 5.3 Analysis on Imputation Model of MIA and Ablation Study

**Impact of training algorithm.** Table 6 compares our masked training to random training, which trains the imputation model on the randomly masked series. Through extensive comparisons on different imputation model architectures and datasets, we convince that masked training consistently

---

[6]Following the official implementation of RA (Levine & Feizi, 2020b), we take $100,000$ samples for the confidence interval estimation.

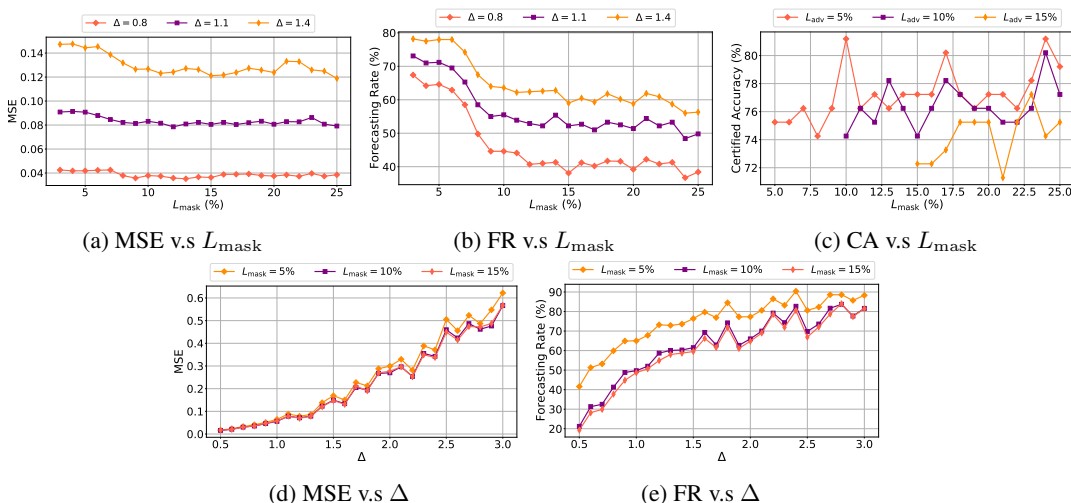

Figure 2: **Top**: impact of $L_{\text{mask}}$ on TSC dataset ProximalPhalanxTW. **Bottom**: impact of $\Delta$ on TSF dataset Traffic ($L_{\text{adv}} = 3\%$).

Table 7: Comparison of four imputation models. The best results are shown in bold-face.

| Model | $L_{\text{mask}}$ | Electricity | | | Exchange | | | Traffic | | |
|---|---|---|---|---|---|---|---|---|---|---|
| | | 2% | 5% | 10% | 2% | 5% | 10% | 2% | 5% | 10% |
| MLP-Mixer | 2% | 0.136 **77.2%** | | | 0.158 **87.1%** | | | 0.144 **89.1%** | | |
| | 5% | 0.134 **78.2%** | 0.134 **78.2%** | | 0.144 **83.2%** | 0.138 **81.2%** | | 0.141 **90.1%** | 0.143 **89.1%** | |
| | 10% | 0.134 **77.2%** | 0.134 **77.2%** | 0.116 **66.3%** | 0.141 **82.2%** | 0.141 **82.2%** | 0.134 **79.2%** | 0.143 **91.1%** | 0.141 **90.1%** | 0.139 **88.1%** |
| BRITS | 2% | 0.118 65.3% | | | 0.118 40.6% | | | 0.142 87.1% | | |
| | 5% | 0.110 71.3% | 0.099 59.4% | | 0.118 49.5% | 0.098 15.8% | | 0.136 84.2% | 0.130 75.2% | |
| | 10% | 0.099 61.4% | 0.100 60.4% | 0.094 45.5% | 0.207 10.9% | 0.277 1.0% | 0.000 0.0% | 0.128 79.2% | 0.129 77.2% | 0.123 73.3% |
| SAITS | 2% | 0.101 58.4% | | | **0.095** 25.7% | | | **0.120** 67.3% | | |
| | 5% | **0.091** 53.5% | **0.090** 52.5% | | 0.137 43.6% | 0.101 27.7% | | 0.128 73.3% | **0.121** 68.3% | |
| | 10% | 0.104 56.4% | 0.103 54.5% | 0.094 44.6% | 0.070 7.9% | 0.048 6.9% | **0.001** 4.0% | 0.125 67.3% | 0.132 69.3% | 0.124 59.4% |
| Transformer | 2% | **0.085** 52.5% | | | 0.145 23.8% | | | 0.130 68.3% | | |
| | 5% | 0.110 52.5% | 0.113 42.6% | | **0.072** 6.9% | **0.001** 2.0% | | **0.127** 74.3% | 0.123 65.3% | |
| | 10% | **0.096** 60.4% | **0.093** 58.4% | **0.093** 34.7% | **0.029** 8.9% | **0.036** 7.9% | 0.001 5.0% | **0.124** 75.2% | **0.127** 74.3% | **0.123** 61.4% |

outperforms random training for MIA by a non-trivial gap. The gap becomes even larger at larger $L_{\text{adv}}$. The results suggest that masked training is suitable for MIA imputation model.

**Impact of architecture of imputation models.** Table 7 reports the performances of different imputation model architectures on MIA. Our results show that MLP-Mixer can offer higher FR and lower MSE simultaneously, suggesting MLP-Mixer is intrinsically more robust than other models.

**Impact of $L_{\text{mask}}$.** Fig. 2a, 2b show: 1) MSE roughly keeps constant w.r.t. $L_{\text{mask}}$, because the discretization technique can diminish the difference between the forecasts that are close to each other. 2) FR decreases with the increase of $L_{\text{mask}}$, because our imputation quality decreases with $L_{\text{mask}}$, making it harder to reach agreement unanimously. Fig. 2c show that the impact of $L_{\text{mask}}$ on CA is not significant. Although the increase of $L_{\text{mask}}$ reduces our imputation quality, it reduces the number of masked series simultaneously.

**Impact of $\Delta$.** Fig. 2d, 2e report the impact of $\Delta$. As $\Delta$ increases, we observe that MSE and FR increase, validating our statement about $\Delta$ in Section 4.1.

## 6 CONCLUSION

In this paper, we propose the first framework for time-series models to certifiably defend against $\ell_0$-norm perturbations. Notably, MIA is a plug-and-play defense, which can be easily applied to any TSF/TSC pretrained model. The only requirement of deploying MIA is to train an imputation model, which has been extensively explored in this work. Moreover, our extensive experiments validate the effectiveness of MIA. We expect our work can inspire more studies on the $\ell_0$-norm robustness for time-series models. Interesting future works include applying MIA to probabilistic models.

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

## A  Proof for Proposition 1

**Proposition 2** (Robustness Certificate of MIA). The forecast/label (not **Abstain**) returned by Algorithm 1 cannot be changed by any temporally-localized perturbation whose $\ell_0$ norm is no larger than $L_{\mathrm{adv}}$ (see proof in Appendix).

**Proof.** Here we prove the robustness certificate for MIA (TSC). The proof for MIA (TSF) is analogous to this proof. Assume that the adversary has changed the classification result of MIA from $y_1$ to $y_2$ via the temporally-localized perturbation $\delta$ ($\ell_0$ norm is $L_{\mathrm{adv}}$). For notational simplicity we denote $M_{[1+k\alpha:\min(L_{mask}+k\alpha,t_0)]}$ by $M^{(k)}, k = 0, \ldots, \lceil(t_0 - L_{\mathrm{mask}})/\alpha\rceil$ and denote $\boldsymbol{\delta}_{[t_{\mathrm{adv}}+1:t_{\mathrm{adv}}+L_{\mathrm{adv}}]}$ by $\boldsymbol{\delta}^{(t_{\mathrm{adv}})}, t_{\mathrm{adv}} = 0, \ldots, t_0 - L_{\mathrm{adv}}$. Then, we have:

$$f(\mathbf{x}_{1:t_0} \odot M^{(0)}) = f(\mathbf{x}_{1:t_0} \odot M^{(1)}) = \ldots = y_1 \tag{12}$$

$$f((\mathbf{x}_{1:t_0} + \boldsymbol{\delta}^{(t_{\mathrm{adv}})}) \odot M^{(0)}) = f((\mathbf{x}_{1:t_0} + \boldsymbol{\delta}^{(t_{\mathrm{adv}})}) \odot M^{(1)}) = \ldots = y_2 \tag{13}$$

Then our next step is to prove that *there exists a mask $M^{(\hat{m})}, \hat{m} \in \{0, \ldots, \lceil t_0 - L_{\mathrm{mask}}\rceil/\alpha\rceil\}$ that can occlude the perturbation*. Specifically, we show that the mask $M^{(\lfloor \frac{t_{\mathrm{adv}}}{\alpha}\rfloor)}$ can cover the perturbation. First, we show the presence of the mask $M^{(\lfloor \frac{t_{\mathrm{adv}}}{\alpha}\rfloor)}$ by proving $\lfloor\frac{t_{\mathrm{adv}}}{\alpha}\rfloor \leq \lceil(t_0 - L_{\mathrm{mask}})/\alpha\rceil$, as follow:

$$\frac{t_{\mathrm{adv}}}{\alpha} - \frac{t_0 - L_{\mathrm{mask}}}{\alpha} \leq \frac{t_0 - L_{\mathrm{adv}} - t_0 + L_{\mathrm{mask}}}{L_{\mathrm{mask}} - L_{\mathrm{adv}} + 1} = \frac{L_{\mathrm{mask}} - L_{\mathrm{adv}}}{L_{\mathrm{mask}} - L_{\mathrm{adv}} + 1} < 1 \tag{14}$$

Second, we show that $M^{(\lfloor \frac{t_{\mathrm{adv}}}{\alpha}\rfloor)}$ covers the perturbation by comparing the starting/end point of the mask $M^{(\frac{t_{\mathrm{adv}}}{\alpha})}$ and the perturbation $\delta$. For the starting point, we have:

$$\underbrace{(\alpha\lfloor\frac{t_{\mathrm{adv}}}{\alpha}\rfloor + 1)}_{\text{Mask}} - \underbrace{(t_{\mathrm{adv}} + 1)}_{\text{Perturbation}} \leq 0 \tag{15}$$

In terms of the end points, we have:

$$\underbrace{(\alpha\lfloor\frac{t_{\mathrm{adv}}}{\alpha}\rfloor + 1 + L_{\mathrm{mask}})}_{\text{Mask}} - \underbrace{(t_{\mathrm{adv}} + L_{\mathrm{adv}})}_{\text{Perturbation}} \tag{16}$$

$$= \alpha\lfloor\frac{t_{\mathrm{adv}}}{\alpha}\rfloor + (L_{\mathrm{mask}} - L_{\mathrm{adv}} + 1) - t_{\mathrm{adv}} \tag{17}$$

$$= \alpha(\lfloor\frac{t_{\mathrm{adv}}}{\alpha}\rfloor + 1) - t_{\mathrm{adv}} \geq 0 \tag{18}$$

As $M^{(\hat{m})}$ occludes the perturbation, thus $(\mathbf{x}_{1:t_0}) \odot M^{(\hat{m})} = (\mathbf{x}_{1:t_0} + \boldsymbol{\delta}^{(t_{\mathrm{adv}})}) \odot M^{(\hat{m})} \Rightarrow y_1 = y_2$. Our proof is completed.

## B  Empirical Evaluation on Risk of Temporally-Localized Perturbations

To support our statement about the risk of temporally-localized perturbations, we specifically propose an algorithm for generating the temporally-localized perturbations. We then evaluate the attack performance visually.

### B.1  Restate Definition of temporally-localized Perturbation

**Definition 2** (Temporally-localized perturbation). Temporally-localized perturbation is to *perturb consecutive time points of* $\mathbf{x}_{1:t_0}$ *w.r.t.* $\ell_0$*-norm constraint*. The perturbed series is:

$$\begin{aligned}
&\mathbf{x}_{1:t_0} + \boldsymbol{\delta}_{[t_{\mathrm{adv}}+1:t_{\mathrm{adv}}+L_{\mathrm{adv}}]} \quad \text{subject to} \quad \|\boldsymbol{\delta}_{[t_{\mathrm{adv}}+1:t_{\mathrm{adv}}+L_{\mathrm{adv}}]}\|_0 \leq L_{\mathrm{adv}} \\
=&\mathbf{x}_{1:t_0} + [0, \ldots, \delta_{t_{\mathrm{adv}}+1}, \ldots, \delta_{t_{\mathrm{adv}}+L_{\mathrm{adv}}}, 0, \ldots, 0] \\
=&[x_1, \ldots, \underbrace{x_{t_{\mathrm{adv}}+1} + \delta_{t_{\mathrm{adv}}+1}, \ldots, x_{t_{\mathrm{adv}}+L_{\mathrm{adv}}} + \delta_{t_{\mathrm{adv}}+L_{\mathrm{adv}}}}_{\text{Perturbed subsequence}}, \ldots, x_{t_0}]
\end{aligned} \tag{19}$$

where $t_{\mathrm{adv}}+1$ and $L_{\mathrm{adv}}$ are the starting point and the $\ell_0$-norm of the temporally-localized perturbation $\boldsymbol{\delta}_{[t_{\mathrm{adv}}+1:t_{\mathrm{adv}}+L_{\mathrm{adv}}]}$.

Table 8: (Traffic) Evaluate MSE between the clean forecasts and the perturbed forecasts. The temporally-localized perturbations is generated subject to different $\ell_0$-norm constraints ($L_{\mathrm{adv}} = 2\%, 5\%, 10\%$) and $\ell_2$-norm constraints ($1.0, 1.5, 2.0, 2.5, 3.0, 3.5$).

| Model | $L_{\mathrm{adv}}$ | 1.0 | 1.5 | 2.0 | 2.5 | 3.0 | 3.5 |
|---|---|---|---|---|---|---|---|
| | 2% | 1.614 | 1.675 | 1.731 | 1.547 | 1.482 | 1.469 |
| MLP | 5% | 1.437 | 1.375 | 1.295 | 1.288 | 1.472 | 1.590 |
| | 10% | 1.318 | 1.204 | 1.137 | 1.418 | 1.686 | 1.865 |
| | 2% | 0.156 | 0.156 | 0.188 | 0.283 | 0.400 | 0.425 |
| MLP-Mixer | 5% | 0.154 | 0.502 | 0.322 | 0.148 | 0.148 | 0.148 |
| | 10% | 0.071 | 0.321 | 0.282 | 0.201 | 0.220 | 0.150 |
| | 2% | 0.059 | 0.132 | 0.152 | 0.422 | 0.212 | 0.176 |
| LSTM | 5% | 0.323 | 0.503 | 0.646 | 0.630 | 0.852 | 0.929 |
| | 10% | 0.166 | 0.290 | 0.652 | 0.803 | 1.301 | 1.753 |

## B.2 ALGORITHM OF GENERATING TEMPORALLY-LOCALIZED PERTURBATIONS.

The objective of our algorithm is to maximize MSE between the original forecasts and the perturbed forecasts, with respect to the $\ell_0$-norm constraint. Specifically, given the forecasting model $f(\mathbf{x}_{1:t_0}) \to \mathbf{x}_{t_0+1,t_0+\tau}$, our objective can be formulated as follows:

$$\arg \max_{\boldsymbol{\delta}} |f(\mathbf{x}_{1:t_0} + \boldsymbol{\delta}) - f(\mathbf{x}_{1:t_0})|_2^2 \tag{20}$$

where $\delta$ corresponds to the perturbation defined in Eq.(19). Actually, the problem of computing the temporally-localized perturbation can be decomposed into two sub-problems: **P1** ) Search for the period $[t_{\mathrm{adv}}+1, t_{\mathrm{adv}}+L_{\mathrm{adv}}]$ to perturb. **P2** ) Fix the period $[t_{\mathrm{adv}}+1, t_{\mathrm{adv}}+L_{\mathrm{adv}}]$, compute the value of the perturbation $\delta_{t_{\mathrm{adv}}+1}, \ldots, \delta_{t_{\mathrm{adv}}+L_{\mathrm{adv}}}$. Here solving **P2** is not hard. If we have determined $\delta_{t_{\mathrm{adv}}+1}, \ldots, \delta_{t_{\mathrm{adv}}+L_{\mathrm{adv}}}$, we can maximize the following loss to compute the perturbation values via projected gradient descent (PGD).

$$\max_{\delta_{t_{\mathrm{adv}}+1}, \ldots, \delta_{t_{\mathrm{adv}}+L_{\mathrm{adv}}}} |f(\mathbf{x}_{1:t_0} + \boldsymbol{\delta}) - f(\mathbf{x}_{1:t_0})|_2^2 \tag{21}$$

Then the main challenge is to determined which period to perturb. Here we solve **P1** by enumerating all the possible perturbing positions $[t_{\mathrm{adv}} + 1 : t_{\mathrm{adv}} + L_{\mathrm{adv}}]$, $t_{\mathrm{adv}} = 0, \ldots, t_0 - L_{\mathrm{adv}}$ and compute the corresponding attacks. Finally, we return the one with the largest MSE loss among $t_0 - L_{\mathrm{adv}} + 1$ perturbations. However, in practice we found that computing the values of perturbation (**P2** ) w.r.t. to the fixed period is hard to converge, as the $\ell_2$ norm of the temporally-localized perturbation will approach $\infty$. We believe that a perturbation attack with $\infty$ $\ell_2$ norm is meaningless in practice. In the sake of practicality, we additionally consider $\ell_2$ norm for the temporally-localized perturbations besides $\ell_0$-norm constraint for the sub-problem **P2** , as follows:

$$\max_{\delta_{t_{\mathrm{adv}}+1}, \ldots, \delta_{t_{\mathrm{adv}}+L_{\mathrm{adv}}}} |f(\mathbf{x}_{1:t_0} + \boldsymbol{\delta}) - f(\mathbf{x}_{1:t_0})|_2^2 \text{subject to} \|\boldsymbol{\delta}\|_2^2 \leq \epsilon \tag{22}$$

where $\epsilon$ is the preset upper bound of the perturbation $\ell_2$ norm.

## B.3 EMPIRICAL EVALUATION OF TEMPORALLY-LOCALIZED PERTURBATIONS.

## B.4 QUANTIFY THE RISK OF TEMPORALLY-LOCALIZED PERTURBATIONS.

Table 8 quantifies the risk of temporally-localized perturbations via computing MSE between the clean forecasts and the perturbed forecasts w.r.t. the $\ell_0$-norm ($L_{\mathrm{adv}}$) and the $\ell_2$-norm ($\epsilon$) constraints. We observe that MLP-Mixer model provides the highest empirical robustness among three models, which partially explains why MLP-Mixer outperforms other models on MIA. In particular, we further compare MLP to MLP+MIA ($\delta = 1.5$) in Table 4 of the main paper. Specifically, MSE of MLP under temporally-localized perturbations ($\epsilon = 3.0$) is $5\% : 1.472, 10\% : 0.1.686$ while MLP+MIA is $5\% : 0.146, 10\% : 0.144$. MIA reduces the MSE to roughly one tenth of the original, which indicates that MIA can effectively prevent our forecasting results from being influenced by the temporally-localized perturbations.

Table 9: Dataset information for TSF and TSC.

| Dataset | Context length | Forecasting length | Number of classes |
|---|---|---|---|
| Electricity | 96 | 24 | N/A |
| Exchange | 120 | 30 | N/A |
| Traffic | 96 | 24 | N/A |
| DistalPhalanxTW | 80 | N/A | 6 |
| MiddlePhalanxTW | 80 | N/A | 6 |
| ProximalPhalanxTW | 80 | N/A | 6 |

### B.5 COMPARE ATTACKING PERFORMANCE OF $\ell_0$ ATTACK TO $\ell_2$ ATTACK

We compare the $\ell_0$ attack and $\ell_2$ attack under under norm constraints $\beta$ (attack rate) on time series forecasting task. Results are shown in Table 30, 31 and 32. The values in tables are calculated as $\frac{\text{MSE}_{\ell_0} - \text{MSE}_{\ell_2}}{\text{MSE}_{\ell_2}} \times 100\%$.

### B.6 VISUALIZE THE RISK OF TEMPORALLY-LOCALIZED PERTURBATIONS.

Fig. 3 illustratively shows the effect of temporally-localized perturbations on our forecasting results. We observe that temporally-localized perturbations of $L_{\text{atk}} = 10\%$ can significantly change our forecasts.

## C EXPERIMENTAL SETUPS

### C.1 DATASET INFORMATION

Table 9 shows the details of each dataset, including context length, forecasting length (for TSF datasets) and number of classes.

**Traffic**   Hourly occupancy rate, between 0 and 1, of 963 San Francisco car lanes (Salinas et al., 2019).

**Electricity**   Hourly time series of the electricity consumption of 370 customers (Salinas et al., 2019).

**Exchange**   Daily exchange rate between 8 currencies (Salinas et al., 2019).

**DistalPhalanxTW, MiddlePhalanxTW, ProximalPhalanxTw**   [7] This series of 11 classification problems were created as part of Luke Davis's PhD titled "Predictive Modelling of Bone Ageing". They are designed to test the efficacy of hand and bone outline detection and whether these outlines could be helpful in bone age prediction. Note that these problems are aligned by subject, and hence can be treated as a multi-dimensional TSC problem. The final three bone classification problems, DistalPhalanxTW, MiddlePhalanxTW and ProximalPhalanxTW, involve predicting the Tanner-Whitehouse score (as labelled by a human expert) from the outline.

**Data Pre-Processing**   We pre-process the input series with *scipy.signal.savgol_filter* with window length 15 and polyorder 5 on both training and testing datasets. Besides, we normalize each input series with its mean value and standard deviation. Mean value and standard deviation will be 0 and 1 respectively for each normalized input series. We use the instance normalization method on both trainsets and testsets.

---

[7]https://timeseriesclassification.com/description.php

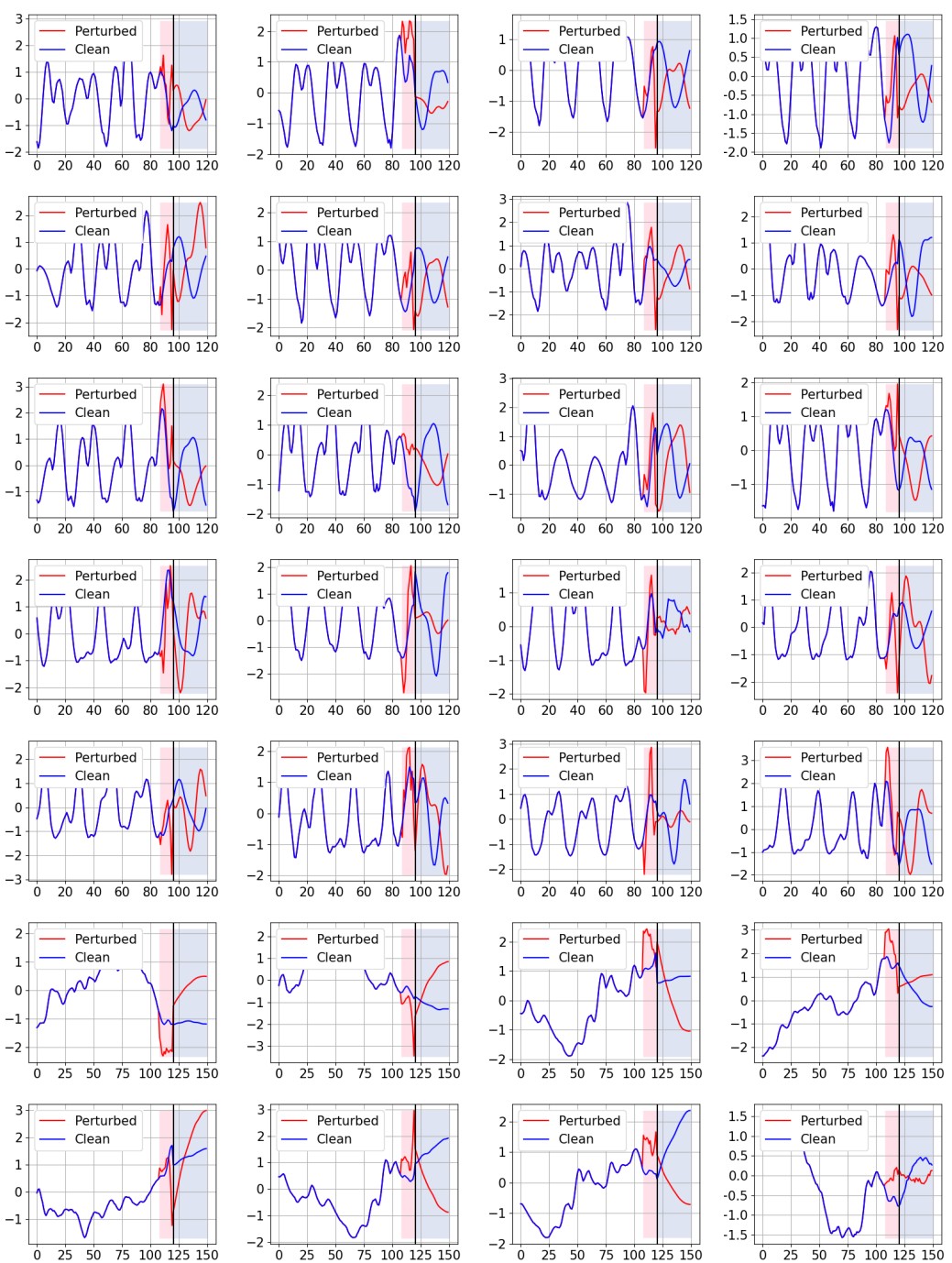

Figure 3: The effect of temporally-localized perturbations ($L_{atk} = 10\%$ and $\epsilon = 3.0$) on different datasets. Clean and Perturbed refer to the normal input series and the temporally-localized perturbations respectively. **Row 1, 2, 3**: Traffic. **Row 4, 5**: Electricity. **Row 6, 7**: Exchange rate. The red background denotes the position of the location of the perturbation. The blue background denotes the output series.

Table 10: MLP structure

| Input dim | Output dim | Activation |
|---|---|---|
| Length of sequence | 96 | LeakyReLU($\alpha = 0.2$) |
| 96 | 96 | LeakyReLU($\alpha = 0.2$) |
| 96 | 96 | LeakyReLU($\alpha = 0.2$) |
| 96 | 96 | LeakyReLU($\alpha = 0.2$) |
| 96 | Length of sequence | LeakyReLU($\alpha = 0.2$) |

Table 11: LSTM/GRU structure

| Number of layers | Hidden dim |
|---|---|
| 4 | 32 |

Table 12: The structure of MLP-Mixer Block, and we stack 4 MLP-Mixer blocks to construct MLP-Mixer for forecasting and imputation.

| Type | Input dim | Output dim | Activation |
|---|---|---|---|
| LayerNorm | 96 | 96 | / |
| Linear | 128 | 128 | GELU Hendrycks & Gimpel (2016) |
| Linear | 128 | 128 | GELU |
| LayerNorm | 96 | 96 | / |
| Linear | Length of sequence | Length of sequence | GELU |
| Linear | Length of sequence | Length of sequence | GELU |

Table 13: The structure of Fully Convolutional Network (FCN) for TSC.

| Layer | Input channel | Output channel | Kernel size |
|---|---|---|---|
| Conv1d | 1 | 96 | 3 |
| BatchNorm1d | 96 | 96 | N/A |
| ReLU | 96 | 96 | N/A |
| Conv1d | 96 | 96 | 3 |
| BatchNorm1d | 96 | 96 | N/A |
| ReLU | 96 | 96 | N/A |
| Conv1d | 96 | 96 | 3 |
| BatchNorm1d | 96 | 96 | N/A |
| ReLU | 96 | 96 | N/A |
| Conv1d | 96 | 96 | 3 |
| BatchNorm1d | 96 | 96 | N/A |
| ReLU | 96 | 96 | N/A |
| GlobalPooling | 96 | 96 | N/A |
| Linear | 96 | 6 | N/A |

## D    INTRODUCTION TO PRETRAINED MODELS

**Model architecture.** We use the classical forecasting and classification models, MLP, MLP-Mixer (Tolstikhin et al., 2021), GRU (Cho et al., 2014), LSTM (Hochreiter & Schmidhuber, 1997), FCN (Ismail Fawaz et al., 2019b) and ResNet-18 (He et al., 2016) as the pretrained models. We show the architecture of these models in Table 10, 11, 12, 13.

**Training.** we uniformly adapt Adam optimizer (Kingma & Ba, 2014) with $lr = 0.0001, \beta = (0.9, 0.999), \epsilon = 10^{-8}$, weight_decay $= 0$, epochs $= 20$ for all the pretrained models.

## E    MORE EXPERIMENTAL RESULTS

**Comparison to peer methods on Mean Absolute Error (MAE).** Table 15 compares MAE and FR of three defenses on Traffic. Analogous to the comparison (Table 2 in main paper) on MSE, MIA achieves both the lowest MAE and highest FR on $L_{\text{adv}} = 1\%, 2\%, 3\%$. DS outperforms MIA at $L_{\text{adv}} = 4\%, \dots, 10\%$ for the great sacrifice on its FR.

Table 14: Comparison of imputation quality (MSE between the imputed series and the original series) of different imputation models on imputing the masked series of different mask length $L_{\text{mask}} = 5\%, 10\%, 15\%, 20\%$. Fixing $L_{\text{mask}}$, we first construct $t_0 - L_{\text{mask}} + 1$ masked series of $L_{\text{mask}}$ and compute the average imputation MSE over imputing these $t_0 - L_{\text{mask}} + 1$ masked series. Bold indicates the best among four generators.

| Generator | Traffic | | | | Electricity | | | | Exchange | | | |
|---|---|---|---|---|---|---|---|---|---|---|---|---|
| | 5% | 10% | 15% | 20% | 5% | 10% | 15% | 20% | 5% | 10% | 15% | 20% |
| MLP-Mixer | **0.0007** | **0.0082** | **0.0170** | **0.0240** | **0.0033** | **0.0320** | **0.0458** | **0.0631** | **0.0002** | **0.0028** | **0.0104** | **0.0212** |
| SAITS | 0.0652 | 0.1110 | 0.1786 | 0.2476 | 0.0878 | 0.1654 | 0.2687 | 0.3586 | 0.0191 | 0.0410 | 0.0631 | 0.0824 |
| Transformer | 0.0642 | 0.1183 | 0.1854 | 0.2629 | 0.0821 | 0.1836 | 0.2404 | 0.3271 | 0.0210 | 0.0389 | 0.0595 | 0.0787 |
| BRITS | 0.0159 | 0.0419 | 0.0656 | 0.0927 | 0.0410 | 0.1192 | 0.1613 | 0.2109 | 0.0099 | 0.0252 | 0.0516 | 0.0705 |

Table 15: (Exchange) Comparison among three certified defenses on TSF dataset.

| Metric | Defense | 1 % | 2 % | 3 % | 4 % | 5 % | 6 % | 7 % | 8 % | 9 % | 10 % |
|---|---|---|---|---|---|---|---|---|---|---|---|
| FR (%) | MIA ($L_{\text{mask}} = 10\%$) | **82.2** | **82.2** | **83.2** | **82.2** | **82.2** | **81.2** | **81.2** | **81.2** | **80.2** | **79.2** |
| | DS ($\eta = 10\%$) | 24.8 | 24.8 | 24.8 | 24.8 | 24.8 | 24.8 | 24.8 | 24.8 | 24.8 | 24.8 |
| | RA ($\eta = 10\%$) | 16.8 | 16.8 | 16.8 | 16.8 | 16.8 | 16.8 | 16.8 | 16.8 | 16.8 | 16.8 |
| | MIA ($L_{\text{mask}} = 15\%$) | **64.4** | **71.3** | **69.3** | **69.3** | **71.3** | **68.3** | **69.3** | **71.3** | **62.4** | **65.3** |
| | DS ($\eta = 15\%$) | 20.8 | 20.8 | 20.8 | 14.9 | 14.9 | 11.9 | 11.9 | 11.9 | 11.9 | 10.9 |
| | RA ($\eta = 15\%$) | 23.8 | 23.8 | 23.8 | 23.8 | 23.8 | 23.8 | 23.8 | 23.8 | 23.8 | 23.8 |
| MAE | MIA ($L_{\text{mask}} = 10\%$) | **0.332** | **0.330** | **0.334** | **0.330** | **0.330** | **0.327** | **0.326** | **0.326** | **0.323** | **0.320** |
| | DS ($\eta = 10\%$) | 0.408 | 0.408 | 0.408 | 0.408 | 0.408 | 0.408 | 0.408 | 0.408 | 0.408 | 0.408 |
| | RA ($\eta = 10\%$) | 0.413 | 0.413 | 0.413 | 0.413 | 0.413 | 0.413 | 0.413 | 0.413 | 0.413 | 0.413 |
| | MIA ($L_{\text{mask}} = 15\%$) | **0.307** | **0.320** | **0.313** | 0.316 | 0.322 | 0.317 | 0.311 | 0.314 | 0.301 | 0.306 |
| | DS ($\eta = 15\%$) | 0.320 | 0.320 | 0.320 | **0.215** | **0.215** | **0.222** | **0.222** | **0.222** | **0.222** | **0.204** |
| | RA ($\eta = 15\%$) | 0.446 | 0.446 | 0.446 | 0.446 | 0.446 | 0.446 | 0.446 | 0.446 | 0.446 | 0.446 |

Table 16: (Traffic) The performance of MIA on different pretrained models. ($c_1$ $c_2$%) reports (MAE, FR%) of MIA. **Baseline** is MAE of the pretrained model without MIA. The lowest MAE and the highest FR for each pretrained model is shown in bold-face.

| Model | Baseline | $L_{\text{mask}}$ | $\Delta = 1.0$ | | | $\Delta = 1.2$ | | | $\Delta = 1.5$ | | |
|---|---|---|---|---|---|---|---|---|---|---|---|
| | | | 2% | 5% | 10% | 2% | 5% | 10% | 2% | 5% | 10% |
| MLP-Mixer | 0.265 | 2% | **0.227** 72.3% | | | **0.213** 77.2% | | | 0.340 89.1% | | |
| | | 5% | 0.231 75.2% | 0.233 73.3% | | 0.225 **80.2%** | 0.218 78.2% | | **0.336** 90.1% | 0.340 89.1% | |
| | | 10% | 0.233 **77.2%** | **0.233** 76.2% | **0.227** 69.3% | 0.225 **80.2%** | 0.225 **80.2%** | **0.216** 76.2% | 0.338 **91.1%** | **0.336** 90.1% | **0.333** 88.1% |
| GRU | 0.283 | 2% | **0.234** 66.3% | | | **0.210** 73.3% | | | 0.339 89.1% | | |
| | | 5% | 0.242 72.3% | 0.239 68.3% | | 0.219 76.2% | **0.214** 74.3% | | 0.340 **91.1%** | 0.337 88.1% | |
| | | 10% | 0.237 74.3% | **0.239** 72.3% | **0.236** 63.4% | 0.217 77.2% | 0.217 77.2% | **0.203** 70.3% | 0.340 **91.1%** | 0.338 90.1% | **0.329** 86.1% |
| LSTM | 0.274 | 2% | **0.235** 66.3% | | | **0.212** 76.2% | | | 0.350 **91.1%** | | |
| | | 5% | 0.237 66.3% | **0.237** 65.3% | | 0.215 77.2% | **0.213** 76.2% | | **0.345** 89.1% | 0.343 88.1% | |
| | | 10% | 0.237 65.3% | 0.239 66.3% | **0.231** 61.4% | 0.215 77.2% | 0.215 77.2% | **0.201** 72.3% | 0.347 89.1% | 0.349 90.1% | **0.344** 87.1% |
| MLP | 0.268 | 2% | **0.226** 67.3% | | | **0.201** 72.3% | | | 0.344 90.1% | | |
| | | 5% | 0.231 71.3% | **0.226** 68.3% | | 0.201 72.3% | 0.201 72.3% | | 0.343 **92.1%** | 0.344 90.1% | |
| | | 10% | 0.230 70.3% | 0.230 70.3% | **0.225** 66.3% | 0.201 72.3% | 0.201 72.3% | **0.199** 70.3% | **0.341** 91.1% | **0.341** 91.1% | **0.338** 89.1% |
| ResNet18 | 0.290 | 2% | **0.244** 64.4% | | | 0.238 75.2% | | | 0.343 88.1% | | |
| | | 5% | 0.249 68.3% | **0.249** 65.3% | | 0.239 **79.2%** | 0.241 76.2% | | **0.345** 89.1% | 0.340 87.1% | |
| | | 10% | 0.249 67.3% | 0.251 67.3% | **0.246** 60.4% | **0.235** 76.2% | **0.235** 76.2% | **0.231** 73.3% | 0.345 88.1% | 0.343 86.1% | **0.337** 83.2% |

Table 17: (Electricity) ($c_1$ $c_2$%) report (MAE FR%) of MIA on different pretrained models.

| Model | Baseline | $L_{\text{mask}}$ | $\Delta = 1.0$ | | | $\Delta = 1.2$ | | | $\Delta = 1.5$ | | |
|---|---|---|---|---|---|---|---|---|---|---|---|
| | | | 2% | 5% | 10% | 2% | 5% | 10% | 2% | 5% | 10% |
| MLP-Mixer | 0.430 | 2% | **0.260** 66.3% | | | 0.262 64.4% | | | 0.312 77.2% | | |
| | | 5% | 0.264 **69.3%** | 0.262 68.3% | | 0.266 67.3% | 0.263 **66.3%** | | 0.308 **78.2%** | 0.308 78.2% | |
| | | 10% | 0.264 67.3% | 0.264 67.3% | **0.257** 55.4% | **0.259** 64.4% | **0.259** 64.4% | **0.255** 59.4% | **0.308** 77.2% | **0.308** 77.2% | **0.287** 66.3% |
| GRU | 0.441 | 2% | **0.269** 62.4% | | | **0.247** 60.4% | | | 0.312 68.3% | | |
| | | 5% | 0.273 64.4% | 0.274 63.4% | | 0.250 62.4% | 0.250 62.4% | | 0.317 **69.3%** | 0.317 **69.3%** | |
| | | 10% | 0.275 65.3% | **0.270** 62.4% | 0.282 49.5% | 0.247 61.4% | 0.251 61.4% | **0.250** 53.5% | 0.317 **69.3%** | 0.317 **69.3%** | **0.295** 64.4% |
| LSTM | 0.447 | 2% | **0.273** 64.4% | | | **0.243** 52.5% | | | 0.319 70.3% | | |
| | | 5% | 0.273 66.3% | 0.275 65.3% | | 0.257 55.4% | 0.257 55.4% | | **0.316** 71.3% | 0.319 70.3% | |
| | | 10% | 0.273 **66.3%** | **0.272** 64.4% | 0.298 51.5% | 0.257 56.4% | 0.257 55.4% | 0.262 48.5% | **0.316** 71.3% | 0.320 70.3% | **0.294** 59.4% |
| MLP | 0.421 | 2% | 0.283 73.3% | | | 0.242 61.4% | | | 0.313 74.3% | | |
| | | 5% | 0.287 **77.2%** | 0.283 75.2% | | 0.249 65.3% | 0.245 63.4% | | **0.304** 77.2% | 0.306 76.2% | |
| | | 10% | **0.283** 75.2% | **0.283** 75.2% | 0.273 60.4% | 0.250 61.4% | 0.250 61.4% | **0.246** 55.4% | **0.304** 77.2% | 0.299 **76.2%** | **0.280** 66.3% |
| ResNet18 | 0.569 | 2% | 0.256 51.5% | | | 0.243 59.4% | | | 0.318 66.3% | | |
| | | 5% | **0.248** 51.5% | **0.248** 50.5% | | 0.242 **60.4%** | 0.242 **60.4%** | | 0.324 **68.3%** | 0.324 **68.3%** | |
| | | 10% | 0.257 **52.5%** | 0.256 51.5% | 0.255 43.6% | **0.240** 56.4% | 0.244 58.4% | **0.237** 54.5% | 0.323 67.3% | **0.318** 66.3% | **0.302** 61.4% |

**Evaluation of MAE on Traffic and Electricity.** Table 16, Table 17 report MAE and FR of MIA on Traffic, Electricity and Exchange respectively, as a supplement to Table 4 and Table 5 in the main paper. We observe that MIA consistently reduces MAE of the pretrained models, similar to

Table 18: (Exchange) $c_1$ $c_2\%$ report MSE FR% of MIA at different $L_{\text{atk}}$ and $L_{\text{def}}$.

| Model | Baseline | $L_{\text{mask}}$ | $\Delta = 1.0$ | | | $\Delta = 1.2$ | | | $\Delta = 1.5$ | | |
|---|---|---|---|---|---|---|---|---|---|---|---|
| | | | 2% | 5% | 10% | 2% | 5% | 10% | 2% | 5% | 10% |
| MLP-Mixer | 0.030 | 2% | 0.060 88.1% | | | 0.100 87.1% | | | 0.158 87.1% | | |
| | | 5% | **0.055** 83.2% | **0.053** 80.2% | | 0.096 84.2% | 0.096 84.2% | | 0.144 83.2% | **0.138** 81.2% | |
| | | 10% | 0.056 83.2% | 0.053 81.2% | **0.051** 78.2% | **0.095** 84.2% | **0.095** 84.2% | 0.088 80.2% | **0.141** 82.2% | 0.141 82.2% | **0.134** 79.2% |
| GRU | 0.040 | 2% | 0.062 83.2% | | | 0.104 83.2% | | | 0.152 84.2% | | |
| | | 5% | **0.060** 79.2% | **0.058** 75.2% | | 0.105 85.1% | **0.099** 80.2% | | **0.150** 83.2% | **0.140** 77.2% | |
| | | 10% | 0.061 82.2% | 0.060 80.2% | 0.058 68.3% | **0.103** 86.1% | 0.100 84.2% | 0.098 75.2% | 0.152 84.2% | 0.152 84.2% | 0.142 78.2% |
| LSTM | 0.042 | 2% | **0.061** 83.2% | | | **0.098** 87.1% | | | 0.162 85.1% | | |
| | | 5% | 0.062 82.2% | **0.058** 78.2% | | 0.099 86.1% | **0.098** 83.2% | | **0.159** 83.2% | **0.152** 79.2% | |
| | | 10% | 0.063 82.2% | 0.062 80.2% | 0.055 75.2% | 0.099 86.1% | 0.099 85.1% | 0.098 82.2% | 0.160 84.2% | 0.160 84.2% | **0.155** 82.2% |
| MLP | 0.658 | 2% | 0.065 27.7% | | | 0.110 41.6% | | | 0.210 56.4% | | |
| | | 5% | **0.057** 26.7% | 0.072 19.8% | | **0.105** 44.6% | **0.094** 27.7% | | **0.209** 53.5% | 0.231 40.6% | |
| | | 10% | **0.057** 26.7% | **0.057** 26.7% | 0.069 18.8% | 0.107 44.6% | 0.106 43.6% | 0.105 35.6% | 0.213 51.5% | **0.216** 52.5% | 0.240 39.6% |
| ResNet18 | 0.053 | 2% | 0.060 78.2% | | | 0.096 83.2% | | | 0.163 88.1% | | |
| | | 5% | **0.058** 79.2% | **0.058** 77.2% | | 0.093 81.2% | 0.093 80.2% | | **0.163** 87.1% | **0.157** 85.1% | |
| | | 10% | 0.058 78.2% | 0.060 78.2% | 0.059 72.3% | 0.091 78.2% | 0.092 78.2% | 0.090 76.2% | 0.163 87.1% | 0.163 87.1% | **0.159** 85.1% |

Table 19: (Exchange) $c_1$ $c_2\%$ report MAE FR% of MIA at different $L_{\text{adv}}$ and $L_{\text{mask}}$.

| Model | Baseline | $L_{\text{mask}}$ | $\Delta = 1.0$ | | | $\Delta = 1.2$ | | | $\Delta = 1.5$ | | |
|---|---|---|---|---|---|---|---|---|---|---|---|
| | | | 2% | 5% | 10% | 2% | 5% | 10% | 2% | 5% | 10% |
| MLP-Mixer | 0.133 | 2% | 0.209 88.1% | | | 0.270 87.1% | | | 0.349 87.1% | | |
| | | 5% | **0.199** 83.2% | **0.194** 80.2% | | 0.264 84.2% | 0.264 84.2% | | 0.333 83.2% | **0.326** 81.2% | |
| | | 10% | 0.200 83.2% | 0.195 81.2% | **0.190** 78.2% | **0.263** 84.2% | **0.263** 84.2% | 0.252 80.2% | 0.330 82.2% | 0.330 82.2% | **0.320** 79.2% |
| GRU | 0.152 | 2% | 0.211 83.2% | | | **0.274** 83.2% | | | 0.342 84.2% | | |
| | | 5% | **0.206** 79.2% | **0.203** 75.2% | | 0.276 85.1% | **0.267** 80.2% | | **0.339** 83.2% | **0.325** 77.2% | |
| | | 10% | 0.210 82.2% | 0.209 80.2% | 0.203 68.3% | 0.274 86.1% | 0.270 84.2% | 0.263 75.2% | 0.342 84.2% | 0.342 84.2% | **0.330** 78.2% |
| LSTM | 0.159 | 2% | **0.209** 83.2% | | | **0.266** 87.1% | | | 0.352 85.1% | | |
| | | 5% | 0.210 82.2% | **0.203** 78.2% | | 0.266 86.1% | **0.264** 83.2% | | **0.348** 83.2% | **0.340** 79.2% | |
| | | 10% | 0.213 82.2% | 0.211 80.2% | **0.199** 75.2% | 0.266 86.1% | 0.266 85.1% | 0.263 82.2% | 0.350 84.2% | 0.350 84.2% | **0.344** 82.2% |
| MLP | 0.639 | 2% | 0.213 27.7% | | | 0.284 41.6% | | | 0.409 56.4% | | |
| | | 5% | **0.196** 26.7% | 0.229 19.8% | | **0.274** 44.6% | **0.260** 27.7% | | **0.405** 53.5% | 0.447 40.6% | |
| | | 10% | **0.196** 26.7% | **0.196** 26.7% | 0.227 18.8% | 0.277 44.6% | 0.274 43.6% | 0.275 35.6% | 0.412 51.5% | **0.416** 52.5% | 0.458 39.6% |
| ResNet18 | 0.181 | 2% | 0.205 78.2% | | | 0.263 83.2% | | | 0.355 88.1% | | |
| | | 5% | **0.201** 79.2% | **0.201** 77.2% | | 0.257 81.2% | 0.257 80.2% | | **0.354** 87.1% | **0.347** 85.1% | |
| | | 10% | 0.202 78.2% | 0.205 78.2% | **0.201** 72.3% | **0.254** 78.2% | **0.256** 78.2% | 0.253 76.2% | 0.354 87.1% | 0.354 87.1% | **0.349** 85.1% |

the results on MSE. Our comprehensive experiments suggest that MIA can effectively improve our forecasting quality in the way of filtering the unconfident forecasts.

**Evaluation of MSE and MAE for MIA on Exchange.** Table 18 and Table 19 evaluate MSE/MAE of MIA on exchange. We observe that, MIA moderately increase MSE/MAE of the pretrained models because of the information loss for the discretization technique. Specifically, we can see that MSE/MAE of the pretrained models are commonly much smaller than that of Traffic and Electricity because Exchange dataset is much simpler. The better forecasting performance implies that we might lose more information for the discretization technique. On Exchange, information loss plays a more conspicuous role than the filtering function of MIA, causing the increase of MSE/MAE. This suggests that discretization technique might lower the forecasting performance when the pretrained models are precise enough.

**Analysis of training algorithm of imputation models.** Table 20 reportS MSE and FR of masked training and random training on the TSF dataset Traffic. Similar to the comparison on TSC datasets (Table 6 in the main paper), our masked training consistently achieves lower MSE than random training on different imputation models.

**Analysis of Different Pretrained Models on MIA.** Table 21 reports the MAE of MIA with different pretrained models, as a supplement to Table 7 in the main paper. The results demonstrate that MLP-Mixer outperforms all the other defenses.

## F MIA ON MULTIVARIATE TIME SERIES FORECASTING (MTSF)

**Apply MIA to Multivariate.** Suppose the input series is a $d_0$-dimension $t_0$-length matrix $\mathcal{X} \in \mathbb{R}^{t_0 \times d_0}$. We can represent the multivariate series uniquely by $d_0$ univariate series $\mathbf{x}_{1:t_0}^{(d)}, d = 1, 2, \ldots, d_0$.

$$\mathcal{X} = [\mathbf{x}_{1:t_0}^{(1)}, \ \mathbf{x}_{2:t_0}^{(2)}, \ \ldots, \ \mathbf{x}_{1:t_0}^{(d_0)}]^T \tag{23}$$

Table 20: (Traffic) Comparison between mask methods. $c_1$ $c_2$% report MSE TPR% of MIA across $L_{\text{def}} = 2\%, 5\%, 10\%$, $L_{\text{atk}} = 2\%, 5\%, 10\%$, $\Delta = 1.0, 1.2, 1.5$. $c_1$. MSE$(c_1)$ reports the forecasting performance.

| Model | Mask Method | $L_{\text{mask}}$ | $\Delta = 1.0$ 2% | 5% | 10% | $\Delta = 1.2$ 2% | 5% | 10% | $\Delta = 1.5$ 2% | 5% | 10% |
|---|---|---|---|---|---|---|---|---|---|---|---|
| GRU | random | 2% | **0.068** 53.5% | | | **0.071** 63.4% | | | **0.144** 76.2% | | |
| | | 5% | **0.071** 42.6% | 0.075 28.7% | | **0.063** 50.5% | **0.057** 43.6% | | **0.145** 73.3% | **0.135** 62.4% | |
| | | 10% | 0.073 30.7% | **0.069** 26.7% | 0.075 23.8% | **0.052** 49.5% | **0.048** 45.5% | **0.051** 40.6% | **0.140** 66.3% | **0.138** 64.4% | **0.132** 59.4% |
| | block | 2% | 0.074 **64.4%** | | | 0.073 **66.3%** | | | 0.148 **84.2%** | | |
| | | 5% | 0.073 **66.3%** | **0.075** 61.4% | | 0.073 **66.3%** | 0.073 **65.3%** | | 0.148 **83.2%** | 0.145 **80.2%** | |
| | | 10% | **0.073** 64.4% | 0.074 63.4% | **0.070** 57.4% | 0.074 **65.3%** | 0.074 **64.4%** | 0.065 **59.4%** | 0.148 **84.2%** | 0.148 **84.2%** | 0.145 **79.2%** |
| LSTM | random | 2% | **0.064** 55.4% | | | 0.073 62.4% | | | 0.144 81.2% | | |
| | | 5% | **0.063** 37.6% | **0.050** 25.7% | | **0.051** 49.5% | **0.055** 38.6% | | **0.129** 67.3% | **0.130** 58.4% | |
| | | 10% | **0.065** 37.6% | **0.053** 27.7% | **0.049** 14.9% | **0.050** 48.5% | **0.053** 45.5% | **0.046** 30.7% | **0.134** 68.3% | **0.128** 64.4% | **0.120** 53.5% |
| | block | 2% | 0.071 61.4% | | | **0.069** 68.3% | | | **0.144** 85.1% | | |
| | | 5% | 0.067 **59.4%** | 0.067 **56.4%** | | 0.069 **68.3%** | 0.068 **66.3%** | | 0.144 **85.1%** | 0.141 **82.2%** | |
| | | 10% | 0.068 **60.4%** | 0.067 **58.4%** | 0.067 **56.4%** | 0.069 **67.3%** | 0.069 **66.3%** | 0.067 **65.3%** | 0.144 **85.1%** | 0.144 **85.1%** | 0.142 **83.2%** |
| MLP-Mixer | random | 2% | **0.073** 54.5% | | | 0.085 66.3% | | | **0.147** 75.2% | | |
| | | 5% | **0.074** 44.6% | **0.067** 27.7% | | **0.072** 59.4% | **0.061** 47.5% | | **0.137** 74.3% | **0.142** 63.4% | |
| | | 10% | **0.072** 42.6% | **0.073** 40.6% | **0.059** 26.7% | **0.065** 57.4% | **0.067** 53.5% | **0.061** 43.6% | **0.134** 74.3% | **0.130** 70.3% | **0.130** 61.4% |
| | block | 2% | 0.079 **67.3%** | | | 0.085 **74.3%** | | | 0.151 **86.1%** | | |
| | | 5% | 0.076 **67.3%** | 0.078 **61.4%** | | 0.083 **76.2%** | 0.079 **72.3%** | | 0.151 **86.1%** | 0.149 **83.2%** | |
| | | 10% | 0.079 **63.4%** | 0.077 **61.4%** | 0.076 **55.4%** | 0.081 **72.3%** | 0.077 **71.3%** | 0.078 **69.3%** | 0.151 **86.1%** | 0.150 **85.1%** | 0.145 **82.2%** |
| MLP | random | 2% | **0.069** 53.5% | | | 0.074 61.4% | | | **0.145** 75.2% | | |
| | | 5% | **0.072** 44.6% | 0.071 37.6% | | **0.059** 58.4% | **0.044** 47.5% | | **0.138** 78.2% | **0.138** 67.3% | |
| | | 10% | **0.072** 42.6% | 0.073 40.6% | 0.077 34.7% | **0.061** 53.5% | **0.062** 52.5% | **0.055** 48.5% | **0.140** 78.2% | **0.140** 75.2% | **0.133** 67.3% |
| | block | 2% | 0.072 **60.4%** | | | 0.068 **68.3%** | | | 0.146 **84.2%** | | |
| | | 5% | **0.071** 61.4% | 0.071 **58.4%** | | 0.070 **66.3%** | 0.070 **66.3%** | | 0.145 **86.1%** | 0.147 **85.1%** | |
| | | 10% | 0.073 **58.4%** | 0.072 **59.4%** | 0.069 **52.5%** | 0.065 **66.3%** | 0.065 **66.3%** | 0.065 **62.4%** | 0.146 **84.2%** | 0.147 **85.1%** | 0.144 **82.2%** |
| ResNet18 | random | 2% | **0.071** 57.4% | | | **0.083** 66.3% | | | **0.143** 82.2% | | |
| | | 5% | **0.073** 33.7% | **0.072** 29.7% | | **0.063** 54.5% | **0.057** 46.5% | | **0.136** 62.4% | **0.139** 55.4% | |
| | | 10% | **0.067** 32.7% | **0.070** 27.7% | 0.076 17.8% | **0.059** 49.5% | **0.061** 49.5% | **0.047** 37.6% | **0.143** 61.4% | **0.130** 53.5% | **0.117** 45.5% |
| | block | 2% | 0.073 **63.4%** | | | 0.089 **71.3%** | | | 0.146 **86.1%** | | |
| | | 5% | 0.074 **62.4%** | 0.074 **62.4%** | | 0.085 **68.3%** | 0.084 **67.3%** | | 0.147 **87.1%** | 0.146 **86.1%** | |
| | | 10% | 0.075 **60.4%** | 0.072 **59.4%** | **0.070** 57.4% | 0.087 **68.3%** | 0.084 **69.3%** | 0.076 **62.4%** | 0.143 **84.2%** | 0.143 **84.2%** | 0.140 **82.2%** |

Table 21: Comparison of four imputation models. The best results are shown in bold-face.

| Generator | Metric | $L_{\text{mask}}$ | Electricity 2% | 5% | 10% | Exchange 2% | 5% | 10% | Traffic 2% | 5% | 10% |
|---|---|---|---|---|---|---|---|---|---|---|---|
| MLP-Mixer | MAE | 2% | 0.312 **77.2%** | | | 0.349 **87.1%** | | | 0.340 **89.1%** | | |
| | | 5% | 0.308 **78.2%** | 0.308 **78.2%** | | 0.333 **83.2%** | 0.326 **81.2%** | | 0.336 **90.1%** | 0.340 **89.1%** | |
| | | 10% | 0.308 **77.2%** | 0.308 **77.2%** | 0.287 **66.3%** | 0.330 **82.2%** | 0.330 **82.2%** | 0.320 **79.2%** | 0.338 **91.1%** | 0.336 **90.1%** | 0.333 **88.1%** |
| BRITS | MAE | 2% | 0.290 65.3% | | | 0.299 40.6% | | | 0.338 87.1% | | |
| | | 5% | 0.283 71.3% | 0.264 59.4% | | 0.288 49.5% | 0.246 15.8% | | 0.327 84.2% | 0.320 75.2% | |
| | | 10% | 0.271 61.4% | 0.272 60.4% | 0.265 45.5% | 0.381 10.9% | 0.526 1.0% | Inf 0.0% | 0.315 79.2% | 0.319 77.2% | 0.312 73.3% |
| SAITS | MAE | 2% | 0.269 58.4% | | | **0.258** 25.7% | | | **0.307** 67.3% | | |
| | | 5% | **0.261** 53.5% | **0.256** 52.5% | | 0.317 43.6% | **0.264** 27.7% | | 0.321 73.3% | **0.308** 68.3% | |
| | | 10% | 0.274 56.4% | 0.278 54.5% | **0.263** 44.6% | 0.179 7.9% | **0.123** 6.9% | **0.035** 4.0% | 0.316 67.3% | 0.325 69.3% | 0.315 59.4% |
| Transformer | MAE | 2% | **0.253** 52.5% | | | 0.329 23.8% | | | 0.322 68.3% | | |
| | | 5% | 0.288 52.5% | 0.291 42.6% | | **0.164** 6.9% | **0.030** 2.0% | | **0.318** 74.3% | 0.312 65.3% | |
| | | 10% | **0.266** 60.4% | **0.263** 58.4% | 0.274 34.7% | **0.112** 8.9% | 0.135 7.9% | 0.037 5.0% | **0.313** 75.2% | **0.317** 74.3% | **0.312** 61.4% |

1. We generate the masks in the same way as **Masking (Step 1)**. The main difference is the way we mask the multivariate series with the univariate mask. Masking multivariate series $\mathcal{X}$ with the mask $\text{M}_{[u:v]}$ is computed as follow:

$$\mathcal{X} \odot \text{M}_{[u:v]} = \left[ \mathbf{x}_{1:t_0}^{(1)} \odot \text{M}_{[u:v]},\ \mathbf{x}_{2:t_0}^{(2)} \odot \text{M}_{[u:v]},\ \ldots, \mathbf{x}_{1:t_0}^{(d_0)} \odot \text{M}_{[u:v]} \right]^T \quad (24)$$

2. With the imputation model $G(\cdot)$, **Imputing (Step 2)** for multivariate series $\mathcal{X}$ is computed as follow:

$$G(\mathcal{X} \odot \text{M}_{[u:v]}) = \left[ G\left( \mathbf{x}_{1:t_0}^{(1)} \odot \text{M}_{[u:v]} \right),\ G\left( \mathbf{x}_{2:t_0}^{(2)} \odot \text{M}_{[u:v]} \right),\ \ldots,\ G\left( \mathbf{x}_{1:t_0}^{(d_0)} \odot \text{M}_{[u:v]} \right) \right]^T \quad (25)$$

3. We aggregate all the predictions of the imputed multivariate series in the same way as **Aggregation (Step 3)** for univariate series.

**Evaluation of MIA on multivariate tasks.** In terms of the multivariate time series forecasting (MTSF) task, we follow the work (Wu et al., 2021) and evaluate our MIA framework on four datasets (ETTh2 (Zhou et al., 2021), ETTm2 (Zhou et al., 2021), weather [8] and illness [9]. Corresponding results are presented in Table 22 to 29. Extensive experiments demonstrate that MIA behaves similarly to that of univariate forecasting tasks.

---

[8] https://www.bgc-jena.mpg.de/wetter/
[9] https://gis.cdc.gov/grasp/fluview/fluportaldashboard.html

Table 22: (ETTh2) Evaluate MAE of MIA with pretrained models on multi-variate time series forecasting (MTSF).

| Model | Baseline | $L_{\text{mask}}$ | $\Delta = 1.0$ 2 % | 5 % | 10 % | $\Delta = 1.2$ 2 % | 5 % | 10 % | $\Delta = 1.5$ 2 % | 5 % | 10 % |
|---|---|---|---|---|---|---|---|---|---|---|---|
| MLP-Mixer | 0.712 | 2 % | 0.363 74.9% | | | 0.434 82.4% | | | 0.550 90.3% | | |
| | | 5 % | 0.363 **75.8%** | 0.360 73.8% | | 0.435 **83.6%** | 0.432 81.8% | | 0.552 **90.8%** | 0.547 89.8% | |
| | | 10 % | **0.360** 75.0% | **0.356** 74.2% | **0.346** 70.1% | **0.431** 82.6% | **0.427** 82.2% | **0.415** 77.5% | **0.548** 90.5% | **0.542** 90.1% | **0.528** 86.9% |
| MLP | 0.815 | 2 % | **0.368** 68.4% | | | **0.436** 76.5% | | | **0.542** 85.5% | | |
| | | 5 % | 0.371 **69.0%** | 0.365 **66.7%** | | 0.438 76.6% | **0.434** 75.5% | | 0.547 **86.6%** | **0.541** 85.1% | |
| | | 10 % | 0.368 67.0% | **0.365** 66.5% | **0.356** 64.4% | 0.438 76.1% | 0.435 **75.9%** | **0.424** 73.3% | 0.545 86.3% | 0.541 **85.8%** | **0.527** 83.5% |
| LSTM | 0.802 | 2 % | 0.374 **69.1%** | | | 0.444 76.6% | | | **0.556** 85.7% | | |
| | | 5 % | 0.375 68.7% | **0.372** 67.9% | | 0.447 **77.1%** | 0.443 **76.0%** | | 0.558 **85.9%** | 0.552 84.7% | |
| | | 10 % | **0.373** 68.4% | 0.373 **68.0%** | **0.370** 66.7% | 0.445 76.7% | **0.443** 76.0% | **0.436** 74.0% | 0.556 85.9% | **0.551** 85.2% | **0.542** 83.6% |
| GRU | 0.782 | 2 % | 0.373 71.6% | | | 0.441 78.1% | | | 0.563 87.1% | | |
| | | 5 % | 0.374 **72.4%** | **0.371** 70.0% | | 0.442 **78.3%** | 0.440 77.2% | | 0.565 **87.6%** | 0.560 85.9% | |
| | | 10 % | **0.372** 72.0% | 0.372 **71.6%** | **0.369** 68.1% | **0.440** 78.0% | **0.439** 77.6% | **0.435** 75.4% | **0.562** 87.3% | **0.556** 86.9% | **0.545** 84.2% |
| RNN | 0.809 | 2 % | 0.384 69.9% | | | 0.454 79.0% | | | 0.560 **87.6%** | | |
| | | 5 % | 0.383 **70.2%** | 0.382 **68.8%** | | 0.454 79.2% | 0.453 78.4% | | 0.560 87.6% | 0.559 **87.0%** | |
| | | 10 % | **0.381** 69.3% | **0.380** 68.4% | **0.378** 67.1% | **0.452** 78.7% | **0.451** 78.3% | **0.448** 76.9% | **0.559** 87.5% | **0.556** 87.0% | **0.551** 86.1% |
| TransformerNormal | 0.852 | 2 % | 0.405 **67.0%** | | | 0.481 **75.2%** | | | 0.593 **83.8%** | | |
| | | 5 % | 0.404 66.6% | 0.403 **66.1%** | | 0.480 75.1% | 0.480 74.8% | | 0.592 83.4% | 0.591 **83.3%** | |
| | | 10 % | **0.400** 65.1% | **0.397** 64.6% | **0.393** 63.1% | **0.478** 74.5% | **0.477** 74.3% | **0.473** 73.4% | **0.589** 83.1% | **0.587** 82.4% | **0.583** 81.9% |
| TransformerPadding | 0.981 | 2 % | 0.346 **57.9%** | | | 0.427 65.3% | | | 0.553 **76.7%** | | |
| | | 5 % | 0.346 57.8% | 0.346 **57.8%** | | 0.426 65.3% | 0.426 65.3% | | 0.552 76.5% | 0.552 **76.5%** | |
| | | 10 % | **0.344** 57.8% | **0.344** 57.8% | **0.344** 57.3% | **0.425** 65.4% | **0.425** 65.4% | **0.424** 65.3% | **0.550** 76.5% | **0.550** 76.5% | **0.550** 76.5% |
| TransformerConv | 0.934 | 2 % | 0.353 **61.1%** | | | 0.427 70.2% | | | 0.538 **79.4%** | | |
| | | 5 % | 0.352 61.0% | 0.351 **60.6%** | | 0.426 **70.3%** | 0.424 **69.7%** | | 0.537 **79.4%** | 0.536 **79.4%** | |
| | | 10 % | **0.350** 60.2% | **0.349** 60.1% | **0.348** 59.7% | **0.423** 69.7% | **0.423** 69.6% | **0.419** 68.7% | **0.535** 79.3% | **0.534** 79.3% | **0.531** 79.1% |

Table 23: (ETTh2) Evaluate MSE of MIA with different pretrained models on multi-variate time series forecasting (MTSF).

| Model | Baseline | $L_{\text{mask}}$ | $\Delta = 1.0$ 2 % | 5 % | 10 % | $\Delta = 1.2$ 2 % | 5 % | 10 % | $\Delta = 1.5$ 2 % | 5 % | 10 % |
|---|---|---|---|---|---|---|---|---|---|---|---|
| MLP-Mixer | 0.930 | 2 % | 0.159 74.9% | | | 0.230 82.4% | | | 0.367 90.3% | | |
| | | 5 % | 0.159 **75.8%** | 0.157 73.8% | | 0.231 **83.6%** | 0.228 81.8% | | 0.370 **90.8%** | 0.365 89.8% | |
| | | 10 % | **0.157** 75.0% | **0.155** 74.2% | **0.149** 70.1% | **0.228** 82.6% | **0.225** 82.2% | **0.215** 77.5% | **0.366** 90.5% | **0.360** 90.1% | **0.347** 86.9% |
| MLP | 1.143 | 2 % | 0.162 68.4% | | | 0.230 76.5% | | | **0.357** 85.5% | | |
| | | 5 % | 0.163 **69.0%** | 0.160 **66.7%** | | 0.231 76.6% | **0.228** 75.5% | | 0.362 **86.6%** | **0.356** 85.1% | |
| | | 10 % | **0.162** 67.0% | **0.159** 66.5% | **0.153** 64.4% | 0.230 76.1% | 0.228 **75.9%** | **0.219** 73.3% | 0.360 86.3% | 0.357 **85.8%** | **0.342** 83.5% |
| LSTM | 1.072 | 2 % | 0.165 **69.1%** | | | 0.235 76.6% | | | 0.370 85.7% | | |
| | | 5 % | 0.165 68.7% | **0.164** 67.9% | | 0.237 **77.1%** | 0.234 **76.0%** | | 0.371 **85.9%** | 0.366 84.7% | |
| | | 10 % | **0.164** 68.4% | 0.164 **68.0%** | **0.162** 66.7% | 0.236 76.7% | 0.233 76.0% | **0.228** 74.0% | **0.369** 85.9% | **0.365** 85.2% | **0.357** 83.6% |
| GRU | 1.042 | 2 % | 0.166 71.6% | | | 0.236 78.1% | | | 0.378 87.1% | | |
| | | 5 % | 0.167 **72.4%** | **0.165** 70.0% | | 0.237 **78.3%** | 0.235 77.2% | | 0.380 **87.6%** | 0.376 85.9% | |
| | | 10 % | **0.166** 72.0% | 0.165 **71.6%** | **0.161** 68.1% | **0.235** 78.0% | **0.233** 77.6% | **0.229** 75.4% | **0.377** 87.3% | **0.373** 86.9% | **0.364** 84.2% |
| RNN | 1.074 | 2 % | 0.171 69.9% | | | 0.244 79.0% | | | 0.375 **87.6%** | | |
| | | 5 % | 0.171 **70.2%** | 0.170 **68.8%** | | 0.243 79.2% | 0.243 78.4% | | 0.375 87.6% | 0.373 **87.0%** | |
| | | 10 % | **0.170** 69.3% | **0.169** 68.4% | **0.166** 67.1% | **0.242** 78.7% | **0.241** 78.3% | **0.238** 76.9% | **0.373** 87.5% | **0.370** 87.0% | **0.366** 86.1% |
| TransformerNormal | 1.241 | 2 % | 0.186 **67.0%** | | | 0.264 **75.2%** | | | 0.406 **83.8%** | | |
| | | 5 % | 0.185 66.6% | 0.184 **66.1%** | | 0.263 75.1% | 0.262 74.8% | | 0.404 83.4% | 0.403 **83.3%** | |
| | | 10 % | **0.182** 65.1% | **0.180** 64.6% | **0.177** 63.1% | **0.261** 74.5% | **0.259** 74.3% | **0.256** 73.4% | **0.401** 83.1% | **0.398** 82.4% | **0.394** 81.9% |
| TransformerPadding | 1.576 | 2 % | 0.149 **57.9%** | | | 0.222 65.3% | | | 0.362 **76.7%** | | |
| | | 5 % | 0.149 57.8% | 0.149 **57.8%** | | 0.222 65.3% | 0.222 65.3% | | 0.361 76.5% | 0.361 **76.5%** | |
| | | 10 % | **0.148** 57.8% | **0.148** 57.8% | **0.148** 57.8% | **0.221** 65.4% | **0.221** 65.4% | **0.221** 65.3% | **0.359** 76.5% | **0.359** 76.5% | **0.359** 76.5% |
| TransformerConv | 1.424 | 2 % | 0.154 **61.1%** | | | 0.224 70.2% | | | 0.353 **79.4%** | | |
| | | 5 % | 0.154 61.0% | 0.153 **60.6%** | | 0.224 **70.3%** | 0.222 **69.7%** | | 0.352 **79.4%** | 0.351 **79.4%** | |
| | | 10 % | **0.153** 60.2% | **0.152** 60.1% | **0.151** 59.7% | **0.222** 69.7% | **0.221** 69.6% | **0.219** 68.7% | **0.350** 79.3% | **0.349** 79.3% | **0.347** 79.1% |

# G  IMPUTATION MODELS.

For the imputation model architectures, we take SAITS, Transformer and BRITS and MLP-Mixer. Specifically, for SAITS, we use the code from [10]. For SAITS (Du et al., 2022), we set $d_{\text{model}} = 32, n_{\text{layers}} = 2, d_{\text{inner}} = 16, n_{\text{head}} = 4, d_{\text{k}} = 8, d_{\text{v}} = 8$. For Transformer (Vaswani et al., 2017), we set $d_{\text{model}} = 32, n_{\text{layers}} = 2, d_{\text{inner}} = 16, n_{\text{head}} = 4, d_{\text{k}} = 8, d_{\text{v}} = 8$. For BRITS (Cao et al., 2018), we set $h_{\text{hidden}} = 32$. For MLP-Mixer (Tolstikhin et al., 2021), we use the same structure as Table 12. In terms of training, we use optimizer Adam with $lr = 0.0001, \beta = (0.9, 0.999), \epsilon = 10^{-8}$, weight_decay $= 0$ and train the model for 30 epochs.

## G.1  CHOICE OF IMPUTATION MODEL ARCHITECTURE.

Table 14 compares the imputation quality of different imputation models on Traffic. The imputation quality is quantified by the mean square error (MSE) between the imputed series and the original series. We observe that MLP-Mixer consistently outperforms other three models across different datasets and $L_{\text{adv}}$, indicating the superior imputation ability of MLP-Mixer architecture.

---

[10]https://github.com/WenjieDu/PyPOTS

Table 24: (ETTm2) Evaluate MAE of MIA with different pretrained models on multi-variate time series forecasting (MTSF).

| Model | Baseline | $L_{mask}$ | $\Delta = 1.0$ | | | $\Delta = 1.2$ | | | $\Delta = 1.5$ | | |
|---|---|---|---|---|---|---|---|---|---|---|---|
| | | | 2% | 5% | 10% | 2% | 5% | 10% | 2% | 5% | 10% |
| MLP-Mixer | 0.858 | 2% | **0.329** 67.6% | | | **0.404** 75.5% | | | **0.526** 83.7% | | |
| | | 5% | **0.340** **68.2%** | **0.312** 62.3% | | 0.421 **76.1%** | **0.385** 70.7% | | 0.548 **84.3%** | **0.504** 80.1% | |
| | | 10% | 0.337 67.2% | 0.336 **66.7%** | 0.299 56.4% | 0.414 75.3% | 0.411 **75.0%** | 0.361 65.0% | 0.538 82.9% | 0.533 **82.8%** | 0.467 75.6% |
| MLP | 0.755 | 2% | **0.350** 74.2% | | | **0.427** 82.0% | | | **0.546** 86.5% | | |
| | | 5% | 0.366 **76.3%** | **0.325** 70.2% | | 0.453 **85.1%** | **0.408** 80.7% | | 0.577 **87.4%** | **0.527** 86.0% | |
| | | 10% | 0.357 74.6% | 0.351 **72.0%** | 0.324 61.9% | 0.445 83.8% | 0.440 **82.4%** | 0.411 72.5% | 0.569 86.8% | 0.567 **87.0%** | 0.529 80.8% |
| LSTM | 1.034 | 2% | 0.364 **64.6%** | | | 0.462 **72.9%** | | | 0.594 **82.4%** | | |
| | | 5% | 0.363 64.2% | 0.357 **63.9%** | | 0.460 72.1% | 0.454 **71.3%** | | 0.594 81.9% | **0.589** 81.6% | |
| | | 10% | **0.359** 63.8% | **0.355** 63.2% | 0.347 62.1% | 0.456 71.3% | 0.452 70.6% | 0.444 69.2% | **0.591** 81.7% | 0.589 **81.8%** | 0.583 81.4% |
| GRU | 1.065 | 2% | 0.388 **62.4%** | | | 0.479 **69.9%** | | | 0.619 **78.9%** | | |
| | | 5% | 0.388 61.8% | 0.371 **60.2%** | | 0.479 69.6% | 0.463 **67.6%** | | 0.620 78.3% | **0.604** 76.8% | |
| | | 10% | **0.382** 59.9% | 0.375 59.0% | 0.353 54.7% | 0.476 67.5% | 0.468 66.5% | 0.445 62.3% | **0.616** 77.2% | 0.610 **77.1%** | 0.589 74.2% |
| RNN | 1.024 | 2% | **0.299** 61.7% | | | **0.363** 70.1% | | | **0.462** 77.4% | | |
| | | 5% | 0.310 64.2% | **0.273** 56.9% | | 0.375 **71.6%** | **0.338** 65.6% | | 0.481 **78.2%** | **0.448** 73.3% | |
| | | 10% | 0.308 **64.6%** | 0.299 **60.5%** | 0.260 49.7% | 0.372 71.6% | 0.358 **68.0%** | 0.310 59.9% | 0.478 77.6% | 0.465 **75.2%** | 0.410 69.5% |
| TransformerNormal | 0.989 | 2% | 0.322 **57.5%** | | | 0.382 **67.5%** | | | 0.468 **77.6%** | | |
| | | 5% | 0.318 56.5% | 0.309 51.2% | | 0.382 66.8% | 0.369 61.8% | | 0.473 76.4% | **0.455** 73.8% | |
| | | 10% | **0.311** 53.4% | **0.305** 51.4% | 0.283 41.0% | 0.377 64.5% | 0.369 62.4% | 0.353 54.6% | 0.466 75.7% | 0.457 **74.1%** | 0.435 68.7% |
| TransformerPadding | 1.096 | 2% | **0.376** 51.8% | | | **0.445** 60.7% | | | **0.559** 73.3% | | |
| | | 5% | 0.392 48.5% | 0.393 **44.0%** | | 0.459 59.5% | 0.459 **54.0%** | | **0.552** 71.4% | **0.551** 68.3% | |
| | | 10% | **0.374** 38.2% | **0.377** 34.8% | 0.373 27.0% | 0.447 48.5% | 0.445 46.1% | 0.434 37.3% | 0.556 63.0% | 0.553 61.6% | 0.543 54.6% |
| TransformerConv | 0.869 | 2% | 0.331 **67.5%** | | | 0.401 72.3% | | | 0.519 78.8% | | |
| | | 5% | 0.314 66.7% | **0.294** 58.9% | | 0.390 **72.8%** | **0.362** 66.4% | | 0.511 **79.2%** | **0.478** 75.3% | |
| | | 10% | **0.312** 61.7% | 0.308 **60.0%** | 0.298 50.4% | 0.384 68.8% | 0.375 66.5% | 0.362 58.7% | **0.504** 77.5% | 0.488 75.1% | 0.467 68.4% |

Table 25: (ETTm2) Evaluate MSE of MIA with different pretrained models on multi-variate time series forecasting (MTSF).

| Model | Baseline | $L_{mask}$ | $\Delta = 1.0$ | | | $\Delta = 1.2$ | | | $\Delta = 1.5$ | | |
|---|---|---|---|---|---|---|---|---|---|---|---|
| | | | 2% | 5% | 10% | 2% | 5% | 10% | 2% | 5% | 10% |
| MLP-Mixer | 1.750 | 2% | **0.139** 67.6% | | | **0.206** 75.5% | | | **0.339** 83.7% | | |
| | | 5% | 0.145 **68.2%** | **0.128** 62.3% | | 0.219 **76.1%** | **0.192** 70.7% | | 0.362 **84.3%** | **0.319** 80.1% | |
| | | 10% | 0.143 67.2% | 0.142 **66.7%** | 0.121 56.4% | 0.214 75.3% | 0.212 **75.0%** | 0.176 65.0% | 0.352 82.9% | 0.348 **82.8%** | 0.287 75.6% |
| MLP | 1.508 | 2% | **0.149** 74.2% | | | **0.222** 82.0% | | | **0.358** 86.5% | | |
| | | 5% | 0.161 **76.3%** | **0.134** 70.2% | | 0.243 **85.1%** | **0.208** 80.7% | | 0.385 **87.4%** | **0.337** 86.0% | |
| | | 10% | 0.156 74.6% | 0.152 **72.0%** | 0.134 61.9% | 0.237 83.8% | 0.233 **82.4%** | 0.211 72.5% | 0.377 86.8% | 0.376 **87.0%** | 0.336 80.8% |
| LSTM | 2.858 | 2% | 0.158 **64.6%** | | | 0.242 **72.9%** | | | 0.391 **82.4%** | | |
| | | 5% | 0.157 64.2% | 0.154 **63.9%** | | 0.241 72.1% | 0.236 **71.3%** | | 0.391 81.9% | 0.386 81.6% | |
| | | 10% | 0.155 63.8% | 0.152 63.2% | 0.149 62.1% | 0.238 71.3% | 0.235 70.6% | 0.229 69.2% | 0.387 81.7% | 0.386 **81.8%** | 0.381 81.4% |
| GRU | 2.685 | 2% | 0.174 **62.4%** | | | 0.260 **69.9%** | | | 0.426 **78.9%** | | |
| | | 5% | 0.174 61.8% | 0.163 **60.2%** | | 0.261 69.6% | 0.247 **67.6%** | | 0.426 78.3% | **0.410** 76.8% | |
| | | 10% | 0.171 59.9% | 0.168 59.0% | 0.154 54.7% | 0.260 67.5% | 0.254 66.5% | 0.236 62.3% | 0.422 77.2% | 0.417 **77.1%** | 0.395 74.2% |
| RNN | 2.598 | 2% | **0.121** 61.7% | | | **0.177** 70.1% | | | **0.280** 77.4% | | |
| | | 5% | 0.127 64.2% | 0.104 56.9% | | 0.185 **71.6%** | **0.159** 65.6% | | 0.297 **78.2%** | 0.266 73.3% | |
| | | 10% | 0.125 **64.6%** | 0.120 **60.5%** | 0.096 49.7% | 0.183 71.6% | 0.173 **68.0%** | 0.138 59.9% | 0.295 77.6% | 0.285 **75.2%** | 0.233 69.5% |
| TransformerNormal | 2.150 | 2% | 0.131 **57.7%** | | | 0.186 **67.5%** | | | 0.286 **77.6%** | | |
| | | 5% | 0.128 56.5% | 0.121 51.2% | | 0.187 66.8% | 0.174 61.8% | | 0.288 76.4% | 0.269 73.8% | |
| | | 10% | **0.124** 53.4% | 0.120 **51.4%** | 0.108 41.0% | 0.181 64.5% | 0.173 62.4% | 0.163 54.6% | 0.279 75.7% | 0.268 **74.1%** | 0.249 68.7% |
| TransformerPadding | 2.828 | 2% | 0.168 **51.8%** | | | 0.238 **60.7%** | | | 0.375 **73.3%** | | |
| | | 5% | 0.175 48.5% | 0.174 **44.0%** | | 0.245 59.5% | 0.243 **54.0%** | | 0.369 71.4% | 0.367 **68.3%** | |
| | | 10% | **0.166** 38.2% | 0.168 34.8% | 0.164 27.0% | 0.236 48.5% | 0.235 46.1% | 0.226 37.3% | 0.367 63.0% | 0.365 61.6% | 0.353 54.6% |
| TransformerConv | 1.918 | 2% | 0.139 **67.5%** | | | 0.203 72.3% | | | 0.332 78.8% | | |
| | | 5% | 0.129 66.7% | **0.118** 58.9% | | 0.195 **72.8%** | **0.175** 66.4% | | 0.323 **79.2%** | **0.291** 75.3% | |
| | | 10% | **0.128** 61.7% | 0.124 **60.0%** | 0.117 50.4% | 0.192 68.8% | 0.184 **66.5%** | 0.174 58.7% | **0.318** 77.5% | 0.304 75.1% | 0.283 68.4% |

## G.2 IMPACT OF GAUSSIAN AUGMENTATION

Table 33, Table 34, Table 35, Table 36 and Table 37 report the impact of Gaussian augmentation at $\sigma = 0.01, 0.02, 0.03, 0.04, 0.05$ on time series classification dataset DistalPhalanxTW ($L_{mask} = 15\%$) respectively. We compare the MIA with Gaussian augmentation to MIA without Gaussian augmentation (baseline) on **Certified Accuracy** at $L_{atk} = 5\%, 10\%, 15\%$. The table reports the relative improvement (abs %) on certified accuracy.

## G.3 VISUALIZE THE IMPUTATION QUALITY.

Fig. 4 intuitively shows the imputation performance of different imputation models on dataset Traffic.

Table 26: (Illness) Evaluate MAE of MIA with different pretrained models on multi-variate time series forecasting (MTSF).

| Model | Baseline | $L_{\text{mask}}$ | Δ=1.0 2% | 5% | 10% | Δ=1.2 2% | 5% | 10% | Δ=1.5 2% | 5% | 10% |
|---|---|---|---|---|---|---|---|---|---|---|---|
| MLP-Mixer | 0.712 | 2% | 0.232 81.3% | | | 0.251 86.5% | | | 0.314 90.6% | | |
| | | 5% | 0.217 77.2% | 0.238 76.6% | | 0.236 81.3% | 0.227 80.7% | | 0.309 88.3% | 0.263 87.7% | |
| | | 10% | 0.216 77.2% | 0.220 77.2% | 0.230 73.1% | 0.254 84.8% | 0.252 84.2% | 0.247 82.5% | 0.315 88.9% | 0.313 88.9% | 0.286 88.9% |
| MLP | 0.930 | 2% | 0.287 54.4% | | | 0.333 66.1% | | | 0.394 82.5% | | |
| | | 5% | 0.258 51.5% | 0.255 50.9% | | 0.306 62.6% | 0.301 61.4% | | 0.377 80.1% | 0.376 79.5% | |
| | | 10% | 0.268 50.9% | 0.263 50.3% | 0.256 50.3% | 0.328 65.5% | 0.325 65.5% | 0.311 63.7% | 0.388 81.9% | 0.389 81.9% | 0.373 79.5% |
| LSTM | 1.079 | 2% | 0.239 45.6% | | | 0.291 59.6% | | | 0.356 77.8% | | |
| | | 5% | 0.244 45.6% | 0.243 45.6% | | 0.297 59.6% | 0.290 59.6% | | 0.355 77.2% | 0.348 77.2% | |
| | | 10% | 0.244 45.6% | 0.243 45.6% | 0.239 45.6% | 0.296 59.6% | 0.296 59.6% | 0.292 59.6% | 0.358 77.2% | 0.355 76.6% | 0.351 76.6% |
| GRU | 0.987 | 2% | 0.300 54.4% | | | 0.336 67.3% | | | 0.373 80.1% | | |
| | | 5% | 0.296 54.4% | 0.299 52.0% | | 0.334 67.3% | 0.343 66.7% | | 0.368 78.9% | 0.373 78.9% | |
| | | 10% | 0.294 54.4% | 0.294 54.4% | 0.296 54.4% | 0.332 67.3% | 0.331 67.3% | 0.333 67.3% | 0.374 80.1% | 0.373 80.1% | 0.368 79.5% |
| RNN | 1.150 | 2% | 0.296 38.6% | | | 0.365 55.0% | | | 0.390 63.7% | | |
| | | 5% | 0.300 38.0% | 0.288 38.0% | | 0.361 52.6% | 0.353 52.6% | | 0.394 63.2% | 0.388 63.2% | |
| | | 10% | 0.297 38.0% | 0.297 38.0% | 0.293 38.0% | 0.360 52.6% | 0.357 52.0% | 0.354 52.0% | 0.394 63.2% | 0.390 62.6% | 0.387 62.6% |
| TransformerNormal | 0.877 | 2% | 0.244 58.5% | | | 0.301 70.2% | | | 0.376 82.5% | | |
| | | 5% | 0.253 55.0% | 0.255 55.0% | | 0.289 67.8% | 0.285 66.7% | | 0.337 79.5% | 0.327 78.4% | |
| | | 10% | 0.242 56.7% | 0.231 55.6% | 0.232 54.4% | 0.280 67.8% | 0.277 67.8% | 0.272 66.1% | 0.336 80.1% | 0.330 79.5% | 0.326 79.5% |
| TransformerPadding | 0.945 | 2% | 0.290 54.4% | | | 0.326 69.0% | | | 0.362 84.2% | | |
| | | 5% | 0.307 52.6% | 0.305 50.3% | | 0.328 66.1% | 0.326 64.3% | | 0.353 81.3% | 0.358 81.3% | |
| | | 10% | 0.297 50.9% | 0.298 49.7% | 0.296 48.0% | 0.341 67.3% | 0.348 66.7% | 0.351 65.5% | 0.372 83.6% | 0.374 82.5% | 0.381 81.9% |
| TransformerConv | 0.820 | 2% | 0.272 66.1% | | | 0.316 73.7% | | | 0.395 80.1% | | |
| | | 5% | 0.266 63.2% | 0.276 60.8% | | 0.317 74.9% | 0.315 72.5% | | 0.385 79.5% | 0.372 78.9% | |
| | | 10% | 0.258 63.2% | 0.263 61.4% | 0.263 56.7% | 0.303 70.8% | 0.302 70.2% | 0.309 67.8% | 0.392 78.9% | 0.385 78.9% | 0.380 77.8% |

Table 27: (Illness) Evaluate MSE of MIA with different pretrained models on multi-variate time series forecasting (MTSF).

| Model | Baseline | $L_{\text{mask}}$ | Δ=1.0 2% | 5% | 10% | Δ=1.2 2% | 5% | 10% | Δ=1.5 2% | 5% | 10% |
|---|---|---|---|---|---|---|---|---|---|---|---|
| MLP-Mixer | 0.970 | 2% | 0.075 81.3% | | | 0.095 86.5% | | | 0.156 90.6% | | |
| | | 5% | 0.067 77.2% | 0.076 76.6% | | 0.084 81.3% | 0.078 80.7% | | 0.148 88.3% | 0.120 87.7% | |
| | | 10% | 0.069 77.2% | 0.071 77.2% | 0.075 73.1% | 0.095 84.8% | 0.093 84.2% | 0.090 82.5% | 0.148 88.9% | 0.146 88.9% | 0.130 88.9% |
| MLP | 1.265 | 2% | 0.104 54.4% | | | 0.137 66.1% | | | 0.203 82.5% | | |
| | | 5% | 0.085 51.5% | 0.084 50.9% | | 0.116 62.6% | 0.112 61.4% | | 0.186 80.1% | 0.186 79.5% | |
| | | 10% | 0.092 50.9% | 0.089 50.3% | 0.086 50.3% | 0.132 65.5% | 0.131 65.5% | 0.121 63.7% | 0.195 81.9% | 0.195 81.9% | 0.181 79.5% |
| LSTM | 1.536 | 2% | 0.074 45.6% | | | 0.120 59.6% | | | 0.178 77.8% | | |
| | | 5% | 0.077 45.6% | 0.076 45.6% | | 0.122 59.6% | 0.120 59.6% | | 0.176 77.2% | 0.172 77.2% | |
| | | 10% | 0.076 45.6% | 0.076 45.6% | 0.074 45.6% | 0.123 59.6% | 0.123 59.6% | 0.121 59.6% | 0.179 77.2% | 0.176 76.6% | 0.173 76.6% |
| GRU | 1.407 | 2% | 0.114 54.4% | | | 0.143 67.3% | | | 0.195 80.1% | | |
| | | 5% | 0.110 54.4% | 0.112 52.0% | | 0.141 67.3% | 0.145 66.7% | | 0.190 78.9% | 0.191 78.9% | |
| | | 10% | 0.109 54.4% | 0.109 54.4% | 0.110 54.4% | 0.140 67.3% | 0.140 67.3% | 0.139 67.3% | 0.195 80.1% | 0.195 80.1% | 0.190 79.5% |
| RNN | 1.727 | 2% | 0.111 38.6% | | | 0.165 55.0% | | | 0.201 63.7% | | |
| | | 5% | 0.114 38.0% | 0.106 38.0% | | 0.161 52.6% | 0.154 52.6% | | 0.203 63.2% | 0.194 63.2% | |
| | | 10% | 0.113 38.0% | 0.113 38.0% | 0.110 38.0% | 0.160 52.6% | 0.158 52.0% | 0.155 52.0% | 0.202 63.2% | 0.198 62.6% | 0.194 62.6% |
| TransformerNormal | 1.126 | 2% | 0.086 58.5% | | | 0.126 70.2% | | | 0.197 82.5% | | |
| | | 5% | 0.087 55.0% | 0.089 55.0% | | 0.122 67.8% | 0.118 66.7% | | 0.171 79.5% | 0.163 78.4% | |
| | | 10% | 0.084 56.7% | 0.078 55.6% | 0.078 54.4% | 0.116 67.8% | 0.114 67.8% | 0.110 66.1% | 0.165 80.1% | 0.159 79.5% | 0.157 79.5% |
| TransformerPadding | 1.236 | 2% | 0.115 54.4% | | | 0.144 69.0% | | | 0.187 84.2% | | |
| | | 5% | 0.122 52.6% | 0.121 50.3% | | 0.143 66.1% | 0.142 64.3% | | 0.183 81.3% | 0.188 81.3% | |
| | | 10% | 0.119 50.9% | 0.121 49.7% | 0.120 48.0% | 0.153 67.3% | 0.157 66.7% | 0.160 65.5% | 0.191 83.6% | 0.190 82.5% | 0.194 81.9% |
| TransformerConv | 1.175 | 2% | 0.105 66.1% | | | 0.144 73.7% | | | 0.219 80.1% | | |
| | | 5% | 0.099 63.2% | 0.103 60.8% | | 0.146 74.9% | 0.143 72.5% | | 0.210 79.5% | 0.202 78.9% | |
| | | 10% | 0.095 63.2% | 0.097 61.4% | 0.095 56.7% | 0.137 70.8% | 0.137 70.2% | 0.137 67.8% | 0.215 78.9% | 0.210 78.9% | 0.206 77.8% |

Table 28: (Weather) Evaluate MAE of MIA with different pretrained models on multi-variate time series forecasting (MTSF).

| Model | Baseline | $L_{\text{mask}}$ | Δ=1.0 2% | 5% | 10% | Δ=1.2 2% | 5% | 10% | Δ=1.5 2% | 5% | 10% |
|---|---|---|---|---|---|---|---|---|---|---|---|
| MLP-Mixer | 0.793 | 2% | 0.362 71.1% | | | 0.440 78.0% | | | 0.566 84.6% | | |
| | | 5% | 0.367 71.5% | 0.359 68.7% | | 0.445 78.0% | 0.436 76.5% | | 0.571 84.5% | 0.560 83.3% | |
| | | 10% | 0.362 70.2% | 0.361 70.0% | 0.347 66.1% | 0.439 77.3% | 0.438 77.3% | 0.425 74.5% | 0.563 84.0% | 0.561 83.9% | 0.545 82.0% |
| MLP | 0.831 | 2% | 0.405 70.0% | | | 0.476 78.1% | | | 0.586 85.9% | | |
| | | 5% | 0.406 71.9% | 0.396 66.6% | | 0.478 79.6% | 0.468 75.1% | | 0.587 86.6% | 0.578 84.1% | |
| | | 10% | 0.405 71.3% | 0.402 70.4% | 0.389 63.0% | 0.478 79.8% | 0.475 79.2% | 0.461 73.3% | 0.586 86.2% | 0.584 86.2% | 0.572 84.1% |
| LSTM | 0.898 | 2% | 0.432 72.3% | | | 0.516 77.2% | | | 0.645 83.6% | | |
| | | 5% | 0.432 72.3% | 0.431 71.8% | | 0.517 77.2% | 0.516 76.7% | | 0.646 83.5% | 0.644 83.0% | |
| | | 10% | 0.432 72.6% | 0.432 72.3% | 0.428 71.4% | 0.516 77.0% | 0.516 77.1% | 0.511 76.5% | 0.646 83.6% | 0.646 83.6% | 0.640 83.2% |
| GRU | 1.135 | 2% | 0.417 51.7% | | | 0.514 63.3% | | | 0.654 73.8% | | |
| | | 5% | 0.419 52.3% | 0.414 49.8% | | 0.516 63.8% | 0.512 61.8% | | 0.655 74.3% | 0.652 72.8% | |
| | | 10% | 0.419 52.4% | 0.417 52.1% | 0.410 49.4% | 0.515 63.7% | 0.513 63.3% | 0.506 59.6% | 0.654 74.1% | 0.652 73.7% | 0.646 70.9% |
| RNN | 0.777 | 2% | 0.385 73.1% | | | 0.465 77.6% | | | 0.587 82.8% | | |
| | | 5% | 0.388 74.8% | 0.367 71.2% | | 0.471 79.3% | 0.447 75.3% | | 0.593 83.7% | 0.568 79.6% | |
| | | 10% | 0.391 74.7% | 0.389 73.6% | 0.363 65.4% | 0.475 79.6% | 0.473 78.7% | 0.446 72.0% | 0.595 83.1% | 0.595 83.1% | 0.567 77.7% |
| TransformerNormal | 0.891 | 2% | 0.412 68.4% | | | 0.491 73.1% | | | 0.619 80.0% | | |
| | | 5% | 0.412 68.2% | 0.409 66.9% | | 0.491 73.1% | 0.489 72.6% | | 0.616 79.2% | 0.614 78.6% | |
| | | 10% | 0.406 66.6% | 0.407 66.4% | 0.399 65.2% | 0.485 72.8% | 0.486 72.5% | 0.479 71.9% | 0.609 78.8% | 0.609 78.5% | 0.601 77.7% |
| TransformerPadding | 0.844 | 2% | 0.392 70.1% | | | 0.480 76.9% | | | 0.611 83.5% | | |
| | | 5% | 0.392 70.0% | 0.387 69.3% | | 0.479 76.5% | 0.474 76.1% | | 0.611 83.4% | 0.606 83.2% | |
| | | 10% | 0.387 69.5% | 0.385 69.0% | 0.382 68.2% | 0.474 76.3% | 0.473 76.0% | 0.469 75.5% | 0.604 82.8% | 0.603 82.7% | 0.600 82.6% |
| TransformerConv | 0.886 | 2% | 0.394 66.9% | | | 0.475 74.5% | | | 0.598 82.3% | | |
| | | 5% | 0.393 66.6% | 0.391 66.1% | | 0.475 74.3% | 0.472 73.7% | | 0.598 82.2% | 0.596 81.6% | |
| | | 10% | 0.389 65.6% | 0.388 65.5% | 0.384 63.9% | 0.468 72.4% | 0.466 72.0% | 0.459 70.5% | 0.592 81.4% | 0.592 81.1% | 0.582 79.9% |

Table 29: (Weather) Evaluate MSE of MIA with different pretrained models on multi-variate time series forecasting (MTSF).

| Model | Baseline | $L_{\text{mask}}$ | Δ = 1.0 2% | 5% | 10% | Δ = 1.2 2% | 5% | 10% | Δ = 1.5 2% | 5% | 10% |
|---|---|---|---|---|---|---|---|---|---|---|---|
| MLP-Mixer | 1.227 | 2% | 0.159 71.1% | | | 0.235 **78.0%** | | | 0.378 **84.6%** | | |
| | | 5% | 0.162 **71.5%** | **0.157** 68.7% | | 0.238 78.0% | 0.232 76.5% | | 0.383 84.5% | **0.373** 83.3% | |
| | | 10% | **0.159** 70.2% | 0.158 **70.0%** | **0.149** 66.1% | **0.234** 77.3% | 0.233 **77.3%** | 0.222 74.5% | **0.376** 84.0% | 0.375 **83.9%** | **0.359** 82.0% |
| MLP | 1.485 | 2% | **0.187** 70.0% | | | **0.262** 78.1% | | | **0.403** 85.9% | | |
| | | 5% | 0.187 **71.9%** | **0.180** 66.6% | | 0.263 79.6% | **0.255** 75.1% | | 0.405 **86.6%** | **0.395** 84.1% | |
| | | 10% | 0.187 71.3% | 0.184 **70.4%** | **0.176** 63.0% | 0.263 **79.8%** | 0.261 **79.2%** | **0.250** 73.3% | 0.404 86.2% | 0.402 **86.2%** | **0.390** 84.1% |
| LSTM | 1.900 | 2% | **0.204** 72.3% | | | **0.293** 77.2% | | | **0.455** 83.6% | | |
| | | 5% | 0.204 72.3% | **0.204** 71.8% | | 0.293 **77.2%** | 0.292 76.7% | | 0.456 83.5% | **0.455** 83.0% | |
| | | 10% | 0.205 72.6% | 0.204 **72.3%** | **0.202** 71.4% | 0.293 77.0% | 0.293 **77.1%** | **0.289** 76.5% | 0.456 83.6% | 0.456 **83.6%** | **0.451** 83.2% |
| GRU | 2.167 | 2% | **0.194** 51.7% | | | **0.290** 63.3% | | | **0.465** 73.8% | | |
| | | 5% | 0.195 52.3% | **0.192** 49.8% | | 0.292 **63.8%** | 0.289 61.8% | | 0.467 **74.3%** | 0.464 72.8% | |
| | | 10% | 0.195 **52.4%** | 0.194 **52.1%** | **0.189** 49.4% | 0.291 63.7% | 0.290 **63.3%** | **0.283** 59.6% | 0.466 74.1% | 0.464 **73.7%** | **0.457** 70.9% |
| RNN | 1.591 | 2% | **0.171** 73.1% | | | **0.249** 77.6% | | | **0.392** 82.8% | | |
| | | 5% | 0.173 **74.8%** | **0.160** 71.2% | | 0.252 79.3% | **0.234** 75.3% | | 0.397 **83.7%** | **0.371** 79.6% | |
| | | 10% | 0.176 74.7% | 0.174 **73.6%** | **0.158** 65.4% | 0.256 79.6% | 0.253 **78.7%** | **0.234** 72.0% | 0.402 **83.7%** | 0.399 **83.1%** | **0.371** 77.7% |
| TransformerNormal | 1.457 | 2% | 0.190 **68.4%** | | | 0.272 **73.1%** | | | 0.432 **80.0%** | | |
| | | 5% | 0.190 68.2% | 0.189 **66.9%** | | 0.272 **73.1%** | 0.271 **72.6%** | | 0.430 79.2% | 0.427 **78.6%** | |
| | | 10% | **0.187** 66.6% | **0.187** 66.4% | **0.182** 65.2% | **0.268** 72.8% | **0.268** 72.5% | **0.262** 71.9% | **0.423** 78.8% | **0.422** 78.5% | **0.414** 77.7% |
| TransformerPadding | 1.502 | 2% | 0.178 **70.1%** | | | 0.263 76.9% | | | 0.418 **83.5%** | | |
| | | 5% | 0.178 70.0% | 0.175 **69.3%** | | 0.262 76.5% | 0.258 **76.1%** | | 0.418 83.4% | 0.413 **83.2%** | |
| | | 10% | **0.175** 69.5% | **0.174** 69.0% | **0.171** 68.2% | **0.258** 76.3% | **0.257** 76.0% | **0.255** 75.5% | **0.411** 82.8% | **0.410** 82.7% | **0.407** 82.6% |
| TransformerConv | 1.599 | 2% | 0.179 **66.9%** | | | 0.261 **74.5%** | | | 0.410 82.3% | | |
| | | 5% | 0.179 66.6% | 0.178 **66.1%** | | 0.261 74.3% | 0.259 **73.7%** | | 0.410 82.2% | 0.407 **81.6%** | |
| | | 10% | **0.176** 65.6% | **0.175** 65.5% | **0.171** 63.9% | **0.256** 72.4% | **0.254** 72.0% | **0.248** 70.5% | **0.405** 81.4% | **0.404** 81.1% | **0.395** 79.9% |

Table 30: Compare $\ell_0$-norm localized perturbation to $\ell_2$-norm perturbation (computed by the algorithm [2]) on the MSE between the original forecast and the perturbed forecast. The table reports the relative improvement of the $\ell_0$-norm perturbation over the $\ell_2$-norm perturbation (averaging among 128 randomly selected samples). The positive value indicates that our $\ell_0$-norm perturbation outperforms $\ell_0$-norm perturbation. For fairness, the $\ell_0$ or $\ell_2$ norm of the perturbation is restricted to be no larger than $\beta\times$ the average value among the $\ell_0$ or $\ell_2$ norm of all the testing samples. Values in tables are calculated as $(\text{MSE}_{\ell_0} - \text{MSE}_{\ell_2})/\text{MSE}_{\ell_2}$.

| Model | Attack Rate ($\beta$) 10% | 20% | 30% | 40% | 50% |
|---|---|---|---|---|---|
| MLP-Mixer | +769.9% | +89.5% | +73.3% | +12.3% | +53.1% |
| GRU | +2.5% | -1.6% | -8.3% | -3.3% | -6.5% |
| LSTM | +23.1% | +15.1% | -33.5% | -14.8% | +2.0% |
| MLP | +265.0% | +376.3% | +211.6% | +114.1% | +58.3% |

Table 31: (Exchange) Time series attack. Difference of MSE between $\ell_0$ and $\ell_2$ attack.

| Model | Attack Rate ($\beta$) 10% | 20% | 30% | 40% | 50% |
|---|---|---|---|---|---|
| MLP-Mixer | +1000.1% | +332.2% | +114.7% | +131.7% | +84.6% |
| GRU | -48.6% | -45.1% | -39.7% | -31.7% | -20.9% |
| LSTM | -8.8% | -28.3% | -24.9% | -16.2% | -6.6% |
| MLP | +528.9% | +101.8% | +36.9% | +8.8% | +6.2% |

Table 32: (Traffic) Time series attack. Difference of MSE between $\ell_0$ and $\ell_2$ attack.

| Model | Attack Rate ($\beta$) 10% | 20% | 30% | 40% | 50% |
|---|---|---|---|---|---|
| MLP-Mixer | +3310.1% | +841.8% | +328.5% | +317.7% | +205.5% |
| GRU | +147.0% | +28.2% | +7.3% | -4.7% | +21.6% |
| LSTM | +239.0% | +66.8% | +14.9% | +2.5% | -2.8% |
| MLP | +1760.4% | +145.7% | +38.4% | +7.5% | -7.6% |

Table 33: (DistalPhalanxTW $L_{mask} = 15\%$) $\sigma = 0.01$

| Model | Metric, $\sigma = 0.010$ | $L_{\text{mask}}$ | DistalPhalanxTW | | | MiddlePhalanxTW | | | ProximalPhalanxTW | | |
|---|---|---|---|---|---|---|---|---|---|---|---|
| | | | 5 % | 10 % | 15 % | 5 % | 10 % | 15 % | 5 % | 10 % | 15 % |
| FCN | ACC | 5 % | 1.0 % | | | -3.0 % | | | 0.0 % | | |
| | | 10 % | -1.0 % | 0.0 % | | 0.0 % | -1.0 % | | -1.0 % | -3.0 % | |
| | | 15 % | 1.0 % | 3.0 % | 0.0 % | -4.0 % | -3.0 % | -4.0 % | -1.0 % | 1.0 % | 3.0 % |
| MLP-Mixer | ACC | 5 % | 0.0 % | | | 0.0 % | | | 1.0 % | | |
| | | 10 % | -2.0 % | -2.0 % | | 0.0 % | -1.0 % | | -1.0 % | -1.0 % | |
| | | 15 % | -2.0 % | 0.0 % | -2.0 % | -1.0 % | 1.0 % | -2.0 % | 1.0 % | 0.0 % | 2.0 % |
| MLP | ACC | 5 % | -1.0 % | | | 2.0 % | | | 0.0 % | | |
| | | 10 % | -1.0 % | -1.0 % | | -1.0 % | 0.0 % | | 0.0 % | 0.0 % | |
| | | 15 % | 1.0 % | 1.0 % | 3.0 % | 2.0 % | 4.0 % | 2.0 % | 1.0 % | 0.0 % | 0.0 % |
| ResNet-18 | ACC | 5 % | -2.0 % | | | -1.0 % | | | 0.0 % | | |
| | | 10 % | 3.0 % | 0.0 % | | 0.0 % | 0.0 % | | 0.0 % | 1.0 % | |
| | | 15 % | -1.0 % | -3.0 % | -1.0 % | 0.0 % | 1.0 % | 1.0 % | -1.0 % | -2.0 % | -1.0 % |

Table 34: (DistalPhalanxTW $L_{mask} = 15\%$) $\sigma = 0.02$

| Model | Metric, $\sigma = 0.020$ | $L_{\text{mask}}$ | DistalPhalanxTW | | | MiddlePhalanxTW | | | ProximalPhalanxTW | | |
|---|---|---|---|---|---|---|---|---|---|---|---|
| | | | 5 % | 10 % | 15 % | 5 % | 10 % | 15 % | 5 % | 10 % | 15 % |
| FCN | ACC | 5 % | -1.0 % | | | 0.0 % | | | 0.0 % | | |
| | | 10 % | 0.0 % | 1.0 % | | 0.0 % | 0.0 % | | 0.0 % | 0.0 % | |
| | | 15 % | 5.0 % | 6.9 % | 3.0 % | 0.0 % | -1.0 % | -2.0 % | 0.0 % | 1.0 % | 2.0 % |
| MLP-Mixer | ACC | 5 % | 0.0 % | | | 1.0 % | | | 1.0 % | | |
| | | 10 % | 1.0 % | 0.0 % | | 0.0 % | 2.0 % | | 1.0 % | -1.0 % | |
| | | 15 % | -1.0 % | 3.0 % | 5.0 % | 2.0 % | 0.0 % | 0.0 % | 0.0 % | 0.0 % | 0.0 % |
| MLP | ACC | 5 % | 0.0 % | | | 2.0 % | | | 0.0 % | | |
| | | 10 % | 3.0 % | 0.0 % | | 1.0 % | 2.0 % | | 0.0 % | 1.0 % | |
| | | 15 % | 0.0 % | 2.0 % | 3.0 % | 2.0 % | 3.0 % | 3.0 % | 0.0 % | 0.0 % | 0.0 % |
| ResNet-18 | ACC | 5 % | -3.0 % | | | 0.0 % | | | 0.0 % | | |
| | | 10 % | 5.0 % | 1.0 % | | -1.0 % | 1.0 % | | 0.0 % | 0.0 % | |
| | | 15 % | 0.0 % | 1.0 % | 1.0 % | 1.0 % | 2.0 % | 2.0 % | 0.0 % | -1.0 % | -1.0 % |

Table 35: (DistalPhalanxTW $L_{mask} = 15\%$) $\sigma = 0.03$

| Model | Metric, $\sigma = 0.030$ | $L_{\text{mask}}$ | DistalPhalanxTW | | | MiddlePhalanxTW | | | ProximalPhalanxTW | | |
|---|---|---|---|---|---|---|---|---|---|---|---|
| | | | 5 % | 10 % | 15 % | 5 % | 10 % | 15 % | 5 % | 10 % | 15 % |
| FCN | ACC | 5 % | -2.0 % | | | 0.0 % | | | 1.0 % | | |
| | | 10 % | -2.0 % | 0.0 % | | -1.0 % | -3.0 % | | -3.0 % | -2.0 % | |
| | | 15 % | 2.0 % | 3.0 % | 2.0 % | -2.0 % | -3.0 % | -2.0 % | -1.0 % | 2.0 % | 2.0 % |
| MLP-Mixer | ACC | 5 % | 0.0 % | | | 1.0 % | | | 2.0 % | | |
| | | 10 % | 0.0 % | 1.0 % | | -1.0 % | 1.0 % | | 1.0 % | 0.0 % | |
| | | 15 % | -1.0 % | 3.0 % | 3.0 % | 1.0 % | 0.0 % | 0.0 % | 1.0 % | 1.0 % | 0.0 % |
| MLP | ACC | 5 % | 0.0 % | | | 2.0 % | | | 0.0 % | | |
| | | 10 % | 1.0 % | 1.0 % | | 0.0 % | 1.0 % | | 0.0 % | 1.0 % | |
| | | 15 % | 1.0 % | 3.0 % | 4.0 % | 0.0 % | 1.0 % | 1.0 % | -1.0 % | 0.0 % | -1.0 % |
| ResNet-18 | ACC | 5 % | -4.0 % | | | 2.0 % | | | 0.0 % | | |
| | | 10 % | 0.0 % | -1.0 % | | 0.0 % | 1.0 % | | 0.0 % | 1.0 % | |
| | | 15 % | -1.0 % | -1.0 % | 0.0 % | 0.0 % | 1.0 % | 1.0 % | 0.0 % | 0.0 % | 0.0 % |

Table 36: (DistalPhalanxTW $L_{mask} = 15\%$) $\sigma = 0.04$

| Model | Metric, $\sigma = 0.040$ | $L_{\text{mask}}$ | DistalPhalanxTW | | | MiddlePhalanxTW | | | ProximalPhalanxTW | | |
|---|---|---|---|---|---|---|---|---|---|---|---|
| | | | 5 % | 10 % | 15 % | 5 % | 10 % | 15 % | 5 % | 10 % | 15 % |
| FCN | ACC | 5 % | 2.0 % | | | -1.0 % | | | 0.0 % | | |
| | | 10 % | 2.0 % | 3.0 % | | -2.0 % | -2.0 % | | -2.0 % | -2.0 % | |
| | | 15 % | 2.0 % | 1.0 % | 0.0 % | -1.0 % | -2.0 % | -2.0 % | -3.0 % | 1.0 % | 1.0 % |
| MLP-Mixer | ACC | 5 % | -1.0 % | | | 1.0 % | | | 0.0 % | | |
| | | 10 % | 0.0 % | 1.0 % | | 0.0 % | -1.0 % | | 0.0 % | 0.0 % | |
| | | 15 % | 0.0 % | 2.0 % | 0.0 % | 0.0 % | 1.0 % | -2.0 % | -1.0 % | 2.0 % | 0.0 % |
| MLP | ACC | 5 % | 1.0 % | | | 1.0 % | | | 1.0 % | | |
| | | 10 % | 1.0 % | 0.0 % | | 0.0 % | 1.0 % | | 1.0 % | 1.0 % | |
| | | 15 % | 1.0 % | 3.0 % | 4.0 % | 1.0 % | 3.0 % | 3.0 % | -1.0 % | 0.0 % | 0.0 % |
| ResNet-18 | ACC | 5 % | -1.0 % | | | 1.0 % | | | -1.0 % | | |
| | | 10 % | 0.0 % | 0.0 % | | 0.0 % | 1.0 % | | 1.0 % | 1.0 % | |
| | | 15 % | 2.0 % | 2.0 % | 0.0 % | 1.0 % | 1.0 % | 2.0 % | 0.0 % | 0.0 % | -1.0 % |

Table 37: (DistalPhalanxTW $L_{mask} = 15\%$) $\sigma = 0.05$

| Model | Metric, $\sigma = 0.050$ | $L_{\text{mask}}$ | DistalPhalanxTW | | | MiddlePhalanxTW | | | ProximalPhalanxTW | | |
|---|---|---|---|---|---|---|---|---|---|---|---|
| | | | 5 % | 10 % | 15 % | 5 % | 10 % | 15 % | 5 % | 10 % | 15 % |
| FCN | ACC | 5 % | 1.0 % | | | 0.0 % | | | 1.0 % | | |
| | | 10 % | 0.0 % | 1.0 % | | -1.0 % | -4.0 % | | -1.0 % | 0.0 % | |
| | | 15 % | 5.0 % | 4.0 % | 2.0 % | 0.0 % | -1.0 % | -2.0 % | 0.0 % | 2.0 % | 1.0 % |
| MLP-Mixer | ACC | 5 % | -1.0 % | | | -2.0 % | | | 1.0 % | | |
| | | 10 % | 1.0 % | 1.0 % | | -2.0 % | -1.0 % | | -1.0 % | -2.0 % | |
| | | 15 % | 2.0 % | 5.0 % | 0.0 % | 3.0 % | 1.0 % | 0.0 % | 0.0 % | -1.0 % | 2.0 % |
| MLP | ACC | 5 % | 1.0 % | | | 2.0 % | | | 0.0 % | | |
| | | 10 % | 4.0 % | 1.0 % | | -1.0 % | 1.0 % | | 0.0 % | 0.0 % | |
| | | 15 % | 2.0 % | 3.0 % | 3.0 % | 2.0 % | 4.0 % | 1.0 % | -1.0 % | 0.0 % | 0.0 % |
| ResNet-18 | ACC | 5 % | -2.0 % | | | 0.0 % | | | -1.0 % | | |
| | | 10 % | 3.0 % | -1.0 % | | -1.0 % | -1.0 % | | 0.0 % | -1.0 % | |
| | | 15 % | 1.0 % | 1.0 % | 0.0 % | 1.0 % | 1.0 % | 1.0 % | -1.0 % | 0.0 % | -1.0 % |

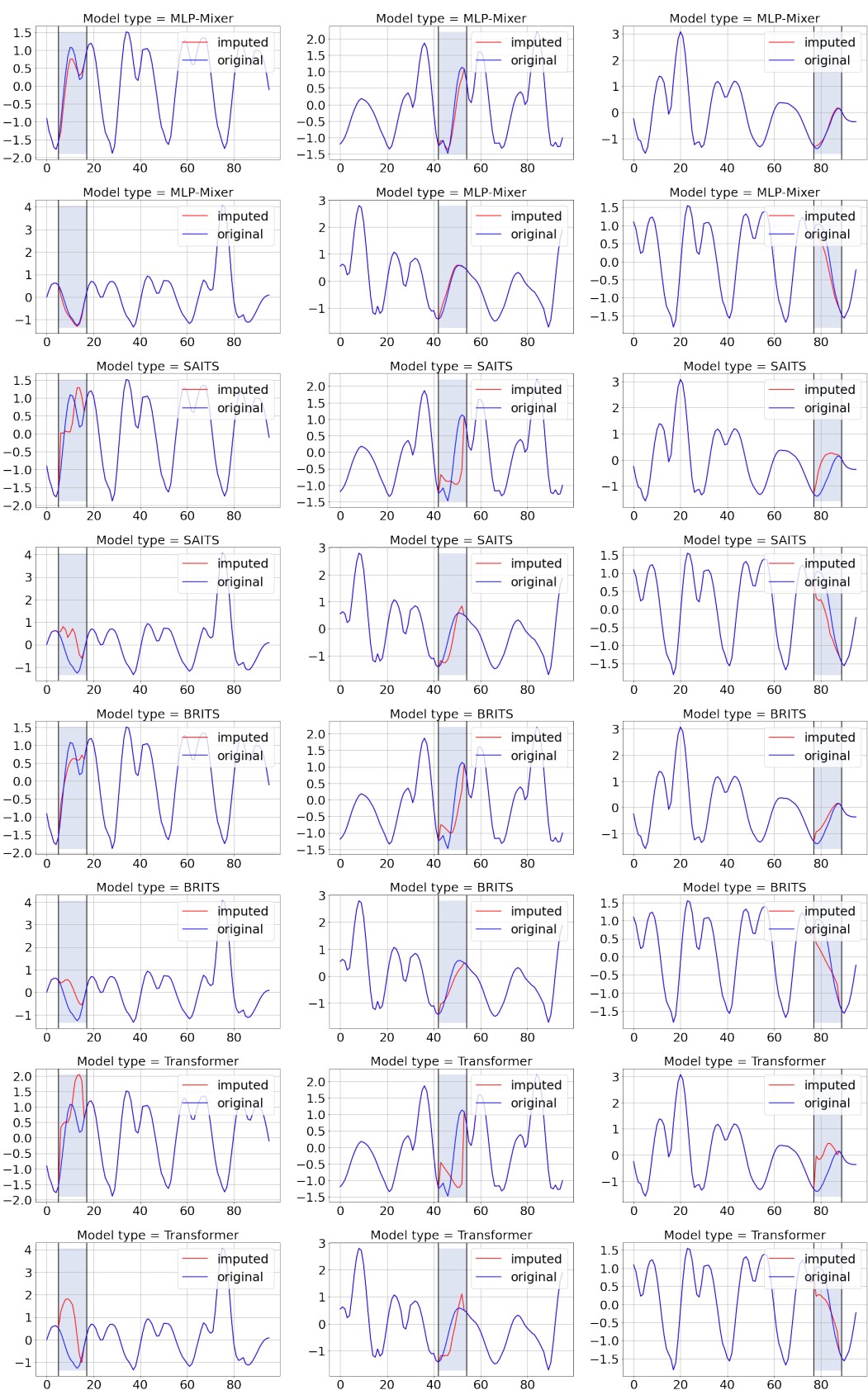

Figure 4: (Traffic) Imputation quality with different imputation models. We set $L_{\mathrm{mask}} = 10\%$. Original and imputed refer to the original time series and the imputed time series respectively.

