# OpenReview forum: "MIA: A Framework for Certified Robustness of Time-Series Classification and Forecasting Against Temporally-Localized Perturbations"
_ICLR.cc/2023/Conference — Submitted to ICLR 2023_

### Official Review · Reviewer_SiPr · 2022-10-21

**Confidence:** 3
**Correctness:** 3
**Technical Novelty And Significance:** 2
**Empirical Novelty And Significance:** 4
**Recommendation:** 6

**Clarity, Quality, Novelty And Reproducibility:**

The writing is generally clear but not of high quality IMHO. The novelty may be a bit limited from the technical perspective but should be sufficient from the application perspective. The reproducibility is great given the open-source code.

**Strength And Weaknesses:**

Strengths:
- A practical and rigorous certification framework with significantly better performance than baselines.
- The framework is novel since no prior work achieves certified time series prediction against $\ell_0$ predictions and no prior work can protect base models without retraining in this domain.

Weaknesses:
- The idea is relatively straightforward: it is like an application of [1,2] in the time series domain. The unique point of imputation model is spiritually similar to [3].

- The writing quality needs further improvements. For example, many typos exist even on the first page:
  1. First para: $ell_0$-robustness -> $\ell_0$-robustness
  2. Second para: "Athalye et al. (Athalye et al., 2018)" repeated - you can use `\citet{}`
  3. Second para: "the concept of certified defense is proposed. guaranteeing the robustness for each prediction mathematically." -> additional "."

In later sections:

4. Eqn. (1): the first line $x_{1:t_0} + \delta, \text{subject to} \|\delta\|_0 \le L\_{adv}$ does not contain the temporal localization constraint, so it is not equal to the second line.

5. Eqn. (3): $G(\mathbf{x}^{(m)}\_{1:t_0})$ -> $G(\mathbf{\hat x}^{(m)}\_{1:t_0})$

6. Remark 1, last sentence: the inference cost could not be 0. It should at least be 1.

7. What is the $\sigma$ value used in practice in Eqn. (7)? It is not specified in later sections. Also, why Gaussian augmentation is needed here? Unlike other randomized smoothing work, here no Gaussian augmentation is used in the inference time, so the use of Gaussian augmentation in the training time may induce domain mismatch and needs further justifications.

8. Eqn. (9): I think the guarantee of RHS should be $2 * (k+L_{adv}+1)$ instead of $k + L_{adv}+1$, since both terms of LHS can change by 1 for each step perturbation leading to gap reduced by 2.

9. Section 5, first paragraph: MiddlePhalanxTW ProximalPhalanx -> MiddlePhalanxTW, ProximalPhalanxTW

[1] PatchGuard: A Provably Robust Defense against Adversarial Patches via Small Receptive Fields and Masking. USENIX Security 21

[2] PatchCleanser: Certifiably robust defense against adversarial patches for any image classifier. In 31st USENIX Security Symposium (USENIX Security 22)

[3] Denoised Smoothing: A Provable Defense for Pretrained Classifiers. NeurIPS 2020

**Summary Of The Paper:**

This paper proposes Masking Imputing Aggregation (MIA), a framework based on masking and aggregation, to achieve certified robustness for time series forecasting models against temporally localized $\ell_0$-norm-bounded perturbations. For the masked time steps, an imputation model is trained to fill in and then new time series is fed into the old model for prediction. When consensus is achieved, certified robustness of the pipeline is achieved. Experiments on Exchange Rate, Traffic, UCI Electricity, DistalPhalanxTW, MiddlePhalanxTW, ProximalPhalanxTW against two baselines extended from previous work (Derandomized Smoothing and Randomized Ablation) demonstrate the effectiveness.

**Summary Of The Review:**

Due to the limited technical novelty and rough writing, the submission may be not ready for publication at the current stage. However, I appreciate the novel application of masking and imputation framework in the certified robustness of the time series domain.

---

> ### Author Response · Authors · 2022-11-12
> **Official Response to Reviewer SiPr (Part 2)**
>
> > 5. Eq. (9): I think the guarantee of RHS should be $2*(k+L_{adv}+1)$ instead of $(k+L_{adv}+1)$.
>
> We sincerely apologize for this mistake, but we point out that this mistake will not affect the superiority of MIA in our empirical comparisons. Changing $(k+L_{adv}+1)$ to $2*(k+L_{adv}+1)$ will enlarge the gap between MIA and DS. We have revised the results in Table 1, Table 2 and Table 3. Below we only provide a part of updated results due to space limitation:
>
> Table 3. (Exchange) Compare MIA to DS and RA on **FR (%)** at $L_{atk}= 2\%, 4\%, 6\%, 8\%, 10\%$ (higher FR is better).
> | Defense $\downarrow \quad L_{atk} \rightarrow$ &nbsp;&nbsp;&nbsp;&nbsp;  | 2% | 4% | 6% | 8% | 10% |
> |--------------------|------|------|------|------|------|
> | DS ($k= 10\%$) | ~~24.8~~ 24.8 | ~~24.8~~ 24.8| ~~24.8~~ 23.8| ~~24.8~~ 23.8 | ~~24.8~~ 22.8|
> | RA ($k= 10\%$) | 16.8 | 16.8 | 16.8 | 16.8 | 16.8 |
> | MIA ($L_{mask}= 10\%$) | **82.2** | **82.2** | **81.2** | **81.2** | **79.2** |
> | DS ($k= 15\%$) | ~~20.8~~ 9.9 | ~~20.8~~ 8.9| ~~20.8~~ 8.9| ~~20.8~~ 2.0 | ~~20.8~~ 1.0|
> | RA ($k= 15\%$) | 23.8 | 23.8 | 23.8 | 23.8 | 23.8 |
> | MIA ($L_{mask}= 15\%$) | **71.3** | **69.3** | **68.3** | **71.3** | **65.3** |
>
> Please refer to our paper for more details. Sorry again for the mistake.
>
> **References**
>
> [1] Xiang, Chong, et al. "{PatchGuard}: A Provably Robust Defense against Adversarial Patches via Small Receptive Fields and Masking." 30th USENIX Security Symposium (USENIX Security 21). 2021.
>
> [2] Xiang, Chong, Saeed Mahloujifar, and Prateek Mittal. "{PatchCleanser}: Certifiably Robust Defense against Adversarial Patches for Any Image Classifier." 31st USENIX Security Symposium (USENIX Security 22). 2022.
>
> [3] Salman, Hadi, et al. "Denoised smoothing: A provable defense for pretrained classifiers." Advances in Neural Information Processing Systems 33 (2020): 21945-21957.
>
> [4] Foster, W. R., F. Collopy, and L. H. Ungar. "Neural network forecasting of short, noisy time series." Computers & chemical engineering 16.4 (1992): 293-297.
>
> [5] Passalis, Nikolaos, et al. "Training noise-resilient recurrent photonic networks for financial time series analysis." 2020 28th European Signal Processing Conference (EUSIPCO). IEEE, 2021.
>
> [6] Hwang, Jeng-Ren, Shyi-Ming Chen, and Chia-Hoang Lee. "Handling forecasting problems using fuzzy time series." Fuzzy sets and systems 100.1-3 (1998): 217-228.
>
> [7] Magdon-Ismail, Malik, Alexander Nicholson, and Yaser S. Abu-Mostafa. "Financial markets: very noisy information processing." Proceedings of the IEEE 86.11 (1998): 2184-2195.
>
> [8] Salinas, David, et al. "DeepAR: Probabilistic forecasting with autoregressive recurrent networks." International Journal of Forecasting 36.3 (2020): 1181-1191.

---

> > ### Comment · Reviewer_SiPr · 2022-11-18
> > **Thanks for the detailed response**
> >
> > Thank the authors for the detailed response! The revision and clarifications improved the presentation quality and addressed my concerns. Hence, I decided to raise the rating from 5 to 6.
> > On the other hand, I feel the technical novelty and contributions of this submission may be still a bit below the bar, though the practical implications are significant. I am not confident in deciding whether to support acceptance in this case. Hence, I lower the confidence rating from 4 to 3.

---

> ### Author Response · Authors · 2022-11-12
> **Official Response to Reviewer SiPr (Part 1)**
>
> We thank the reviewer for the thoughtful comments.  our ICLR submission is an updated version which has addressed several concerns from the reviews such as the typos and Gaussian augmentation. Next we provide detailed answers to the questions.
>
> > 1. The idea is relatively straightforward: it is like an application of [1,2] in the time series domain. The unique point of imputation model is spiritually similar to [3].
>
> We are glad that the reviewer acknowledge the novelty of this paper. We admit MIA is a relatively direct extension of [1,2] and imputation model is spiritually similar to [3]. However, our contribution is non-trivial for:
> + This is the first work that explores the $\ell_0$ norm certificate for both TSF and TSC models.
> + MIA also produces robustness certificates for the forecasting models, not just for classification models.
> + We propose masked training algorithm for training the imputation model, which takes into consideration the sliding-window masking nature of MIA masked series.
> + We comprehensively compare MIA to two randomized smoothing based approaches on certified accuracy/MSE/MAE.
>
>
> > 2. Eq. (1): the first line does not contain the temporal localization constraint, so it is not equal to the second line.
>
> Sorry for the typos in Eq. (1). We have updated Eq. (1) in our revision. Please see our paper and let us know if there is any additional unclarity.
>
>
>
>
> > 3. Remark 1, last sentence: the inference cost could not be 0. It should at least be 1.
>
> We thank the reviewer for pointing this out. We admit that the inference cost should be at least $1$ and we have updated it in our revision.
>
>
>
>
> > 4. What is the $\sigma$ value used in practice in Eq. (7)? It is not specified in later sections. Also, why Gaussian augmentation is needed here? Unlike other randomized smoothing work, here no Gaussian augmentation is used in the inference time, so the use of Gaussian augmentation in the training time may induce domain mismatch and needs further justifications.
>
> We thank the reviewer for pointing this out.
> + We clarify that we use $\sigma=0.02$ and Gaussian augmentation in our masked training is used to **deal with noisy samples and avoid overfitting**, as extensive literature validates the presence of random noise in time series data [4,5,6], e.g., [7] demonstrates that financial data are noisy. DeepAR [8] explicitly models such random noise with Gaussian distribution.
> + **We do not add Gaussian noise to input series in inference stage**.
> + We additionally provide the experiments to show the impact of Gaussian augmentation. We hope this clarifies the reviewer’s concern.
>
> Table 1. (DistalPhalanxTW) We compare MIA on time series classification dataset DistalPhalanxTW ($L_{mask}=$ 15%) with Gaussian augmentation ($\sigma=$ 0.01, 0.02, 0.03) to MIA without Gaussian augmentation (baseline) on certified accuracy at $L_{atk} =$ 5%, 10%, 15%. The table reports the relative improvement (abs $\%$) on the certified accuracy (positive value means Gaussian augmentation improves the certified accuracy).
> |  Model $\downarrow$ &nbsp;&nbsp;&nbsp;&nbsp;&nbsp;&nbsp;&nbsp;&nbsp;&nbsp;&nbsp;&nbsp;&nbsp;&nbsp;&nbsp;| $\sigma \downarrow$ &nbsp;&nbsp;|  5 % &nbsp;&nbsp;&nbsp;&nbsp;&nbsp;&nbsp;&nbsp;&nbsp;&nbsp;&nbsp;&nbsp;&nbsp;| 10% &nbsp;&nbsp;&nbsp;&nbsp;&nbsp;&nbsp;&nbsp;&nbsp;&nbsp;&nbsp;&nbsp;&nbsp;| 15% &nbsp;&nbsp;&nbsp;&nbsp;&nbsp;&nbsp;&nbsp;&nbsp;&nbsp;&nbsp;&nbsp;&nbsp;| $\sigma \downarrow$ &nbsp;&nbsp;|  5 % &nbsp;&nbsp;&nbsp;&nbsp;&nbsp;&nbsp;&nbsp;&nbsp;&nbsp;&nbsp;&nbsp;&nbsp;| 10% &nbsp;&nbsp;&nbsp;&nbsp;&nbsp;&nbsp;&nbsp;&nbsp;&nbsp;&nbsp;&nbsp;&nbsp;| 15% &nbsp;&nbsp;&nbsp;&nbsp;&nbsp;&nbsp;&nbsp;&nbsp;&nbsp;&nbsp;&nbsp;&nbsp;|$\sigma \downarrow$ &nbsp;&nbsp;|  5 % &nbsp;&nbsp;&nbsp;&nbsp;&nbsp;&nbsp;&nbsp;&nbsp;&nbsp;&nbsp;&nbsp;&nbsp;| 10% &nbsp;&nbsp;&nbsp;&nbsp;&nbsp;&nbsp;&nbsp;&nbsp;&nbsp;&nbsp;&nbsp;&nbsp; | 15% &nbsp;&nbsp;&nbsp;&nbsp;&nbsp;&nbsp;&nbsp;&nbsp;&nbsp;&nbsp;&nbsp;&nbsp;|
> |:----:|:----:|:----:|:----:|:----:|:----:|:----:|:----:|:----:|:----:|:----:|:----:|:----:|
> | MLP-Mixer | 0.01 | - 2.0 |  0.0 | - 2.0 | 0.02 | - 1.0 | + 3.0 | + 5.0 | 0.03 | + 1.0 | + 3.0 | + 0.0 |
> | FCN | 0.01 | + 1.0 | + 3.0 |  0.0 | 0.02 | + 5.0 | + 6.9 | + 3.0 | 0.03 | + 1.0 | + 3.0 | + 0.0 |
> | MLP | 0.01 | + 1.0 | + 1.0 | + 3.0 | 0.02 |  0.0 | + 2.0 | + 3.0 | 0.03 | + 1.0 | + 3.0 | + 0.0 |
> |ResNet | 0.01 | - 1.0 | - 3.0 | - 1.0 | 0.02 | 0.0 | + 1.0 | + 1.0 | 0.03 | + 1.0 | + 3.0 | + 0.0 |

---

### Official Review · Reviewer_k4J1 · 2022-10-24

**Confidence:** 4
**Clarity, Quality, Novelty And Reproducibility:** See strength and weakness.
**Correctness:** 2
**Technical Novelty And Significance:** 2
**Empirical Novelty And Significance:** 2
**Recommendation:** 5

**Strength And Weaknesses:**

Strength
1. The paper studies a new problem and introduces a three-step approach for detecting $l_{0}$ norm perturbation in time series, which is easy to apply to different time series predictive models.
2. The paper provides some theoretical analysis about the robustness of the proposed method, and comparison with randomized smoothing defenses.
3. The experimental results demonstrate the effectiveness of the proposed method by comparing with a couple of baseline methods.

Weakness
1. The practical significance to detect $l_{0}$ norm perturbation in time series is obscure. The paper could provide some examples to illustrate why detect $l_{0}$ norm perturbation in time series is practically useful, compared to $l_{2}$ norm perturbation.
2. The presentation of the technical development is hard to follow. Many equations are given without sufficient description on their intuition. For example, Eq. 1, it is unclear why the norm $L_{adv}$ is used as the time point index and what is the relationship between $\delta$ and $\delta_{t}$. Eq. 4, it is unclear on why should all M predictions be consistent instead of allowing some tolerance. Some theoretical analysis is desired. Similarly, Eq 5, 7, etc., are presented without introducing intuition.
3. Below Eq 1, the authors claim in most cases short-term events should not have large impacts on the long-term outcomes. Then the question is why resistance to the short-term perturbation is essential.
4. The proposed method requires the masks to cover arbitrary temporally-localized perturbation of $L_{adv}$. It is better to provide some guidelines on how to ensure this requirement is satisfied.
5. In proposition 1, to conclude the forecast/label cannot be changed, it is better to quantify the change and define what is considered unchanged quantitatively.
6. In Remark 3, it is better to also describe the impact of certificate only for $f_{MIA}$ instead of $f$.
7. It is good to see the comparison to randomized smoothing defenses. It is better to further justify on why this comparison is important.
8. Is there any discussion on whether the proposed method useful for both univariate and multivariate time series?


**Summary Of The Paper:**

This paper presents a method for detecting $l_{0}$ norm perturbation in time series. The key idea is to do multiple masking over the time series by sliding windows, impute the masked areas individually, and measure the disagreement of the predictions using the imputed time series. A disagreement is an indicator of perturbation. The method is a flexible framework and can be applied to different base predictive models.

**Summary Of The Review:**

The paper presents a method for dealing with a new problem on detecting $l_{0}$ norm perturbation in time series. The method is a flexible framework to different forecasting/classification methods. The practical significance of the studied problem requires some justification. The presentation of the method design should be improved for clarity. The current paper is not easy to evaluate the validity of the proposed method. See details in strength and weakness.

---

> ### Author Response · Authors · 2022-11-12
> **Official Response to Reviewer k4J1 (Part 5)**
>
>
> **References**
>
> [1] Piotroski, Joseph D. "Value investing: The use of historical financial statement information to separate winners from losers." Journal of Accounting Research (2000): 1-41.
>
> [2] Dang-Nhu, Raphaël, et al. "Adversarial attacks on probabilistic autoregressive forecasting models." International Conference on Machine Learning. PMLR, 2020.
>
> [3] Yoon, TaeHo, et al. "Robust Probabilistic Time Series Forecasting." International Conference on Artificial Intelligence and Statistics. PMLR, 2022.
>
> [4] Cohen, Jeremy, Elan Rosenfeld, and Zico Kolter. "Certified adversarial robustness via randomized smoothing." International Conference on Machine Learning. PMLR, 2019.
>
> [5] Yang, Greg, et al. "Randomized smoothing of all shapes and sizes." International Conference on Machine Learning. PMLR, 2020.
>
> [6] Salman, Hadi, et al. "Provably robust deep learning via adversarially trained smoothed classifiers." Advances in Neural Information Processing Systems 32 (2019).
>
> [7] Li, Linyi, et al. "Tss: Transformation-specific smoothing for robustness certification." Proceedings of the 2021 ACM SIGSAC Conference on Computer and Communications Security. 2021.
>
> [8] Hao, Zhongkai, et al. "GSmooth: Certified robustness against semantic transformations via generalized randomized smoothing." International Conference on Machine Learning. PMLR, 2022.
>
> [9] Wu, Haixu, et al. "Autoformer: Decomposition transformers with auto-correlation for long-term series forecasting." Advances in Neural Information Processing Systems 34 (2021): 22419-22430.
>
> [10] Zhou, Haoyi, et al. "Informer: Beyond efficient transformer for long sequence time-series forecasting." Proceedings of the AAAI Conference on Artificial Intelligence. Vol. 35. No. 12. 2021.
>
> [11] https://www.bgc-jena.mpg.de/wetter/
>
> [12] https://gis.cdc.gov/grasp/fluview/fluportaldashboard.html

---

> ### Author Response · Authors · 2022-11-12
> **Official Response to Reviewer k4J1 (Part 4)**
>
>
> > 5. In proposition 1, to conclude the forecast/label cannot be changed, it is better to quantify the change and define what is considered unchanged quantitatively.
>
> We thank the reviewer for pointing this out.
> + Time series classification: the label is considered to have been changed if the MIA classifier **returns a different label** (not Abstain).
> + Time series forecasting: the forecast is considered to have been changed if there is **any minor change** to the forecast value. We also notice there is another line of investigating the robustness of forecasts: bounding the local Lipschitz constant [3]. In comparison, our definition is much stricter. Our strict definition might be more secure in practical applications because a minor change on the forecast value could lead to a completely different decision (e.g., the predicted growth rate of the stock price changes from -0.1\% to +0.1\%).
>
>
> > 6. The impact of certificate only for $f_{\rm{MIA}}(\cdot)$ instead of $f(\cdot)$
>
> We thank the reviewer for the valuable question.
> + **Why we derive the certificate only for $f_{\rm{MIA}}(\cdot)$**: since we have no requirement on $f(\cdot)$, it is almost impossible to derive its robusness certificate. We can derive the certificate of $f_{\rm{MIA}}(\cdot)$ for its unique nature.
> + **The advantage of certifying robustness for $f_{\rm{MIA}}(\mathbf{x}\_{1:t_0})$:** we can always certify the robustness for $f_{\rm{MIA}}(\mathbf{x}_{1:t_0})$ whatever the base model $f(\cdot)$ is.
> + **Limitations:** 1) $f_{\rm{MIA}}(\mathbf{x}\_{1:t_0})$ sometimes returns Abstain; 2) the forecast of $f_{\rm{MIA}}(\mathbf{x}_{1:t_0})$ could be inconsistent with $f(\mathbf{x}\_{1:t_0})$.
>
>
>
> > 7. It is good to see the comparison to randomized smoothing defenses. It is better to further justify on why this comparison is important.
>
> We thank the reviewer for the valuable comment.
> + **Why we compare MIA to randomized smoothing:** randomized smoothing [4] is a well-know model-agnostic method in the field of certified defenses, which has been applied to defend various types of attacks. Remarkably, it achieves superior certified robustness to other certified defenses in their respective fields, including $\ell_0/ \ell_1/\ell_2/\ell_\infty$-norm perturbations [5,6], image translation, Gaussian blur, rotation and scaling [7,8]. *For the widespread success of randomized smoothing, regarding randomize smoothing as a baseline is very natural*.
> + **Why we compare MIA to DS and RS:** to our best knowledge, there is no other model-agnostic $\ell_0$-norm certified defense with a comparable performance to DS and RS in the domain of image classification.
> + **The conclusions drawn from the comparison:** extensive results demonstrate that MIA outperforms these two baselines on robustness (quantified by certified accuracy/ MSE/ MAE). Moreover, MIA do not require retraining the base model, indicating that MIA is much more practical. To summarize, *MIA surpasses two baselines on robustness and practicality*.
>
> > 8. Is there any discussion on whether the proposed method useful for both univariate and multivariate time series?
>
> We thank the reviewer for the valuable comment.
> + **How to apply MIA to multivariate time series?** We can easily apply MIA to multivariate time series through repeating the process of univariate version of Masking (Step 1) and Imputing (Step 2) on each variable. Then we obtain a list of imputed multi-variate time series. Finally, we aggregate the labels/forecasts of all the imputed multi-variate time series in the same way as Aggregation (Step 3) for univariate time series.
> + **Additional experiments on multivariate tasks:** we evaluate MIA on four multi-variate time series datasets (ETTm2, ETTh2 [9], Illness [10] and Weather [11]), following the work of Autoformer[12]. MIA is evaluated with six forecasting model architectures (MLP-Mixer, MLP, LSTM, GRU, RNN and Transformer). The results are reported in Table 22 ~ 29 in Appendix. F in our revision. Our evaluation results demonstrate that MIA behaves similarly to that of univariate forecasting tasks.

---

> ### Author Response · Authors · 2022-11-12
> **Official Response to Reviewer k4J1 (Part 3)**
>
> >***Q3. The authors claim in most cases short-term events should not have large impacts on the long-term outcomes. Then the question is why resistance to the short-term perturbation is essential.***
>
> We thank the reviewer for the constructive question. We have replaced "in most cases short-term events should ..." with "in some cases ..." in our revision. We admit that in some cases the resistance to the short-term perturbation is not so important. However, in some long-term forecasting/prediction scenarios, the resistance to short-term perturbations is essential. For instance, if we want to forecast the future of a certain industry (e.g., new energy industry), since the future of an industry is believed unaffected by the short-term events, the resistance to short-term perturbations seems essential.
>
> >***Q4. The proposed method requires the masks to cover arbitrary temporally-localized perturbation of $L_{adv}$. It is better to provide some guidelines on how to ensure this requirement is satisfied.***
>
> We thank the reviewer for the valuable suggestion. We have formally proved that **for an arbitrary temporally-localized perturbation of $L_{atk}$, there always exists one of the masks generated by Masking （Step 1） that occludes it** in Appendix Section A.
>
> ---
>
> For ease of understanding, below we illustrate how our masks cover all temporally-localized perturbations with a toy example.
>
> 1) **Given $L_{adv}$, we can list the possible perturbations: $\boldsymbol{\delta}\_{[1\\; :  \\;1+L_{adv}]}, \boldsymbol{\delta}\_{[2\\; :  \\;2+L_{adv}]}, \ldots, \boldsymbol{\delta}\_{[t_0-L_{adv}\\; :  \\;t_0]}$.** Consider an example where the adversary attacks $\mathbf{x}\_{1:5}$ ($t_0=5$) subject to $L_{atk} = 2$. Then there are totally four possible perturbed series: **1)** $\mathbf{x}\_{1:5}+\boldsymbol{\delta}\_{[1:2]}$;&nbsp; **2)** $\mathbf{x}\_{1:5}+\boldsymbol{\delta}\_{[2:3]}$; &nbsp; **3)** $\mathbf{x}\_{1:5}+\boldsymbol{\delta}\_{[3:4]}$; &nbsp; **4)**  $\mathbf{x}\_{1:5}+\boldsymbol{\delta}\_{[4:5]}$.
>
> | $x_1+\delta_1$  | $x_2+\delta_2$ | $x_3$ | $x_4$ | $x_5$ |
> |----|----|----|----|----|
>
> | $x_1$  | $x_2+\delta_2$ | $x_3+\delta_3$ | $x_4$ | $x_5$ |
> |----|----|----|----|----|
>
> | $x_1$  | $x_2$ | $x_3+\delta_3$ | $x_4+\delta_4$ | $x_5$ |
> |----|----|----|----|----|
>
> | $x_1$  | $x_2$ | $x_3$ | $x_4+\delta_4$ | $x_5+\delta_5$ |
> |----|----|----|----|----|
>
> 2) **A $L_{mask}$-length mask is capable of covering $L_{mask} - L_{atk} + 1$ number of perturbations**. Back to the example, the mask $M\_{\rm{[1:3]}}$ can cover two perturbations, $\boldsymbol{\delta}\_{[1:2]}$ and $\boldsymbol{\delta}\_{[2:3]}$. The masked series $\mathbf{x}\_{1:5}+ \boldsymbol{\delta}\_{[1:2]} \odot M\_{\rm{[1:3]}}$ and $\mathbf{x}_{1:5}+ \boldsymbol{\delta}\_{[2:3]} \odot M\_{\rm{[1:3]}}$ are as follows:
>
> | $\color{#F00}{x_1+\delta_1 \rightarrow 0}$  | $\color{#F00}{x_2+\delta_2 \rightarrow 0}$ | $\color{#F00}{x_3 \rightarrow 0}$ | $x_4$ | $x_5$ |
> |----|----|----|----|----|
>
> |  $\color{#F00}{x_1 \rightarrow 0}$ | $\color{#F00}{x_2 + \delta_2 \rightarrow 0}$ | $\color{#F00}{x_3+\delta_3 \rightarrow 0}$ | $x_4$ | $x_5$ |
> |----|----|----|----|----|
>
> 3) **The step size to the next mask is $L_{mask}-L_{adv}+1$.** Back to our example, the step size is $L_{mask}-L_{adv}+1=2$, so the next mask is $M\_{[3:5]}$. Obviously $M\_{[3:5]}$ can cover the other two perturbations $\boldsymbol{\delta}_{[3:4]}, \boldsymbol{\delta}\_{[4:5]}$ as following:
>
> | $x_1$ | $x_2$ | $\color{#F00}{x_3+\delta_3 \rightarrow 0}$ | $\color{#F00}{x_4+\delta_4 \rightarrow 0}$  | $\color{#F00}{x_5 \rightarrow 0}$ |
> |----|----|----|----|----|
>
> | $x_1$ | $x_2$ | $\color{#F00}{x_3 \rightarrow 0}$ | $\color{#F00}{x_4+\delta_4 \rightarrow 0}$  | $\color{#F00}{x_5+\delta_5 \rightarrow 0}$ |
> |----|----|----|----|----|
>
>
> 4) Therefore, we slide the $L_{mask}$-size mask through $\mathbf{x}_{1:t_0}$ with the step size $L_{mask}-L_{adv}+1$, where any temporally-localized perturbation is guaranteed to be covered:
> \begin{equation}
> \begin{aligned}
>  &M_{[1 + k \alpha  \\; :  \\; \min(L\_{mask} + k \alpha, t_0) ]}, \;  k=0, \ldots, \lceil (t_0- L_{mask})/ \alpha \rceil \\
> &\text{where} \quad \alpha= L_{mask}-L_{adv}+1
> \end{aligned}
> \end{equation}
>
> ---
>  We hope this clarifies the reviewer’s problem. Please let us know if there is additional unclarity.

---

> ### Author Response · Authors · 2022-11-12
> **Official Response to Reveiwer k4J1 (Part 2)**
>
> >***Q2. Eq. (1), it is unclear why the norm is used as the time point index. Eq. 4, it is unclear on why should all M predictions be consistent instead of allowing some tolerance. Eq 5, 7, etc., are presented without introducing intuition.***
>
> We are deeply sorry for the unclarity in the technical part. We have updated them accordingly in our revision.
>
> + Eq. (1): Since $\ell_0$ norm of a series is the number of non-zero values in this series, we can represent an arbitrary temporally-localized perturbations of $L_{atk}$ with $\boldsymbol{\delta}\_{[t\_{adv}+1  \\; :  \\; t\_{adv}+ L\_{adv}]}, t\_{adv}=0,\ldots, t\_0-L\_{adv}$, of which the $\ell_0$ norm is exactly $L_{adv}$.
> + Eq. (4): Our aggregation does not allow any tolerance because the robustness of MIA would not hold if a disagreement is allowed. Note that, under our masking scheme, we can only guarantee there is a masked series that is unaffected by the perturbation. Meanwhile, the other masked series all retain the perturbed area (these masked series are considered to be under the adversary's control). If we allow a disagreer, the ensemble prediction is totally under the adversary's control, because all except one masked series are under the adversary's control (the unaffected masked series would become that disagreer).
> + Eq. (5): $f_{\rm{dis}} (\mathbf{x}) = \Delta \cdot \lfloor f(\mathcal{x}\_{1:t_0})/ \Delta \rfloor$ formulates the discretization. For instance, if $\Delta=0.5$, then $\Delta\cdot \lfloor f(\mathcal{x}\_{1:t_0})/ \Delta \rfloor$ is to round up $f(\mathcal{x}\_{1:t_0})$ to the integer. We specifically adopt discretization is for the fact that it is almost impossible for the forecasts of different imputed series to reach agreement unanimously.
> + Eq. (7): The loss function of training the imputation model is the average MSE loss of all the masked series with Gaussian augmentation. We compute the average loss among all the masked series generated in Masking (Step 1) because we only require to the imputation model to impute these series. We adopt Gaussian augmentation to deal with the random noise in the time series data and avoid overfitting.
>
> We hope this clarifies the reviewer’s concern. Please let us know if there is additional unclarity and we really look forward to further improving our paper based on the suggestions.

---

> ### Author Response · Authors · 2022-11-12
> **Official Response to Reviewer k4J1 (Part 1)**
>
> We thank the reviewer for insightful comments and suggestions. Our new ICLR submission is an updated version which has addressed several concerns from the reviews such as the typos, unclear presentation of the technical part, etc. In addition, in our current version, we include the experiments of evaluating MIA on multivariate forecasting datasets. Next we provide detailed answers to the questions.
>
>
>
> >***Q1. The paper could provide some examples to illustrate why detect $\ell_0$ norm localized perturbation in time series is practically useful, compared to $\ell_2$ norm perturbation.***
>
>
> Detecting $\ell_0$-norm localized perturbation is meaningful for:
> 1) For the **temporal nature** of time series data, we naturally worry about $\ell_0$ norm adversarial perturbation in time series data.
> 2) Temporally-localized perturbation can represent **short-term volatility**, as short-term volatility can be regarded as the normal data added with a localized perturbation. The resistance to short-term volatility is often necessary in long-term forecasting/prediction. A typical example is "Value Investing"[1], where the "intrinsic value" of a business is considered to be invariant under short-term volatility.
> 3) **Local anomaly detection** can be regarded as a sub-problem of detecting temporally-localized perturbation, which is practically useful in many real-world scenarios, such as abnormal traffic detection, IoT device monitoring, etc.
> 4) We empirically show $\ell_0$-norm perturbation can change the forecasts of undefended forecasters greatly in Appendix B.3.
> 5) We compare the attacking performance of $\ell_0$-norm perturbation to $\ell_0$-norm perturbation. The empirical results (Table 1, Table 2, Table 3) suggest that **forecasting models might be more sensitive to $\ell_0$-norm perturbations**.
>
> Table 1. (Electricity) Compare $\ell_0$-norm localized perturbation to $\ell_2$-norm perturbation (computed by the algorithm [2]) on the MSE between the original forecast and the perturbed forecast. The table reports the relative improvement of the $\ell_0$-norm perturbation over the $\ell_2$-norm perturbation (averaging among $128$ randomly selected samples). **The positive value indicates that our $\ell_0$-norm perturbation outperforms $\ell_0$-norm perturbation**. For fairness, the $\ell_0$ or $\ell_2$ norm of the perturbation is restricted to be no larger than $\boldsymbol{\beta} \times$ the average value among the $\ell_0$ or $\ell_2$ norm of all the testing samples. Values in tables are calculated as $(MSE_{\ell_0} - MSE_{\ell_2}) / MSE_{\ell_2}$.
>
> |  Model $\downarrow \quad$ $\boldsymbol{\beta} \rightarrow$| 10% | 20% | 30% | 40% | 50% |
> |:----:|:----:|:----:|:----: |:----:|:----:|
> |MLP-Mixer | +769.9 % | +89.5 % | +73.3 % | +12.3 % | +53.1 \%|
> |GRU | +2.5 % | -1.6 % | -8.3 % | -3.3 % | -6.5 \%|
> |LSTM | +23.1 % | +15.1 % | -33.5 % | -14.8 % | +2.0 \%|
> |MLP | +265.0 % | +376.3 % | +211.6 % | +114.1 % | +58.3 % |
>
> Table 2. (Exchange) Compare $\ell_0$-norm localized perturbation to $\ell_2$-norm perturbation on relative improvement on MSE.
> |  Model $\downarrow \quad$ $\boldsymbol{\beta} \rightarrow$| 10% | 20% | 30% | 40% | 50% |
> |:----:|:----:|:----:|:----: |:----:|:----:|
> |MLP-Mixer |  +1000.1 % | +332.2 % | +114.7 % | +131.7 % | +84.6 % |
> |GRU |  -48.6 % | -45.1 % | -39.7 % | -31.7 % | -20.9 |
> |LSTM | -8.8 % | -28.3 % | -24.9 % | -16.2 % | -6.6 \% |
> |MLP | +528.9 % | +101.8 % | +36.9 % | +8.8 % | +6.2 \% |
>
> Table 3. (Traffic) Compare $\ell_0$-norm localized perturbation to $\ell_2$-norm perturbation on relative improvement on MSE.
> |  Model $\downarrow \quad$ $\boldsymbol{\beta} \rightarrow$| 10% | 20% | 30% | 40% | 50% |
> |:----:|:----:|:----:|:----: |:----:|:----:|
> |MLP-Mixer | +3310.1 % | +841.8 % | +328.5 % | +317.7 % | +205.5 \%|
> |GRU | +147.0 % | +28.2 % | +7.3 % | -4.7 % | +21.6 \% |
> |LSTM | +239.0 % | +66.8 % | +14.9 % | +2.5 % | -2.8 \%|
> |MLP | +1760.4 % | +145.7 % | +38.4 % | +7.5 % | -7.6 \% |

---

> > ### Comment · Reviewer_k4J1 · 2022-12-05
> > **Thanks for the response**
> >
> > Thank the authors for the response, which has addressed some of my questions. From the paper and the response, the significance to study $l_0$ norm localized perturbation is still obscure to me. It looks to be stemmed from the observation that $l_2$ norm robustness has been studied on forecasting models, but $l_0$ norm was not, rather than a real application that demonstrates the importance to detect such perturbations. Thus it is a sort of imaginary problem to some extent, and makes the experiments rely on the simulated data. The tables in the response demonstrate the effectiveness of the method on detecting such perturbations, but not to answer the question. However, I understand it could be potentially useful, but it is obscure for now. In the response to Q3, I am not clear why the resistance to short-term perturbations seems essential when the future of an industry is believed unaffected by the short term events. Regarding Q5, the obscure thing is in time series forecasting how minor the change could lead to a difference in the decisions in the forecasting method. From the response to Q8, it seems the method could not process multivariate time series directly so that the interactions between different variables are captured during representation learning. When comparing the method with the baseline methods, did the baseline methods process the multivariate time series in the univariate manner? It is better to let them directly embed multivariate time series if possible.

---

> > > ### Author Response · Authors · 2022-12-06
> > > **Thank you for your feedback and our further explanations (Part II)**
> > >
> > > > ***Q10. I am not clear why the resistance to short-term perturbations seems essential when the future of an industry is believed unaffected by the short-term events.***
> > >
> > > Sorry for the confusion. We just intuitively believe that the future of a certain industry, such a huge thing should not be affected by short-term events.
> > >
> > > > ***Q11. The obscure thing is in time series forecasting how minor the change could lead to a difference in the decisions in the forecasting method***
> > >
> > > Thanks for the question. It depends on the forecasting task and the original forecasting value. The attacker can perturb those forecasts that are close to the decision boundary, making them cross the decision boundary to change the final decisions. Consider a buy/sell decision-making example, where our decision on whether to buy or sell the stock is based on whether our forecast about the future stock price is larger than the threshold $\gamma$ or not. If the original forecast is $\gamma+0.01$, the attacker could perturb the forecast to be $\gamma-0.01$ to change the decision.
> > >
> > > > ***Q12. When comparing the method with the baseline methods, did the baseline methods process the multivariate time series in the univariate manner? It is better to let them directly embed multivariate time series if possible.***
> > >
> > > Thank you for the comment and we believe there are some misunderstandings here. We clarify that both the baseline and MIA process the input multivariate series **in the way of embedding the whole multivariate series**.
> > >
> > > To be specific, denote $\mathcal{X} = [\mathbf{x}^{(1)}\_{1:t_0}, \\; \mathbf{x}^{(2)}\_{1:t_0}, \\;  \ldots, \\; \mathbf{x}^{(d)}\_{1:t_0}]^T \in \mathbb{R}^{d \times t_0}$ the input multivariate series. MIA processes the multivariate series $\mathcal{X}$ as follows:
> > >
> > > * **Multivariate Masking:** we mask $\mathcal{X}$ with a multivariate mask (denoted by $\mathcal{M}_{[u:v]}= [M\_{[u:v]} \times d ]^T $). The masked multivariate series is $\mathcal{X} \odot \mathcal{M}\_{[u:v]}= [\mathbf{x}^{(1)}\_{1:t_0} \odot M\_{[u:v]}, \\; \mathbf{x}^{(2)}\_{1:t_0} \odot M\_{[u:v]} , \\; \ldots, \\; \mathbf{x}^{(d)}\_{1:t_0} \odot M\_{[u:v]}]^T$
> > >
> > > * **Multivariate Imputing:** For the multivariate task, the imputation model $G(\cdot): \mathbb{R}^{d \times t_0} \rightarrow \mathbb{R}^{d \times t_0}$ takes a multivariate time series as input, and output a multivariate time series, which does not process the multivariate series in a univariate manner.
> > >
> > > * **Aggregation:** The only difference in the aggregation step is that we take a multivariate time series classifier $f(\cdot): \mathbb{R}^{d \times t_0} \rightarrow \mathbb{R}^{t_0}$ as the base model, instead of taking a univariate classifier.
> > >
> > > Accordingly, MIA does not process the multivariate time series in a univariate manner, as both the imputation model and base classifier take multivariate series as input. Sorry again for the misunderstandings that our rebuttal may cause you and we will add more details in our revision.
> > >
> > >
> > > We are also happy to address your further questions and concerns before the rebuttal ends.

---

> > > ### Author Response · Authors · 2022-12-06
> > > **Thank you for your feedback and our further explanations (Part I)**
> > >
> > > We greatly appreciate your feedback on our rebuttal and the further insightful questions and comments. Please kindly find our explanations about the remaining concerns as follows
> > >
> > > > ***Q9. The significance of detecting $\ell_0$-norm localized perturbation***
> > >
> > > We thank the reviewer for the valuable question. **$\ell_0$-norm localized perturbation have the corresponding counterpart, *subsequence anomaly*, in the field of time-series anomaly detection, since *subsequence anomaly* can be considered to be a normal data added with a $\ell_0$-norm localized perturbation.** Although the notion of $\ell_0$-norm localized perturbation might be rarely mentioned, its counterpart *subsequence anomaly* has been studied for a long time in the field of time-series anomaly detection[1-4]. In particular, with our proposed method MIA, we could roughly localize the anomalous subsequence by analyzing the disagreer masked series, which is especially useful in the healthcare domain, IoT-enabled manufacturing and software tools. Here are some promising real-world applications for $\ell_0$-norm localized perturbation detection.
> > >     1. We can adopt $\ell_0$-norm localized perturbation detection to detect the heart anomaly in Electrocardiograms (ECGs) datasets [5] (a time series of the electrical potential between two points on the surface of the body caused by a beating heart, which are arguably ***the most important time series*** [6]). MIA allows us to localize which period causes
> > >     2. We can apply the $\ell_0$-norm localized perturbation detection to IoT-enabled manufacturing [7], in order to localize the abnormal states of the IoT devices.
> > >     3. We can apply the $\ell_0$-norm localized perturbation detection to identify the anomaly in the user activities and server logs, in order to alert for the potential incidents [8].
> > >
> > > Accordingly, detecting $\ell_0$-norm localized perturbation is useful. We will add the experiments on real-world datasets in our final version, including biomedical dataset ECGs and software dataset Yahoo [9] in our final version.
> > >
> > >
> > > [1] Aggarwal, Charu C. "An introduction to outlier analysis." Outlier analysis. Springer, Cham, 2017. 1-34.
> > >
> > > [2] Singh, Karanjit, and Shuchita Upadhyaya. "Outlier detection: applications and techniques." International Journal of Computer Science Issues (IJCSI) 9.1 (2012): 307.
> > >
> > > [3] Ukil, Arijit, et al. "IoT healthcare analytics: The importance of anomaly detection." 2016 IEEE 30th international conference on advanced information networking and applications (AINA). IEEE, 2016.
> > >
> > > [4] Gupta, Manish, et al. "Outlier detection for temporal data: A survey." IEEE Transactions on Knowledge and data Engineering 26.9 (2013): 2250-2267.
> > >
> > > [5] http://www.physionet.org/challenge/2015/
> > >
> > > [6] Keogh, Eamonn, et al. "Finding unusual medical time-series subsequences: Algorithms and applications." IEEE Transactions on Information Technology in Biomedicine 10.3 (2006): 429-439.
> > >
> > > [7] Zhan, Peng, et al. "Temporal anomaly detection on IIoT-enabled manufacturing." Journal of Intelligent Manufacturing 32.6 (2021): 1669-1678.
> > >
> > > [8] Ren, Hansheng, et al. "Time-series anomaly detection service at microsoft." Proceedings of the 25th ACM SIGKDD international conference on knowledge discovery & data mining. 2019.
> > >
> > > [9] https://yahooresearch.tumblr.com/post/114590420346/a-benchmark-dataset-for-time-series-anomaly

---

### Official Review · Reviewer_F33Z · 2022-10-25

**Confidence:** 4
**Correctness:** 2
**Technical Novelty And Significance:** 2
**Empirical Novelty And Significance:** 2
**Recommendation:** 5

**Clarity, Quality, Novelty And Reproducibility:**

Clarity and Quality
- As mentioned in the first and second weaknesses, several drawbacks exist.
- This is just a question, but even if a mask hides the perturbed area, as this paper claims, it seems to make little sense since it is aggregated with the results from other masked time series where the perturbed area remains. How do the authors think the assumption makes sense?

Novelty
- Masking some parts of data is well-known to the community for self-supervised learning and data augmentation.

Reproducibility
- Sufficient reproducibility is expected. The protocol description of the experiment is detailed, and the source code is provided.


**Strength And Weaknesses:**

Strengths
1. Extensive experimental results are reported using multiple methods such as MLP, RNN, CNN, Transformer, and MLP-Mixer on multiple datasets of time series data.

Weaknesses
1. The authors should proofread the manuscript repeatedly. There are examples found in Section 1.
  - ell_0-robustness -> \ell_0-robustness
  - guaranteeing the robustness for each prediction mathematically. -> The first letter of the word at the beginning of a sentence should be capitalized.
  - BDPA should be first introduced by its correct name, not the abbreviation.
  - The authors should break paragraphs before describing the first contribution point.
1. Mathematical explanations are also hard to follow.
  - In equation (1), what does it mean that the delta index decreases as the x index increases?
  - Most critical point is that the definition of the mask in equation (2) is unclear; the reader is ultimately forced to imagine it. Surprisingly, there are three different capital M's in equation (2) and Algorithm 1: Roman, Italic, and Calligraphic Ms. As for Roman's M and Calligraphy's M, their definitions cannot be understood. Since the purpose is to mask the time-series data, the reviewer can only imagine that Ms are series data, with some being 1 and others being 0; the indexes for the value 0 are not stated clearly. Some figures showing mask patterns would be helpful.
1. (Yoon et al., 2022) is cited but not compared to the proposed method in the experimental results.
1. The second paragraph of the introduction makes the point about heuristic defense methods, arguing that a certified defense is needed to end the cat-and-mouse game. However, the algorithm in this paper also assumes that only a local part of the time series data is perturbated. Would the cat-and-mouse game continue even with the proposed method when other types of attacks are considered?

**Summary Of The Paper:**

This paper proposes a plug-and-play method for robust estimation of perturbed data in time series identification and prediction. The proposed method assumes perturbations in a part of time series data and performs estimation by aggregating multiple masked and imputed time series data outputs. Experimental results with multiple data sets and multiple machine learning models are reported.

**Summary Of The Review:**

Overall, the reviewer leans toward rejecting this paper. A response from the authors regarding the above weaknesses could improve the score.

---

> ### Author Response · Authors · 2022-11-12
> **Official Response to Reviewer F33Z**
>
> We thank the reviewer for the valuable comment and sincerely apologize for our typos. We provide our responses below.
>
>
> >***Q1. In Eq. (1), what does it mean that the $\delta$ index decreases as $x$ index increases?***
>
> This is a typo in Eq. (1) and we are deeply sorry. We have revised Eq. (1) in our paper.
>
>
> > ***Q2. Unclear definition of mask in Eq. (2).***
>
> We sincerely apologize for the inconsistent notation for mask. For clarity, we redefine the mask in Eq. (2). We denote mask by $M_{[u : v]}$ and the masking operation by $\mathbf{x}\_{1:5} \odot M\_{[u : v]}$, which is to remove out the values of the period $[u : v]$.
>
> For clarity, we present an example:
>
> 1. Consider a time series $\mathbf{x}_{1:5}$:
> | $x\_1$ | $x\_2$ | $x\_3$ | $x\_4$ | $x\_5$ |
> |----|----|----|----|----|
>
> 2. The mask $M_{[2 : 4]}$ is:
> | 1 | 0 | 0 | 0 | 1 |
> |----|----|----|----|----|
>
> 3. The masked series $\mathbf{x}\_{1:5} \odot M\_{[2:4]}$ is:
> | $x\_1$ | 0 | 0 | 0 | $x\_5$ |
> |----|----|-----|----|----|
>
> >***Q3. (Yoon et al., 2022) is not compared to the proposed method in the experimental results.
>
> We thank the reviwer for pointing this out. The reason is that the fundamentally different objectives of MIA and (Yoon et al., 2022). MIA focus on $\ell_0$-norm robustness while (Yoon et al., 2022) focus on $\ell_2$-norm robustness. Comparing a $\ell_0$-norm defense to a $\ell_2$-norm defense is hard because *there lacks a standard that how much $\ell_0$ norm is equal to one unit of $\ell_2$ norm*.
>
> >***Q4. Would the cat-and-mouse game continue even with the proposed method when other types of attacks are considered?***
>
> We thank the reviewer for the insightful question, and we agree that **MIA can hardly defend other types of attacks (e.g., $\ell_2$-norm perturbations), because MIA is only designed for defending temporally-localized perturbations**. However, that does not reduce the significance of MIA. We point out that all the current multi-attack defenses are based upon the development of single-attack defenses (e.g., the universal defense [1] is based on the specific $\ell_p$-perturbation defense [2]. The universal defense [3] is an extension of randomized smoothing [4]). Since MIA is the first $\ell_0$-norm defense in the domain of time series, we believe it would facilitate developing multi-attack defenses for time-series models.
>
> >***Q5. Even if a mask hides the perturbed area, it seems to make little sense since it is aggregated with the results from other masked time series where the perturbed area remains.***
>
> We thank the reviewer for the thoughtful question. We first point out that MIA uses the one-veto based aggregation, instead of the commonly-used majority-vote based aggregation. With one-veto based aggregation, **the masked time series of which the perturbed area remains could only change the output of MIA classifier to be Abstain, but cannot change the prediction to be another label**. Specifically, we consider two cases of $f_{MIA}(\mathcal{x}_{1:t_0})$.
> + Case 1: $f_{\rm{MIA}}(\mathcal{x}_{1:t_0})$ returns a label, then the label must be consistent with the prediction on that unaffected masked series. **The label is irrelevant with the perturbation**
> + Case 2: $f_{\rm{MIA}}(\mathcal{x}_{1:t_0})$ returns Abstain, alerting us of a possible temporally-localized perturbation.
>
> **References**
>
> [1] Croce, Francesco, and Matthias Hein. "Provable robustness against all adversarial $\ell_p$-perturbations for $p\geq 1$." International Conference on Learning Representations. 2020.
>
> [2] Croce, Francesco, Maksym Andriushchenko, and Matthias Hein. "Provable robustness of relu networks via maximization of linear regions." the 22nd International Conference on Artificial Intelligence and Statistics. PMLR, 2019.
>
> [3] Hong, Hanbin, Binghui Wang, and Yuan Hong. "UniCR: Universally Approximated Certified Robustness via Randomized Smoothing." European Conference on Computer Vision. Springer, Cham, 2022.
>
> [4] Cohen, Jeremy, Elan Rosenfeld, and Zico Kolter. "Certified adversarial robustness via randomized smoothing." International Conference on Machine Learning. PMLR, 2019.

---

### Author Response · Authors · 2022-11-12
**Paper Revision Summary**

We thank all the reviewers for their insightful feedback and suggestions for improving the paper. We are glad that the reviewers found our paper found our paper studies a new and important problem, novel, contains comprehensive comparison to two baselines and extensive experimental evaluation. Below is a summary of major paper updates:

1. **[Section 1]** Adjust our presentation and fix some typos, following Reviewer $\color{Blue}{\rm{F33Z}}$, Reviewer $\color{Green}{\rm{K4J1}}$'s suggestions.
1. **[Section 3 - Definition 1]** Revise Eq. (1) in Definition 1 and change the **notation of perturbation** ($\boldsymbol{\delta}_{u:v}$ denotes the perturbation series and $\delta_t$ denotes a single perturbation value added to the time point $x_t$), following Reviewer $\color{Blue}{\rm{F33Z}}$, Reviewer $\color{Green}{\rm{K4J1}}$ and Reviewer $\color{Maroon}{\rm{SiPr}}$'s suggestions.
2. **[Section 3]** Add a detailed discussion on the significance of temporally-localized perturbation, following Reviewer $\color{Green}{\rm{K4J1}}$'s suggestions.
3. **[Section 4.1 - Eq. (2)]** Clarify the notation of mask in Eq. (2), following Reviewer $\color{Blue}{\rm{F33Z}}$, Reviewer $\color{Green}{\rm{K4J1}}$ and Reviewer $\color{Maroon}{\rm{SiPr}}$'s suggestions.
4. **[Section 4.1 - Eq. (3)]** Revise the typo in Eq. (3) following Reviewer $\color{Maroon}{\rm{SiPr}}$'s suggestions.
5. **[Section 4.3 - Remark 2]** Add a discussion on why MIA aggregation cannot tolerate a disagreement, following $\color{Green}{\rm{K4J1}}$'s suggestions.
6. **[Section 4.3, Appendix F]** Provide additional experiments on evaluating MIA on multivariate forecasting tasks, following $\color{Green}{\rm{K4J1}}$'s suggestions.
7. **[Section 4.4]** Add a discussion on why we adopt Gaussian augmentation when training imputation models, following Reviewer $\color{Maroon}{\rm{SiPr}}$'s suggestions.
8. **[Section 4.4, Appendix G.2]** Provide additional experiments on evaluating the impact of Gaussian augmentation, following Reviewer $\color{Maroon}{\rm{SiPr}}$'s suggestions.
9. **[Section 4.5]** Revise $k+ L_{adv} -1$ to $2 (k+ L_{adv} -1)$, following Reviewer $\color{Maroon}{\rm{SiPr}}$'s suggestions.
10. **[Section 4.5]** Add the explanation on why the comparison to two baselines is important, following $\color{Green}{\rm{K4J1}}$'s suggestions.
11. **[Appendix F]** Add the discussion and empirical evaluation on applying MIA to multivariate time series, following $\color{Green}{\rm{K4J1}}$'s suggestions.


Please also let us know if there are other questions, and we are really looking forward to the discussion with the reviewers to further improve our paper. Thanks!

---

### Author Response · Authors · 2022-11-16
**Dear Reviewers:**

Thank you again for your wisdom and valuable comments. We have provided experimental or complete explanations for all the questions. Since the discussion period is approaching its end, we would be glad to hear from you whether our rebuttal has addressed your concerns. Feel free to comment on our rebuttal if you have further questions and considerations.

---

### Author Response · Authors · 2022-11-18
**Dear Reviewers:**

Approaching the pdf updating DDL, is there anything needing added?

---

### Author Response · Authors · 2022-11-26
**Dear Reviwers:**

Since the discussion period is approaching its end, we would be glad to hear from you whether our rebuttal has addressed your concerns. Please feel free to comment on our rebuttal!

---

### Decision · Program_Chairs · 2023-01-20

**Decision:**

Reject

**Justification For Why Not Higher Score:**

Application is narrow and theoretical results are not significant.

**Justification For Why Not Lower Score:**

N/A

**Metareview: Summary, Strengths And Weaknesses:**

The authors develop a framework for detecting and certifying robustness to perturbations to time series inputs that are localized in time with an l0 norm constraint. The motivation for the thread model comes from subsequence anomaly detection, a notion that has been previously studied in the time series literature.

Strengths:
1. New framework for robustness certification for l0 localized perturbations to time series models.
2. Empirical results demonstrating effectiveness of the framework.

Weaknesses:
1. The contributions relative to prior work on the theory of randomized smoothing is minimal.
2. The application to subsequence anomaly detection seems highly specific and may not be appropriate for the ICLR conference.


**Summary Of Ac-Reviewer Meeting:**

No meeting